# Predictive Uncertainty Quantification for Bird's Eye View Segmentation: A Benchmark and Novel Loss Function

**Linlin Yu** [†*]**, Bowen Yang** [‡*] **, Tianhao Wang**[†]**, Kangshuo Li**[†]**, Feng Chen**[†]

[†]Department of Computer Science, The University of Texas at Dallas, Richardson, TX, USA
[‡]Cypress Woods High School, Cypress, TX , USA
[†]{linlin.yu, tianhao.wang, kangshuo.li, feng.chen}@utdallas.edu,
[‡] bowenymail@gmail.com

## Abstract

The fusion of raw sensor data to create a Bird's Eye View (BEV) representation is critical for autonomous vehicle planning and control. Despite the growing interest in using deep learning models for BEV semantic segmentation, anticipating segmentation errors and enhancing the explainability of these models remain underexplored. This paper introduces a comprehensive benchmark for predictive uncertainty quantification in BEV segmentation, evaluating multiple uncertainty quantification methods across three popular datasets with three representative network architectures. Our study focuses on the effectiveness of quantified uncertainty in detecting misclassified and out-of-distribution (OOD) pixels while also improving model calibration. Through empirical analysis, we uncover challenges in existing uncertainty quantification methods and demonstrate the potential of evidential deep learning techniques, which capture both aleatoric and epistemic uncertainty. To address these challenges, we propose a novel loss function, Uncertainty-Focal-Cross-Entropy (UFCE), specifically designed for highly imbalanced data, along with a simple uncertainty-scaling regularization term that improves both uncertainty quantification and model calibration for BEV segmentation.

## 1 Introduction

Bird's Eye View (BEV) semantic segmentation is a critical component of modern vehicular technology and has received increased attention in recent years. It has been adopted in advanced autonomous vehicle systems, such as Tesla's Autopilot. BEV representations offer a top-down perspective of the environment surrounding a vehicle, created by fusing data from multiple sensors such as cameras, LiDAR, and radar (Philion & Fidler, 2020). This comprehensive view allows autonomous systems to accurately perceive the position and movement of nearby objects, including other vehicles, pedestrians, and obstacles. Therefore, BEV semantic segmentation (BEVSS) plays a vital role in both autonomous driving systems and advanced driver-assistance systems (ADAS) (Liu et al., 2023).

Identifying potential errors before they lead to dangerous outcomes is crucial for ensuring the safety and reliability of BEVSS. However, Deep Neural Networks (DNNs) commonly used for BEV representation learning tend to make overconfident predictions on unseen data (Guo et al., 2017) and underconfident predictions on noisy data (Wang et al., 2021). In 2018, an Uber autonomous vehicle in Arizona failed to correctly identify a pedestrian crossing outside a designated crosswalk at night, resulting in a fatal collision (Goodman, 2018). In March 2022, a Tesla Model S on Autopilot crashed into a stationary vehicle on a Florida highway, injuring five police officers (Shepardson, 2022).

Uncertainty prediction in segmentation can enable autonomous systems to return control to the driver when necessary. There are two primary types of uncertainty (Kendall & Gal, 2017). *Aleatoric uncertainty*, which arises from inherent randomnesses, such as noisy data and labels. By identifying areas with high aleatoric uncertainty, the vehicle can make better decisions, especially in complex or

---

[*] Equal contribution.

ambiguous situations. *Epistemic uncertainty*, on the other hand, stems from a lack of knowledge, such as when the test-time input differs significantly from the training data. Quantifying epistemic uncertainty helps the system handle out-of-distribution scenarios and adapt to unexpected conditions, especially in dynamic driving environments.

Uncertainty quantification methods can be broadly categorized based on their computational complexity and approach. The first category involves multiple forward passes to estimate a model's predictive uncertainty, including methods such as deep ensembles (Lakshminarayanan et al., 2017) and dropout-based approaches (Gal & Ghahramani, 2016). While effective, these methods are computationally expensive, making them impractical for real-time applications. The second category utilizes deterministic single forward-pass neural networks. Conjugate-prior-based methods, like evidential neural networks (Sensoy et al., 2018), predict a conjugate prior distribution of class probabilities, providing multi-dimensional uncertainty estimates. Additionally, post-hoc methods, such as those based on softmax (Hendrycks & Gimpel, 2016) and energy models (Liu et al., 2020), are notable for their ease of adoption without requiring modifications to the underlying network architecture,

This study investigates the uncertainty-aware BEVSS task, which involves both pixel-level classification and the associated uncertainty estimations. Our main contributions are :

- We introduce the first benchmark for evaluating uncertainty quantification methods in BEVSS, analyzing five representative approaches (softmax entropy, energy, deep ensemble, dropout, and evidential) across three popular datasets (CARLA (Dosovitskiy et al., 2017), nuScenes (Caesar et al., 2020), and Lyft (Kesten et al., 2019)) using three BEVSS network architectures (Lift-Splat-Shoot (Philion & Fidler, 2020), Cross-View-Transformer (Zhou & Krähenbühl, 2022), and Simple-BEV (Harley et al., 2023)).

- We propose the UFCE loss, which we theoretically demonstrate can implicitly regularize sample weights, mitigating both under-fitting and over-fitting. In addition, we introduce a simple uncertainty scaling regularization term that explicitly adjusts sample weights based on epistemic uncertainty.

- Extensive experiments demonstrate that our proposed framework consistently achieves the best epistemic uncertainty estimation, improving the AUPR for OOD detection by an average of 4.758% over the runner-up model. Additionally, it delivers top-tier aleatoric uncertainty performance, as evaluated through calibration and misclassification detection, all while maintaining high segmentation accuracy [1].

## 2 UNCERTAINTY QUANTIFICATION ON BEVSS

**Problem Formulation**. Suppose we are given $n$ images from RGB camera views surrounding the ego vehicle. Let $\mathbf{X} := \{\mathbf{X}_k, \mathbf{E}_k, \mathbf{I}_k\}_{k=1}^n$ denote the input, where each camera view has a feature matrix $\mathbf{X}_k \in \mathbb{R}^{3 \times H \times W}$ (with $H$ and $W$ representing the height and width of the input image), an extrinsic matrix $\mathbf{E}_k \in \mathbb{R}^{3 \times 4}$, and an intrinsic matrix $\mathbf{I}_k \in \mathbb{R}^{3 \times 3}$.

The goal of uncertainty-aware BEV semantic segmentation is to predict pixel-level classes in the BEV coordinate frame, represented by $\mathbf{Y} \in \{0, 1\}^{C \times M \times N}$, along with aleatoric uncertainty $u_{i,j}^{alea}$ and epistemic uncertainty $u_{i,j}^{epis}$ for each pixel indexed by $(i, j)$. Here, $C$ denotes the number of classes, and $M$ and $N$ represent the height and width of the BEV frame, respectively.

**Network architecture**. We directly apply the network architecture from the well-established BEVSS task. A common BEV semantic segmentation neural network has the general form:

$$\mathbf{P} = \sigma_{\text{softmax}}\left(f(\mathbf{X}; \boldsymbol{\theta})\right), \tag{1}$$

where $\mathbf{P} \in [0, 1]^{C \times M \times N}$ are the pixel-wise class probabilities, and $\boldsymbol{\theta}$ refers to the network parameters. We use $\mathbf{p}_{i,j}$ to denote the class-probability vector of the BEV pixel at index $(i, j)$.

Uncertainty quantification methods usually apply post-processing techniques or introduce slight modifications to the network architecture, training, or inference strategy. In our experiments, we consider three network architectures: Lift-Splat-Shoot (LSS) (Philion & Fidler, 2020), Cross-View Transformer (CVT) (Zhou & Krähenbühl, 2022), and Simple-BEV (Harley et al., 2023). LSS lifts 2D

---

[1]The code is available at https://github.com/bluffish/ubev

camera images into a 3D space, projects them onto a BEV plane, and then processes the BEV features for semantic segmentation. CVT, on the other hand, leverages attention mechanisms to transform multi-view image inputs into a unified BEV representation, enabling more effective cross-view feature fusion. Simple-BEV defines a 3D coordinate volume over the BEV plane and projects each coordinate into the corresponding camera image. Details are provided in Appendix A.2.1.

**Uncertainty Quantification Benchmark**: In this benchmark, we aim to investigate the performance of various uncertainty quantification methods on the uncertainty-aware BEVSS task.

We apply five representative uncertainty quantification methods in the deep learning domain. One widely used metric is *entropy* (Hendrycks & Gimpel, 2016), which captures aleatoric uncertainty for in-distribution samples and is preferred due to its simplicity and low computational overhead. Additionally, Liu et al. (2020) proposed the use of an *energy* score to distinguish between in-distribution and OOD samples, which is also a post-hoc method. We also consider the *ensemble-based* method (Lakshminarayanan et al., 2017), which estimates uncertainty by training multiple models and averaging their predictions. Similarly, the *dropout-based* method (Gal & Ghahramani, 2016) approximates Bayesian inference by performing dropout during inference. These two methods generate multiple class probability predictions, and the variance is used as an epistemic uncertainty metric. These four methods do not require significant architectural changes to the network. In contrast, the *evidential neural network (ENN)* (Sensoy et al., 2018) replaces the standard softmax activation with ReLU function and produces the multinomial opinions in subjective logic. Although this approach modifies the model architecture, it requires only a single training and inference pass to generate both aleatoric and epistemic uncertainty, along with the class probabilities. Detailed descriptions of the models and uncertainty calculations are presented in Appendix A.2.3.

**Evidential Neural Networks (ENN)**: Due to computational efficiency and explainability, we extend ENNs to conduct uncertainty quantification for BEVSS, where pixel-level prediction is expected. We replace the last softmax in BEVSS network architecture (LSS, CVT or Simple-BEV) with the ReLU activation function to predict concentration parameters of a non-degenerate Dirichlet distribution.

$$\mathbf{A} = \sigma_{\text{ReLU}} \left( f(\mathbf{X}; \boldsymbol{\theta}) \right) + \mathbf{1}, \mathbf{A} \in \mathbb{R}^{+^{C \times M \times N}} \tag{2}$$

We assume that the target class label is drawn from a categorical distribution parameterized by probabilities $\boldsymbol{p}_{i,j}$. In this case, we can probabilistically model these probabilities using their conjugate prior, the Dirichlet distribution, which itself is parameterized by the concentration parameters $\boldsymbol{\alpha}_{i,j} := \mathbf{A}[:, i, j]$. For simplicity, we omit the index $(i, j)$ in the subsequent discussion.

$$\mathbf{y} \sim \text{Cat}(\mathbf{y}|\mathbf{p}), \ \mathbf{p} \sim \text{Dir}(\boldsymbol{p}|\boldsymbol{\alpha}). \tag{3}$$

The expected class probability can be derived as shown in Equation 4, where $\alpha_0 = \sum_{c=1}^{C} \alpha_c$:

$$\bar{\mathbf{p}} := \mathbb{E}[\mathbf{p}|\boldsymbol{\alpha}] = \boldsymbol{\alpha}/\alpha_0 \tag{4}$$

Based on subjective logic opinion (Jøsang, 2016), there is a bijection between subjective opinions and Dirichlet PDFs, allowing a C-dimensional Dirichlet probability distribution to represent a multinomial opinion. Intuitively, we introduce the concept of "evidence", defined as a metric indicating the volume of supportive observations gathered from training data which suggests a sample belongs to a specific class. Let $e_c \geq 0$ represent the evidence for class $c$, with $e_c = \alpha_c - 1$. Higher evidence demonstrates stronger confidence in classifying a sample into the corresponding category, whereas lower overall evidence across all classes suggests a lack of similarity with the training data, indicating a higher likelihood of the sample being out-of-distribution.

Then we discuss the optimization loss for evidential-based models. UCE loss associated with an entropy regularizer is commonly used in evidential-based models (Sensoy et al., 2018; Charpentier et al., 2020; Li et al., 2024), With a training dataset $\mathcal{D}^{train}$,

$$\mathcal{L}^{\text{UCE-ENT}}(\boldsymbol{\theta}) = \mathbb{E}_{(\boldsymbol{X}, \mathbf{y}) \sim \mathcal{D}^{train}} \left[ \mathcal{L}^{\text{UCE}}(\boldsymbol{\theta}, \boldsymbol{X}, \mathbf{y}) - \beta \mathcal{L}^{\text{ENT}}(\boldsymbol{\theta}, \boldsymbol{X}) \right] \tag{5}$$

The first component, called UCE loss, aims to minimize the expected cross-entropy loss between the predicted class probabilities and the target categorical distribution. Here, the predicted categorical distribution is sampled from the anticipated Dirichlet distribution, and the target distribution follows a one-hot encoding format. This optimization strengthens the evidential support for the true class

while reducing support for other classes. Here, $\psi(\cdot)$ is the digamma function.

$$\mathcal{L}^{\text{UCE}}(\boldsymbol{\theta}, \boldsymbol{X}, \mathbf{y}) = \mathbb{E}_{\mathbf{p} \sim \text{Dir}(\boldsymbol{\alpha}(\boldsymbol{\theta}, \boldsymbol{X}))} [\mathbb{H}(\mathbf{p}, \boldsymbol{y})] = \sum_{c=1}^{C} y_c (\psi(\alpha_0) - \psi(\alpha_c))$$

The second component, termed the Entropy Regularizer (ER), involves the entropy of the predicted Dirichlet distribution. This can be viewed as the Kullback-Leibler (KL) divergence between the predicted Dirichlet distribution and a uniform Dirichlet prior, promoting a smooth Dirichlet distribution. The closed-form of the Bayesian loss is provided in Appendix A.1.

$$\mathcal{L}^{\text{ENT}}(\boldsymbol{\theta}, \boldsymbol{X}) = \mathbb{H}(\text{Dir}(\boldsymbol{\alpha}(\boldsymbol{\theta}, \boldsymbol{X}))) = \text{KL}(\text{Dir}(\mathbf{p}|\boldsymbol{\alpha}(\boldsymbol{\theta}, \boldsymbol{X})) \parallel \text{Dir}(\mathbf{p}|\mathbf{1}))$$

## 3 METHODOLOGY

In this section, we begin by discussing the limitations of the commonly used UCE loss for ENN models in Section 3.1. Next, we formally introduce our proposed UFCE loss in Section 3.2, highlighting how it partially addresses the limitations of UCE. Finally, we present our proposed uncertainty quantification framework in Section 3.3.

### 3.1 LIMITAION OF UCE

In an uncertainty-aware classification task, there are three levels of ground-truth and prediction pairs:

1. A one-hot encoded ground truth class label $\hat{\mathbf{y}}$, compared to the predicted label $\mathbf{y}$.

2. A ground-truth categorical distribution $\text{Cat}(\hat{\mathbf{p}})$ aligned with the predicted expected categorical distribution $\text{Cat}(\mathbf{p})$ from an ENN.

3. A ground-truth distribution over the categorical distribution $\text{Dir}(\hat{\boldsymbol{\alpha}})$, compared to the predicted Dirichlet distribution $\text{Dir}(\boldsymbol{\alpha})$, parameterized by the ENN.

Aleatoric uncertainty (uncertainty of the class prediction) can be calculated with the negative maximum class probability or the entropy of the categorical distribution. Ideally, aleatoric uncertainty should reflect the model's confidence in a perfectly calibrated network. However, over-parameterized deep neural networks, trained with the conventional cross-entropy objective, often exhibit overconfidence, leading to significant calibration issues (Guo et al., 2017). The overconfidence issue is frequently correlated with overfitting the negative log-likelihood (NLL), since even with a classification error of zero (indicative of perfect calibration), the NLL can remain positive. The optimization algorithm may continue to reduce this value by increasing the probability of the predicted class. In this section, we demonstrate that the UCE loss suffers from a similar issue. For example, even with perfect evidence volume prediction, the UCE loss remains positive and increasing the evidence for the predicted class further decreases the UCE loss. To address this, we propose the UFCE loss, which aims to improve the calibration performance and the quality of uncertainty estimation.

Epistemic uncertainty (uncertainty on the categorical distribution) can be calculated with the predicted total evidence $\alpha_0$. In the commonly used UCE loss (Sensoy et al., 2018; Charpentier et al., 2020), minimization continues when total evidence increases. Bengs et al. (2022) highlighted that learners employing UCE loss with first-level ground truth (class label) tend to peak the third-level distribution (Dirichlet distribution). This creates a false impression of complete certainty rather than accurately reflecting uncertainty. To address this, we propose epistemic uncertainty scaling and regularized evidential learning to improve epistemic uncertainty prediction.

Overall, aleatoric and epistemic uncertainty can be estimated based on the Dirichlet parameters $\boldsymbol{\alpha}$:

$$u^{alea} = -\max_c \bar{p}_c, \ u^{epis} = C/\alpha_0, \tag{6}$$

**Proposition 1.** *Given a predicted distribution* $\mathbf{p} \sim \text{Dir}(\boldsymbol{\alpha})$*, where* $\boldsymbol{\alpha} = (\alpha_1, \alpha_2, \ldots, \alpha_C)$ *and* $C$ *is the number of categories, and a target distribution* $\mathbf{q} \sim \text{Dir}(\hat{\boldsymbol{\alpha}})$*, assuming a one-hot style target distribution such that* $\hat{\alpha}_i = 1$ *for all* $i \neq c^*$ *where* $c^*$ *is the ground truth label and* $\hat{\alpha}_{c^*} = 2$*, we have,*

$$\mathcal{L}^{UCE} = KL(\text{Dir}(\boldsymbol{\alpha}) \parallel \text{Dir}(\hat{\boldsymbol{\alpha}})) + \mathbb{H}(\text{Dir}(\boldsymbol{\alpha})) - \log(B(\hat{\boldsymbol{\alpha}})) \tag{7}$$

where $\log(B(\hat{\boldsymbol{\alpha}}))$ is a constant with $B(\cdot)$ is Beta function.

Proposition 1 shows that optimizing UCE loss minimizes the sum of the KL divergence between the predicted and ground truth distributions and the entropy of the predicted distribution. Consequently, minimizing UCE loss forces the predicted Dirichlet distribution to align the one-hot target evidence distribution while simultaneously making it more peaked. Since epistemic uncertainty is measured by the spread of the Dirichlet distribution, this results in an overconfident model prone to overfitting.

## 3.2 UFCE - Implicit weight regularization

Motivated by the better calibration capability of Focal loss (Lin et al., 2017) compared to cross-entropy, we propose the Uncertainty Focal Cross Entropy (UFCE) loss, which takes the expectation of focal loss rather than cross-entropy loss.

$$
\begin{aligned}
\mathcal{L}^{\text{UFCE}}(\boldsymbol{\theta}, \boldsymbol{X}, \mathbf{y}) &= \mathbb{E}_{(\boldsymbol{X},\mathbf{y})\sim\mathcal{D}^{train}} \mathbb{E}_{\mathbf{p}\sim\text{Dir}(\boldsymbol{\alpha}(\boldsymbol{\theta},\boldsymbol{X}))}\left[-\sum_{c=1}^{C} y_c(1-p_c)^\gamma \log p_c\right] \\
&= \mathbb{E}_{(\boldsymbol{X},\mathbf{y})\sim\mathcal{D}^{train}}\left(\frac{B(\alpha_0,\gamma)}{B(\alpha_0-\alpha_{c^*},\gamma)}\left[\psi(\alpha_0+\gamma)-\psi(\alpha_{c^*})\right]\right)
\end{aligned}
\tag{8}
$$

where $\gamma$ is a hyperparameter, $c^*$ is the ground truth class index, and $B(\cdot)$ is the Beta function. When $\gamma = 0$, the UFCE loss is equivalent to the UCE loss.

**Can the UFCE loss improve calibration over UCE loss?** We investigate this question by analyzing the relationship between UFCE and UCE losses and their gradient behavior.

The general form of the UFCE loss has the following lower bound (see Appendix Proposition 4):
$$
\mathcal{L}^{\text{UFCE}} \geq \mathcal{L}^{\text{UCE}} - \gamma \cdot \mathbb{E}_{\mathbf{p}\sim\text{Dir}(\boldsymbol{\alpha})}\left[\mathbb{H}(\mathbf{p})\right]
\tag{9}
$$
This bound indicates that minimizing UFCE loss leads to both minimizing UCE loss and maximizing the expected entropy function ($\mathbb{H}(\mathbf{p})$) of the categorical distribution $\text{Cat}(\mathbf{p})$. The hyperparameter $\gamma$ balances these two terms. Maximizing the entropy term reduces the gap between $\alpha_{c^*}$ and the evidence of false classes $\alpha_c, \forall c \neq c^*$, implying that UFCE loss models are generally less confident in their evidence predictions than UCE loss models, particularly for correctly classified samples.

**Proposition 2.** *Comparing $\mathcal{L}^{UFCE}$ and $\mathcal{L}^{UCE}$ with numerical analysis on the gradient of the parameters $\mathbf{w}_{c^*}$ in the last linear layer, we have,*
$$
\left\|\frac{\partial\mathcal{L}^{UFCE}}{\partial\mathbf{w}_{c^*}}\right\| - \left\|\frac{\partial\mathcal{L}^{UCE}}{\partial\mathbf{w}_{c^*}}\right\|
\begin{cases}
\geq 0 & \text{if } \bar{p}_{c^*} \leq \tau_1(\alpha_{c^*},\gamma) \\
< 0 & \text{if } \bar{p}_{c^*} > \tau_2(\alpha_{c^*},\gamma)
\end{cases},
$$
*where $\tau_1(\alpha_{c^*},\gamma)$ and $\tau_2(\alpha_{c^*},\gamma)$ are two thresholds within $(\frac{1}{\alpha_0}, 1 - \frac{1}{\alpha_0})$. respectively.*

Proposition 2 shows the relationship between the norms of the gradients of the last linear layer for UFCE and UCE loss under the same network architecture. It is clear that for every $\gamma$ and $\alpha_{c^*}$, there exists a threshold $\tau_1$ such that for all $\bar{p}_{c^*} \in (\frac{1}{\alpha_0}, \tau_1]$, $\left\|\frac{\partial\mathcal{L}^{\text{UFCE}}}{\partial\mathbf{w}_{c^*}}\right\| \geq \left\|\frac{\partial\mathcal{L}^{\text{UCE}}}{\partial\mathbf{w}_{c^*}}\right\|$. This implies that the evidence belonging to the ground truth class predicted by the UFCE mode will initially increase faster than that of the UCE model.

Moreover, there exists a $\tau_2$, and for all $\bar{p}_{c^*} \in [\tau_2, 1 - \frac{1}{\alpha_0})$, $\left\|\frac{\partial\mathcal{L}^{\text{UFCE}}}{\partial\mathbf{w}_{c^*}}\right\| \leq \left\|\frac{\partial\mathcal{L}^{\text{UCE}}}{\partial\mathbf{w}_{c^*}}\right\|$. It implies once $p_{c^*}$ surpasses the threshold $\tau$, UFCE will apply a regularizing effect to prevent the model from continuing to focus on examples it is already confident about, thus avoiding overfitting.

Figure 1 shows an example with $\gamma = 1$ and $\alpha_{c^*} = 5$, $\tau_1 = \tau_2 \approx 0.4$. This also implicitly acts as a weight regularizer by pushing the model to focus more on less confident scenarios, which is crucial for highly imbalanced data. Further analysis can be found in Figure 2 and Figure 3.

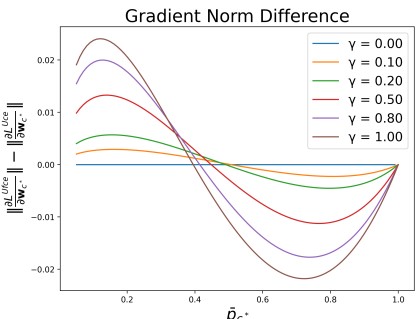

Figure 1: The y-axis represents the difference between the L1-norms of the UFCE and UCE gradients with $\alpha_{c^*} = 5$, while the x-axis corresponds to $\bar{p}_{c^*}$, the expected predicted probability of the ground truth class.

### 3.3 UNCERTAINTY QUANTIFICATION FRAMEWORK

**Epistemic uncertainty scaling (EUS)**: Owing to the imbalance between in-distribution (ID) and OOD pixels within the BEVSS datasets, we observe a tendency for the model to erroneously classify ID pixels that are proximal to OOD objects. This often leads to the model assigning low confidence to these ID pixels, resulting in high epistemic uncertainty. Consequently, this issue significantly increases the occurrence of false positives during the OOD detection task. To address this, we propose using epistemic uncertainty scaling to explicitly apply different weights for samples based on their predicted epistemic uncertainty in the previous step of optimization. The equation 8 is adjusted to:

$$\mathcal{L}^{\text{UFCE-EUS}} = \left(1 + \frac{C \cdot \xi}{\alpha_0}\right) \cdot \mathcal{L}^{\text{UFCE}} \tag{10}$$

where $\xi$ is a hyperparameter and $\alpha_0$ is the value predicted by the neural network in the previous optimization step. An example with low evidence in the training set will catch more attention during the training process and may lead to better performance on such "difficult" samples.

**Evidence-Regularized Learning (ER)**. We consider the OOD-exposure setting to further enhance the epistemic uncertainty estimation. Assume $\mathcal{D}_{in}^{train}$ is the ID training data and $\mathcal{D}_{out}^{train}$ is the auxiliary OOD training data. Note that we have true OOD ($\mathcal{D}_{out}^{test}$) for the evaluation where the auxiliary OOD and true OOD belong to different distributions. We propose training the model to minimize the KL divergence between predictions and a flat Dirichlet distribution for these auxiliary OOD samples.

$$\mathcal{L}^{\text{ER}} = \mathbb{E}_{(\boldsymbol{X}_{out}, \mathbf{y}) \sim \mathcal{D}_{out}^{train}} \text{KL}\left(\text{Dir}(\boldsymbol{\alpha}) \,\|\, \text{Dir}(\mathbf{1})\right) \tag{11}$$

For fair comparison, we also include a setting with OOD exposure for the energy-based model. With the same strategy as (Liu et al., 2020), we consider energy-bounded learning for OOD detection. Details can be found in Appendix A.2.3.

**Proposed framework**: We replace the last softmax activation function of the original BEVSS model with a ReLU function in order to produce concentrated parameters for pixel-wise Dirichlet distribution. The model is optimized by the following objective functions:

$$\mathcal{L}^{\text{UFCE-EUS-ER}} = \mathcal{L}^{\text{UFCE-EUS}} + \lambda \mathcal{L}^{\text{ER}} \tag{12}$$

Our proposed framework comprises three primary components: the Uncertainty Focal Cross-Entropy (UFCE) loss, epistemic uncertainty scaling (EUS), and Evidence-Regularized Learning (ER). We consider three hyperparameters. $\gamma$ in the UFCE, $\xi$ in the EUS and $\lambda$ for the ER term.

## 4 EXPERIMENTS

In this section, we introduce a benchmark that evaluates uncertainty-aware BEVSS. This benchmark includes three datasets, three BEVSS network architectures, five uncertainty quantification models, and two uncertainty estimation tasks. We address the following research questions:

**RQ1:** Are aleatoric and epistemic uncertainties estimated by these models reliable? Specifically, can the aleatoric uncertainty metrics accurately detect misclassified pixels, and can the epistemic uncertainty metrics effectively identify OOD samples?

**RQ2:** Does our proposed UFCE loss enhance model performance in terms of calibration and uncertainty estimation compared to the traditionally used UCE loss in ENNs?

**RQ3:** Does the proposed uncertainty quantification framework improve the prediction of epistemic uncertainty as evaluated by the OOD detection task?

Following these three research questions, we provide extensive experiments and further discussions.

### 4.1 BENCHMARK SETUPS

**Evaluation.** Following the task setting from (Philion & Fidler, 2020) and (Zhou & Krähenbühl, 2022), we conduct both vehicle segmentation and drivable area segmentation. We evaluate performance using four metrics: (1) *Pure segmentation* via Intersection-over-Union (IoU). (2) *Calibration* via

Expected Calibration Error (ECE). (3) *Aleatoric Uncertainty* via the misclassification detection to identify the misclassified pixels, measured with Area Under ROC Curve (AUROC) and Area Under PR Curve (AUPR). (4) *Epistemic Uncertainty* via the OOD detection to identify the OOD pixels, measured with AUROC and AUPR.

**Datasets.** We utilize the CARLA simulator-generated dataset [2], two real-world datasets (nuScenes and Lyft), and a corrupted version, nuScenes-C (Xie et al., 2024). In the default setting, "motorcycle" and "bicycle" serve as the true and pseudo-OOD classes for nuScenes and Lyft, respectively, while CARLA uses "deer" as the true OOD and "bears, horses, cows, elephants" as pseudo-OODs. To evaluate robustness to pseudo-OOD choices, an alternative configuration sets "traffic cones, pushable/pullable objects, motorcycles" as pseudo-OODs and "barriers" as the true OOD for nuScenes, with "kangaroo" as the true OOD for CARLA, which reduces similarity to pseudo-OODs compared to the default setting. Detailed dataset descriptions and statistics are in Appendix A.4.

**BEVSS Network Architecture.** We utilize LSS, CVT, and Simple-BEV as model backbones with publicly available implementations. LSS converts raw camera inputs into BEV representations by predicting depth distributions, constructing feature frustums, and rasterizing them onto a BEV grid, while CVT employs a transformer-based approach with cross-attention and camera-aware positional embeddings to align features into the BEV space. In contrast, Simple-BEV bypasses depth estimation entirely, projecting 3D coordinate volumes onto camera images to sample features, emphasizing efficiency and robustness to projection errors. Detailed descriptions can be found in Appendix A.2.1.

**Uncertainty Quantification Methods.** We consider widely used uncertainty quantification models in traditional deep learning. The "Entropy" and "Energy" models perform post-hoc processing on predicted logits, with "Energy" model being particularly popular for its adaptability and strong OOD detection performance. We also include the widely-used "Dropout" and "Ensemble" models. The "Ensemble" model consists of three separately trained models with different initialization seeds, while the "Dropout" model employs activated dropout layers during inference with 10 forward passes. Both models use either cross-entropy or focal loss. Lastly, the "ENN" model with UCE-ENT loss serves as a baseline to validate our framework. Notably, only the "Energy" and "ENN" models can be trained with or without pseudo-OOD information. Details are in Appendix A.2.3.

**Hyperparameters**. LSS and CVT are trained with a batch size of 32, while Simple-BEV uses a batch size of 16, all for 20 epochs on NVIDIA A6000 or A100 GPUs. Network-specific hyperparameters for LSS, CVT, and Simple-BEV are adopted from their original studies. We use the learning rate scheduler from CVT, setting the learning rate to 4e-3 for focal loss variants (as in CVT) and 1e-3 for cross-entropy variants (as in LSS), with the Adam optimizer and a weight decay of 1e-7, consistent with both LSS and CVT. We tune three regularization weights, $\lambda$, $\xi$, and $\gamma$, based on the AUPR metric for pseudo-OOD detection on the validation set. To manage computational complexity, we adopt a step-by-step tuning approach, fixing two parameters while adjusting the third, instead of performing a full grid search. Detailed information on hyperparameter tuning strategy, optimal values, and sensitivity analysis can be found in Appendix A.3.1.

## 4.2 RESULTS

In the main paper, we present results on nuScenes using LSS and CVT as backbones, covering predicted segmentation and aleatoric uncertainty (Table 1), epistemic uncertainty (Table 2), running time (Table 3), robustness analysis (Table 4), and ablation studies (Table 5).

In the appendix, we provide results on CARLA and Lyft using all backbones including LSS, CVT, and Simple-BEV (Tables 13–16). Additionally, we include hyperparameter analysis (Tables 7–9) and robustness experiments (Tables 17–23), which encompass detailed results on corrupted nuScenes-C, diverse town and weather conditions in CARLA, and pseudo-OOD selections. Qualitative comparisons are provided in Figures 4–6.

**Benchmark Observations (RQ1)**: Pure segmentation: (1) Across a comprehensive range of configurations (50 in total), focal-based losses demonstrate superior segmentation performance when compared to the standard cross-entropy loss in all but two instances, where the difference is marginal (within 0.7%). This pattern is particularly pronounced in models utilizing the CVT backbone, where focal-based losses yield more significant improvements. (2) The UFCE loss for the ENN consistently

---

[2] Due to the large size of the dataset, we will provide it upon request.

Table 1: Calibration and misclassification detection performance on the nuScence dataset for vehicle segmentation. Best and Runner-up results are highlighted in red and blue.

| Model | Loss | LSS | | | | | CVT | | | | |
|---|---|---|---|---|---|---|---|---|---|---|---|
| | | Pure Classification | | Misclassification | | | Pure Classification | | Misclassification | | |
| | | IoU ↑ | ECE↓ | AUROC ↑ | AUPR ↑ | FPR95 ↓ | IoU ↑ | ECE↓ | AUROC ↑ | AUPR ↑ | FPR95 ↓ |
| | | | | | | Without pseudo OOD | | | | | |
| Entropy | CE | 0.332 | 0.00887 | 0.909 | 0.315 | 0.234 | 0.277 | 0.00374 | 0.949 | 0.325 | 0.213 |
| | Focal | 0.347 | 0.00301 | 0.941 | 0.332 | 0.197 | 0.325 | 0.00341 | 0.949 | 0.321 | 0.206 |
| Energy | CE | 0.332 | 0.00887 | 0.909 | 0.315 | 0.234 | 0.277 | 0.00374 | 0.949 | 0.325 | 0.213 |
| | Focal | 0.347 | 0.00301 | 0.941 | 0.332 | 0.197 | 0.325 | 0.00341 | 0.949 | 0.321 | 0.206 |
| Ensemble | CE | 0.355 | 0.00569 | 0.933 | 0.317 | 0.218 | 0.301 | 0.00276 | 0.951 | 0.315 | 0.216 |
| | Focal | 0.370 | 0.00233 | 0.946 | 0.315 | 0.203 | 0.344 | 0.00243 | 0.953 | 0.324 | 0.195 |
| Dropout | CE | 0.332 | 0.00819 | 0.905 | 0.315 | 0.235 | 0.279 | 0.00373 | 0.946 | 0.332 | 0.213 |
| | Focal | 0.347 | 0.00261 | 0.936 | 0.325 | 0.208 | 0.325 | 0.00363 | 0.948 | 0.327 | 0.202 |
| ENN | UCE | 0.341 | 0.00429 | 0.819 | 0.273 | 0.335 | 0.291 | 0.00371 | 0.900 | 0.305 | 0.224 |
| | UFCE | 0.343 | 0.00332 | 0.873 | 0.310 | 0.225 | 0.319 | 0.0019 | 0.918 | 0.319 | 0.208 |
| | | | | | | With pseudo OOD | | | | | |
| Energy | CE | 0.348 | 0.00721 | 0.949 | 0.331 | 0.200 | 0.296 | 0.00238 | 0.951 | 0.316 | 0.219 |
| | Focal | 0.346 | 0.00466 | 0.951 | 0.331 | 0.192 | 0.333 | 0.00186 | 0.955 | 0.321 | 0.196 |
| ENN | UCE | 0.343 | 0.00342 | 0.838 | 0.288 | 0.296 | 0.282 | 0.00336 | 0.919 | 0.313 | 0.227 |
| | Ours | 0.356 | 0.00193 | 0.911 | 0.317 | 0.190 | 0.319 | 0.00019 | 0.934 | 0.321 | 0.196 |

**Observations:** 1. Involving pseudo-OOD in the training phase does not impact pure segmentation, calibration and misclassification detection. 2. Without pseudo-OOD, the proposed UFCE loss for the ENN model consistently outperforms the commonly used UCE loss across all metrics, showing average improvements of 0.0014 in calibration ECE, 3.6% in misclassification AUROC, 2.5% in AUPR and 6.3% on FPR95 across two backbones. 3. No single model consistently performs best across all metrics, but Focal consistently performs better than CE.

Table 2: OOD detection performance for vehicle segmentation . Best and Runner-up results are highlighted in red and blue.

| Model | Loss | nuScenes | | | | | | CARLA | | | | | |
|---|---|---|---|---|---|---|---|---|---|---|---|---|---|
| | | LSS | | | CVT | | | LSS | | | CVT | | |
| | | AUROC ↑ | AUPR ↑ | FPR95 ↓ | AUROC ↑ | AUPR ↑ | FPR95 ↓ | AUROC ↑ | AUPR ↑ | FPR95 ↓ | AUROC ↑ | AUPR ↑ | FPR95↓ |
| | | | | | | | Without pseudo OOD | | | | | | |
| Entropy | CE | 0.584 | 0.00052 | 0.799 | 0.728 | 0.00057 | 0.824 | 0.693 | 0.00236 | 0.782 | 0.813 | 0.00307 | 0.704 |
| | Focal | 0.647 | 0.00056 | 0.764 | 0.690 | 0.00053 | 0.828 | 0.693 | 0.00236 | 0.782 | 0.794 | 0.00271 | 0.736 |
| Energy | CE | 0.602 | 0.00049 | 0.794 | 0.720 | 0.00060 | 0.801 | 0.683 | 0.00217 | 0.762 | 0.813 | 0.00304 | 0.708 |
| | Focal | 0.564 | 0.00050 | 0.781 | 0.639 | 0.00053 | 0.823 | 0.683 | 0.00217 | 0.762 | 0.788 | 0.00265 | 0.737 |
| Ensemble | CE | 0.385 | 0.00016 | 0.979 | 0.478 | 0.00021 | 0.960 | 0.488 | 0.00066 | 0.965 | 0.505 | 0.00068 | 0.963 |
| | Focal | 0.537 | 0.00025 | 0.941 | 0.503 | 0.00021 | 0.964 | 0.455 | 0.00066 | 0.953 | 0.491 | 0.00067 | 0.962 |
| Dropout | CE | 0.411 | 0.00017 | 0.975 | 0.384 | 0.00017 | 0.957 | 0.441 | 0.00062 | 0.966 | 0.36 | 0.00052 | 0.971 |
| | Focal | 0.348 | 0.00015 | 0.994 | 0.402 | 0.00019 | 0.931 | 0.390 | 0.00056 | 0.964 | 0.317 | 0.00049 | 0.974 |
| ENN | UCE | 0.717 | 0.00075 | 0.791 | 0.661 | 0.00049 | 0.857 | 0.620 | 0.00172 | 0.795 | 0.685 | 0.00235 | 0.814 |
| | UFCE | 0.518 | 0.00034 | 0.892 | 0.683 | 0.00066 | 0.816 | 0.535 | 0.00163 | 0.835 | 0.748 | 0.00237 | 0.775 |
| | | | | | | | With pseudo OOD | | | | | | |
| Energy | CE | 0.774 | 0.04740 | 0.408 | 0.839 | 0.02060 | 0.356 | 0.897 | 0.07450 | 0.259 | 0.929 | 0.05140 | 0.159 |
| | Focal | 0.821 | 0.04440 | 0.378 | 0.860 | 0.02370 | 0.319 | 0.908 | 0.07800 | 0.183 | 0.948 | 0.07880 | 0.137 |
| ENN | UCE | 0.889 | 0.315 | 0.315 | 0.921 | 0.212 | 0.306 | 0.889 | 0.147 | 0.272 | 0.970 | 0.111 | 0.161 |
| | Ours | 0.929 | 0.335 | 0.219 | 0.928 | 0.269 | 0.244 | 0.960 | 0.204 | 0.180 | 0.979 | 0.237 | 0.125 |

**Observations:** 1. Without pseudo-OOD, no uncertainty quantification models can predict satisfying epistemic uncertainty, as shown by extremely low OOD detection PR, implying epistemic uncertainty estimation is challenging for BEVSS. 2. Compared to all baselines, our proposed model (last line in the table) has the best OOD detection performance on all 12 metrics, with an average improvement of 2.7% in AUROC, 6.5% in AUPR and 4.3% in FPR95 over the runner-up model.

outperforms the UCE loss across all 14 configurations. Calibration: In the majority of scenarios, models with focal-based loss exhibit lower ECE scores, indicating better calibration than cross-entropy based models. Similarly, UFCE consistently outperforms UCE. No single model consistently performs best in terms of calibration. Misclassification detection: (1) Despite most models achieving AUROC scores over 90%, the AUPR values remain below 40% across two datasets, indicating significant room for improvement in aleatoric uncertainty estimation. (2) Models using a focal-based loss perform better in misclassification detection across most settings compared to those using a CE-based loss. Our model performs consistently better than the UCE-based ENN.

Table 3: Running time comparison between uncertainty quantification models

| Model | Task | Energy | Ensemble | Dropout | ENN-UCE | ENN-UFCE | Ours |
|---|---|---|---|---|---|---|---|
| LSS | Training | 10 hours | 30 hours | 10 hours | 10 hours | 10 hours | 10 hours |
| | Inference | 54 ms | 147 ms | 457 ms | 55 ms | 51 ms | 56 ms |
| CVT | Training | 8 hours | 24 hours | 8 hours | 8 hours | 8 hours | 8 hours |
| | Inference | 15 ms | 32 ms | 123 ms | 14 ms | 16 ms | 15 ms |

OOD detection: (1) Without pseudo-OOD exposure, all models perform poorly in OOD detection, as evidenced by low AUPR values. (2) Only the energy-based and evidential-based models can utilize pseudo-OOD data. With pseudo-OOD exposure, both models show significant improvements in AUROC and AUPR, indicating that utilizing pseudo-OODs may be a promising direction. Our proposed framework achieves the highest OOD detection performance across all eight evaluated settings. (3) OOD detection in BEVSS is a challenging task, as evidenced by the low AUPR.

Complexity analysis: Table 3 presents the estimated duration for training on a pair of A100 GPUs and inference on a single A6000 GPU. Ensemble models require significantly more time for training, whereas the Dropout model incurs a longer duration during inference. Conversely, the ENN demonstrates reduced time complexity for both training and inference processes. Our proposed model has similar training and inference time cost with the energy model.

**Analysis of UFCE Loss (RQ2)**: We discuss the effect of UFCE loss from three views. Calibration: (1) Compared to UCE loss without pseudo-OOD exposure, models using UFCE loss achieve slightly higher IoU and lower ECE scores across most configurations for vehicle detection and driveable region detection tasks on three datasets and three network architectures. Additionally, under pseudo-OOD supervision, UFCE-based models consistently show better IoU and ECE scores across all experimental setups, highlighting UFCE's superior segmentation accuracy and model calibration.

(2) Compared to all models with the same configuration, the ENN using UFCE loss achieves results comparable to those designed solely for segmentation. Notably, the ENN with UFCE loss achieves the lowest ECE score on 4 out of 6 configurations and another second lowest on 1 configuration, demonstrating its effectiveness in calibration. In addition, with pseudo-OOD exposure, the ENN shows improved segmentation performance. This enhancement is due to the model's capability to predict accurate Dirichlet distributions, especially for pixels near OOD instances.

Table 4: Robustness study on the selection of pseudo-OOD with vehicle segmentation task on nuScenes. Best and Runner-up results are highlighted in red and blue.

| Model | LSS | | | CVT | | |
|---|---|---|---|---|---|---|
| | OOD Detection | | | OOD Detection | | |
| | AUROC ↑ | AUPR ↑ | FPR95 ↓ | AUROC ↑ | AUPR ↑ | FPR95 ↓ |
| UFCE-EUS-ER | 0.895 | 0.215 | 0.274 | 0.940 | 0.153 | 0.272 |
| UCE-EUS-ER | 0.914 | 0.208 | 0.291 | 0.908 | 0.117 | 0.346 |
| UCE-ER | 0.862 | 0.192 | 0.302 | 0.887 | 0.0934 | 0.354 |
| UFCE | 0.495 | 0.000609 | 0.919 | 0.727 | 0.00118 | 0.861 |

We evaluate the alternative pseudo-OOD setting, using traffic cones, pushable/pullable objects, and motorcycles as pseudo-OOD, with barriers as the true OOD. Our model outperforms the runner-up in 5 out of 6 metrics, achieving an average AUPR improvement of 18%, with both UFCE and EUS significantly contributing to these gains.

Misclassification detection: UFCE loss surpasses UCE loss with consistently higher AUROC and AUPR scores, generally ranking in the mid-to-upper range among models.

OOD detection: Compared to UCE loss with pseudo-OOD supervision, our proposed framework with UFCE loss demonstrates significantly better performance, with up to a 12% increase in AUPR and a 4% boost in AUROC. These results highlight the significant potential of the UFCE approach to improve the reliability of epistemic uncertainty estimation.

Table 5: Ablation study for vehicle segmentation on nuScenes . Best and Runner-up results are highlighted in red and blue.

| Model | Pure Classification | | Misclassification | | | OOD Detection | | |
|---|---|---|---|---|---|---|---|---|
| | IoU ↑ | ECE ↓ | AUROC ↑ | AUPR ↑ | FPR95 ↓ | AUROC ↑ | AUPR ↑ | FPR95 ↓ |
| LSS | | | | | | | | |
| UFCE-EUS-ER | 0.356 | 0.00193 | 0.911 | 0.317 | 0.190 | 0.929 | 0.335 | 0.219 |
| UCE-EUS-ER | 0.339 | 0.00363 | 0.842 | 0.290 | 0.290 | 0.842 | 0.340 | 0.329 |
| UCE-ER | 0.342 | 0.00342 | 0.838 | 0.289 | 0.296 | 0.889 | 0.315 | 0.315 |
| UCE | 0.341 | 0.00429 | 0.819 | 0.273 | 0.335 | 0.717 | 0.00075 | 0.791 |
| CVT | | | | | | | | |
| UFCE-EUS-ER | 0.319 | 0.00019 | 0.934 | 0.321 | 0.196 | 0.928 | 0.269 | 0.244 |
| UCE-EUS-ER | 0.281 | 0.00360 | 0.920 | 0.314 | 0.217 | 0.931 | 0.210 | 0.326 |
| UCE-ER | 0.282 | 0.00336 | 0.919 | 0.313 | 0.227 | 0.921 | 0.212 | 0.306 |
| UCE | 0.291 | 0.00371 | 0.900 | 0.305 | 0.224 | 0.661 | 0.00049 | 0.857 |

The base model is "UCE" and we progressively add the proposed components. First, we introduce ER to obtain "UCE-ER", followed by adding EUS to create "UCE-EUS-ER". Finally, we replace UCE with UFCE, resulting in the model "UFCE-EUS-ER". **Observation:** Adding "ER" largely improve the OOD detection performance (average 26%), "EUS" further improve he misclassification detection and OOD detection slightly. The "UFCE" improves the calibration and misclassification detection with a significant gap ( ECE:0.002555, AUROC:4.15%, AUPR:1.7%, FPR95:6.05%)

**Ablation Study (RQ3)**: There are three primary components: UFCE, EUS, ER. We conduct ablation studies using the nuScenes dataset with the LSS and CVT backbones to assess the impact of each component on system performance without compromising generality. The results are summarized in Table 5. Starting with the standard ENN model using the UCE loss as the baseline, we progressively add components to assess their contributions. First, we introduce the ER term, which incorporates pseudo-OOD data during training. This addition leads to a significant improvement in OOD detection performance, with up to a 31% improvement in AUPR. Next, we add the EUS regularization, which further enhances both misclassification detection and OOD detection performance. Finally, we replace the UCE loss with our proposed UFCE loss, achieving the best overall results. This change results in up to a 6% improvement in AUPR, particularly benefiting calibration and misclassification detection without sacrificing segmentation accuracy.

**Discussion on the pseudo-OOD:** Intuitively, greater similarity between true and pseudo-OOD pairs enhances OOD detection performance, while overfitting to pseudo-OODs raises concerns about the model's generalization ability. To further investigate, we conducted experiments using dissimilar pseudo-OOD and true OOD pairs compared to the default setting (Table 2). The results for nuScenes are shown in Table 4, while the results for CARLA are provided in Table 17 in Appendix A.5.5. The findings confirm our intuition: higher similarity between true and pseudo-OOD pairs leads to better OOD detection performance. Notably, our proposed model consistently outperformed the best baseline methods across all eight scenarios, achieving improvements of up to 12.6% in AUPR. These results underscore the robustness of our approach, even in settings with less similar OOD pairs.

## 5 CONCLUSION

This paper presents a comprehensive evaluation of various uncertainty quantification methods for BEVSS. Our findings reveal that current methods do not achieve satisfactory results in uncertainty quantification, particularly in OOD detection, highlighting the need for advancements in this domain. Inspired by the robust calibration properties of Focal Loss, we introduce the UFCE loss, which significantly enhances model calibration. Our proposed uncertainty quantification framework, based on evidential deep learning, consistently outperforms baseline models in predicting epistemic uncertainty, as well as in aleatoric uncertainty and calibration, across a wide range of scenarios.

ACKNOWLEDGMENTS

This work is partially supported by the National Science Foundation (NSF) under Grant No. 2414705, 2220574, 2107449, and 1954376.

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

# A APPENDIX

## A.1 PROOFS

Assume the target distribution $q \sim \text{Dir}(\hat{\alpha})$ and the predicted distribution $p \sim \text{Dir}(\alpha)$, where $\alpha = (\alpha_1, \alpha_2, \ldots, \alpha_C)$ and $\hat{\alpha} = (\hat{\alpha}_1, \hat{\alpha}_2, \ldots, \hat{\alpha}_C)$ with $C$ is the number of categories. We first provide some preliminary results related to Dirichlet distribution.

The probability density function for a Dirichlet distribution for a vector $p$ is given by:

$$D(p|\alpha) = \frac{1}{B(\alpha)} \prod_{i=1}^{C} p_i^{\alpha_i - 1} \tag{13}$$

where $B(\alpha)$ is the beta function for the vector $\alpha$.

$$B(\alpha) = \frac{\prod_{i=1}^{C} \Gamma(\alpha_i)}{\Gamma(\alpha_0)} \tag{14}$$

We use $\alpha_0 = \sum_{i=1}^{C} \alpha_i$ and $\hat{\alpha}_0 = \sum_{i=1}^{C} \hat{\alpha}_i$ for simplification.

Then, the Kullback-Leibler divergence of $p$ from $q$ is given by

$$\text{KL}[p \parallel q] = \log B(\hat{\alpha}) - \log B(\alpha) + \sum_{i=1}^{C} (\alpha_i - \hat{\alpha}_i) \left[ \psi(\alpha_i) - \psi(\alpha_0) \right] \tag{15}$$

The entropy of $p$ is given by:

$$H(p) = \log B(\alpha) + (\alpha_0 - C) \psi(\alpha_0) - \sum_{i=1}^{C} (\alpha_i - 1) \psi(\alpha_i) \tag{16}$$

**Proposition 1.** *Given a predicted distribution* $\mathbf{p} \sim \text{Dir}(\alpha)$*, where* $\alpha = (\alpha_1, \alpha_2, \ldots, \alpha_C)$ *and* $C$ *is the number of categories, and a target distribution* $q \sim \text{Dir}(\hat{\alpha})$*, assuming a one-hot style target distribution such that* $\hat{\alpha}_i = 1$ *for all* $i \neq c^*$ *where* $c^*$ *is the ground truth label and* $\hat{\alpha}_{c^*} = 2$*, we have,*

$$\mathcal{L}^{UCE} = KL(\text{Dir}(\alpha) \parallel \text{Dir}(\hat{\alpha})) + \mathbb{H}(\text{Dir}(\alpha)) - \log(B(\hat{\alpha})) \tag{7}$$

*Proof.* Combine Equation 15 and 16, then we have

$$\text{KL}(\text{Dir}(\alpha) \parallel \text{Dir}(\hat{\alpha})) + \text{H}(\text{Dir}(\alpha)) = \log B(\hat{\alpha}) + \sum_{i=1}^{C} (\hat{\alpha}_i - 1) \left[ \psi(\alpha_0) - \psi(\alpha_i) \right] \tag{17}$$

Given that $\hat{\alpha}$ is a perfect separable target distribution, i.e. , $\hat{\alpha}_i = 1$ for all $i \neq c^*$, then we have

$$\text{KL}(\text{Dir}(\alpha) \parallel \text{Dir}(\hat{\alpha})) + \text{H}(\text{Dir}(\alpha)) = (\hat{\alpha}_{c^*} - 1) \left[ \psi(\alpha_0) - \psi(\alpha_{c^*}) \right] + \log B(\hat{\alpha}) \tag{18}$$

The uncertainty cross entropy loss is the expectation of cross entropy loss, i.e.

$$\mathcal{L}^{\text{UCE}} = \mathbb{E}_{\mathbf{p} \sim \text{Dir}(\alpha)} \left[ -\sum_{i=1}^{C} y_c \log p_c \right] = \psi(\alpha_0) - \psi(\alpha_{c^*}) \tag{19}$$

Then we have:

$$\mathcal{L}^{\text{UCE}} = \frac{\text{KL}(\text{Dir}(\alpha) \parallel \text{Dir}(\hat{\alpha})) + \text{H}(\text{Dir}(\alpha)) - \log B(\hat{\alpha})}{\hat{\alpha}_{c^*} - 1} \tag{20}$$

$$= \frac{\text{KL}(\text{Dir}(\alpha) \parallel \text{Dir}(\hat{\alpha}))}{\hat{\alpha}_{c^*} - 1} + \frac{\text{H}(\text{Dir}(\alpha))}{\hat{\alpha}_{c^*} - 1} + \frac{\log B(\hat{\alpha})}{\hat{\alpha}_{c^*} - 1} \tag{21}$$

Considering the loss for a single point, the term $e_c = \alpha_c - 1$ represents the number of events that occurred given the uniform prior, with 1 being the maximum value 0 being the minimum value. Therefore, for each sample observed, we only have $\hat{\alpha}_{c^*} = e_{c^*} + 1 = 2$ while $\hat{\alpha}_c = e_c + 1 = 1, \forall c \neq c^*$, which corresponds to the only event occurred at ground-truth class $c^*$. Consequently, the loss function is given by:

$$\mathcal{L}^{\text{UCE}} = \text{KL}(\text{Dir}(\alpha) \parallel \text{Dir}(\hat{\alpha})) + \text{H}(\text{Dir}(\alpha)) - \log(B(\hat{\alpha})) \tag{22}$$

$\square$

Based on Proposition 1, when we minimize the UCE loss, we are minimizing the KL divergence between a predicted Dirichlet distribution with a target distribution with 'one-hot' evidence, as well as minimize the entropy of the predicted Dirichlet distribution. This will lead the Dirichlet distribution to peak at some point and can not spread to denote the true distribution, leading to overfitting.

**Proposition 3.** *Given a random variable $\boldsymbol{p}$ following a Dirichlet distribution $\mathtt{Dir}(\boldsymbol{\alpha})$, then the expectation of Focal loss has the following analytical form:*

$$\mathcal{L}^{UFCE} = \mathbb{E}_{\boldsymbol{p} \sim Dir(\boldsymbol{\alpha})} \left[ -(1 - p_{c^*})^\gamma \log p_{c^*} \right] \tag{23}$$

$$= \frac{\Gamma(\alpha_0 - \alpha_{c^*} + \gamma)\Gamma(\alpha_0)}{\Gamma(\alpha_0 + \gamma)\Gamma(\alpha_0 - \alpha_{c^*})} \left[ \psi(\alpha_0 + \gamma) - \psi(\alpha_{c^*}) \right] \tag{24}$$

*where $B(\cdot)$ denote Beta function and $\boldsymbol{\alpha} \in [1, +\infty)^C$ is the predicted strength parameter and $C$ is the number of classes.*

*Proof.* Let $B(\cdot)$ denote the Beta function and $\mathtt{Beta}(\cdot)$ denote the Beta distribution. Define the Gamma function as $\Gamma(\cdot)$, the Digamma function as $\psi(\cdot)$, and the Trigamma function as $\psi_1(\cdot)$. Let $\mathbf{y}$ be the ground-truth target one-hot vector and $\mathbf{p}$ be the predicted probability distribution over the simplex $\Delta$. For simplicity, denote $c^*$ as the index of the ground-truth class and define $\alpha_0 = \sum_c \alpha_c$.

The analytical form of Uncertainty Focal Loss can be derived as

$$
\begin{aligned}
\mathcal{L}^{\text{UFCE}} &= \mathbb{E}_{\mathbf{p} \sim \mathtt{Dir}(\boldsymbol{\alpha})} \left[ -\sum_{c=1}^{C} y_c(1 - p_c)^\gamma \log p_c \right] \\
&= -\int (1 - p_{c^*})^\gamma \log(p_{c^*}) \mathtt{Dir}(\mathbf{p}|\boldsymbol{\alpha}) d\mathbf{p} \\
&= -\int (1 - p_{c^*})^\gamma \log(p_{c^*}) \mathtt{Beta}(p_{c^*}|\alpha_{c^*}, \alpha_0 - \alpha_{c^*}) dp_{c^*} \\
&= -\frac{1}{B(\alpha_{c^*}, \alpha_0 - \alpha_{c^*})} \int (1 - p_{c^*})^\gamma \log(p_{c^*}) p_{c^*}^{\alpha_{c^*}-1}(1 - p_{c^*})^{\alpha_0 - \alpha_{c^*}-1} dp_{c^*} \\
&= -\frac{1}{B(\alpha_{c^*}, \alpha_0 - \alpha_{c^*})} \int \left( \frac{d}{d\alpha_{c^*}} p_{c^*}^{\alpha_{c^*}-1} \right)(1 - p_{c^*})^\gamma (1 - p_{c^*})^{\alpha_0 - \alpha_{c^*}-1} dp_{c^*} \\
&= -\frac{1}{B(\alpha_{c^*}, \alpha_0 - \alpha_{c^*})} \frac{d}{d\alpha_{c^*}} \int \left[ (1 - p_{c^*})^\gamma p_{c^*}^{\alpha_{c^*}-1}(1 - p_{c^*})^{\alpha_0 - \alpha_{c^*}-1} \right] dp_{c^*} \\
&= -\frac{1}{B(\alpha_{c^*}, \alpha_0 - \alpha_{c^*})} \frac{d}{d\alpha_{c^*}} \int \left[ p_{c^*}^{\alpha_{c^*}-1}(1 - p_{c^*})^{\alpha_0 - \alpha_{c^*}+\gamma-1} \right] dp_{c^*} \\
&= -\frac{1}{B(\alpha_{c^*}, \alpha_0 - \alpha_{c^*})} \frac{d}{d\alpha_{c^*}} B(\alpha_{c^*}, \alpha_0 - \alpha_{c^*} + \gamma) \\
&= -\frac{1}{B(\alpha_{c^*}, \alpha_0 - \alpha_{c^*})} \frac{d}{d\alpha_{c^*}} \frac{\Gamma(\alpha_{c^*})\Gamma(\alpha_0 - \alpha_{c^*} + \gamma)}{\Gamma(\alpha_0 + \gamma)} \\
&= -\frac{1}{B(\alpha_{c^*}, \alpha_0 - \alpha_{c^*})} B(\alpha_{c^*}, \alpha_0 - \alpha_{c^*} + \gamma) \left[ \psi(\alpha_{c^*}) - \psi(\alpha_0 + \gamma) \right] \\
&= \frac{\Gamma(\alpha_{c^*})\Gamma(\alpha_0 - \alpha_{c^*} + \gamma)/\Gamma(\alpha_0 + \gamma)}{\Gamma(\alpha_{c^*})\Gamma(\alpha_0 - \alpha_{c^*})/\Gamma(\alpha_0)} \left[ \psi(\alpha_0 + \gamma) - \psi(\alpha_{c^*}) \right] \\
&= \frac{\Gamma(\alpha_0)\Gamma(\alpha_0 - \alpha_{c^*} + \gamma)}{\Gamma(\alpha_0 + \gamma)\Gamma(\alpha_0 - \alpha_{c^*})} \left[ \psi(\alpha_0 + \gamma) - \psi(\alpha_{c^*}) \right].
\end{aligned}
\tag{25}
$$

$\square$

**Proposition 4.** *Given that $\gamma \geq 1$, the UFCE loss has the lower bound involving UCE loss as:*

$$\mathcal{L}^{UFCE} \geq \mathcal{L}^{UCE} - \gamma \cdot \mathbb{E}_{\mathbf{p} \sim \mathtt{Dir}(\boldsymbol{\alpha})} \left[ H(\mathbf{p}) \right]. \tag{26}$$

*Proof.* For any $\gamma \geq 1$, by applying Bernoulli's inequality and Hölder's inequality, we have the following relation when having the same predicted $\boldsymbol{\alpha}$:

$$
\begin{aligned}
\mathcal{L}^{\text{UFCE}} &= \mathbb{E}_{\mathbf{p}\sim\text{Dir}(\boldsymbol{\alpha})}\left[ -\sum_{c=1}^{C}(1-p_c)^{\gamma}y_c\log(p_c) \right] \\
&\geq \mathbb{E}_{\mathbf{p}\sim\text{Dir}(\boldsymbol{\alpha})}\left[ -\sum_{c=1}^{C}(1-\gamma\cdot p_c)y_c\log(p_c) \right] \\
&= \mathbb{E}_{\mathbf{p}\sim\text{Dir}(\boldsymbol{\alpha})}\left[ -\sum_{c=1}^{C}y_c\log(p_c) + \sum_{c=1}^{C}\gamma y_c p_c\log(p_c) \right] \\
&= \mathbb{E}_{\mathbf{p}\sim\text{Dir}(\boldsymbol{\alpha})}\left[ -\sum_{c=1}^{C}y_c\log(p_c) - \left|\sum_{c=1}^{C}\gamma y_c p_c\log(p_c)\right| \right] \\
&= \mathbb{E}_{\mathbf{p}\sim\text{Dir}(\boldsymbol{\alpha})}\left[ -\sum_{c=1}^{C}y_c\log(p_c) - \gamma\left\|\mathbf{y}^{\top}(\mathbf{p}\circ\log(\mathbf{p}))\right\|_1 \right] \\
&\geq \mathbb{E}_{\mathbf{p}\sim\text{Dir}(\boldsymbol{\alpha})}\left[ -\sum_{c=1}^{C}y_c\log(p_c) - \gamma\|\mathbf{y}\|_{\infty}\|\mathbf{p}\circ\log(\mathbf{p})\|_1 \right] \qquad (27) \\
&= \mathbb{E}_{\mathbf{p}\sim\text{Dir}(\boldsymbol{\alpha})}\left[ -\sum_{c=1}^{C}y_c\log(p_c) - \gamma\left(\max_{j}y_j\right)\left|\sum_{c=1}^{C}p_c\log(p_c)\right| \right] \\
&= \mathbb{E}_{\mathbf{p}\sim\text{Dir}(\boldsymbol{\alpha})}\left[ -\sum_{c=1}^{C}y_c\log(p_c) - \gamma\left|\sum_{c=1}^{C}p_c\log(p_c)\right| \right] \\
&= \mathbb{E}_{\mathbf{p}\sim\text{Dir}(\boldsymbol{\alpha})}\left[ -\sum_{c=1}^{C}y_c\log(p_c) \right] - \gamma\cdot\mathbb{E}_{\mathbf{p}\sim\text{Dir}(\boldsymbol{\alpha})}\left[ -\sum_{c=1}^{C}p_c\log(p_c) \right] \\
&= \mathbb{E}_{\mathbf{p}\sim\text{Dir}(\boldsymbol{\alpha})}\left[ -\sum_{c=1}^{C}y_c\log(p_c) \right] - \gamma\cdot\mathbb{E}_{\mathbf{p}\sim\text{Dir}(\boldsymbol{\alpha})}\left[ H(\mathbf{p}) \right] \\
&= \mathcal{L}^{\text{UCE}} - \gamma\cdot\mathbb{E}_{\mathbf{p}\sim\text{Dir}(\boldsymbol{\alpha})}\left[ H(\mathbf{p}) \right],
\end{aligned}
$$

$\square$

where $\circ$ denotes element-wise product.

Proposition 4 shows that the lower bound of UFCE loss is equivalent to the UCE loss minus the entropy where $\gamma$ is a trade-off parameter.

**Proposition 5.** *For uncertainty focal loss $\mathcal{L}^{UFCE}$, the partial derivative with respect to the ground truth class $c^*$ has the form:*

$$
\frac{\partial}{\partial\alpha_{c^*}}\mathcal{L}^{UFCE} = \frac{\Gamma(\alpha_0)\Gamma(\alpha_0-\alpha_{c^*}+\gamma)}{\Gamma(\alpha_0+\gamma)\Gamma(\alpha_0-\alpha_{c^*})}\Bigg\{\Big[\psi(\alpha_0)-\psi(\alpha_0+\gamma)\Big]\cdot\Big[\psi(\alpha_0+\gamma)-\psi(\alpha_{c^*})\Big] \qquad (28)
$$

$$
+\Big[\psi_1(\alpha_0+\gamma)-\psi_1(\alpha_{c^*})\Big]\Bigg\}. \qquad (29)
$$

*Proof.* for UFCE loss, the gradient is

$$
\begin{aligned}
\frac{\partial}{\partial \alpha_{c^*}} \mathcal{L}^{\text{UFCE}} &= \frac{\partial}{\partial \alpha_{c^*}} \left[ \frac{\Gamma(\alpha_0)\Gamma(\alpha_0 - \alpha_{c^*} + \gamma)}{\Gamma(\alpha_0 + \gamma)\Gamma(\alpha_0 - \alpha_{c^*})} \cdot [\psi(\alpha_0 + \gamma) - \psi(\alpha_{c^*})] \right] \\
&= \left[ \frac{\partial}{\partial \alpha_{c^*}} \frac{B(\alpha_0, \gamma)}{B(\alpha_0 - \alpha_{c^*}, \gamma)} \right] \cdot [\psi(\alpha_0 + \gamma) - \psi(\alpha_{c^*})] \\
&\quad + \frac{B(\alpha_0, \gamma)}{B(\alpha_0 - \alpha_{c^*}, \gamma)} \cdot \frac{\partial}{\partial \alpha_{c^*}} \left[ \psi(\alpha_0 + \gamma) - \psi(\alpha_{c^*}) \right] \\
&= \left[ \frac{B(\alpha_0, \gamma)}{B(\alpha_0 - \alpha_{c^*}, \gamma)} \right] \cdot [\psi(\alpha_0) - \psi(\alpha_0 + \gamma)] \cdot [\psi(\alpha_0 + \gamma) - \psi(\alpha_{c^*})] \qquad (30) \\
&\quad + \frac{B(\alpha_0, \gamma)}{B(\alpha_0 - \alpha_{c^*}, \gamma)} \cdot \left[ \psi_1(\alpha_0 + \gamma) - \psi_1(\alpha_{c^*}) \right] \\
&= \frac{\Gamma(\alpha_0)\Gamma(\alpha_0 - \alpha_{c^*} + \gamma)}{\Gamma(\alpha_0 + \gamma)\Gamma(\alpha_0 - \alpha_{c^*})} \left\{ \left[ \psi(\alpha_0) - \psi(\alpha_0 + \gamma) \right] \cdot \left[ \psi(\alpha_0 + \gamma) - \psi(\alpha_{c^*}) \right] \right. \\
&\quad \left. + \left[ \psi_1(\alpha_0 + \gamma) - \psi_1(\alpha_{c^*}) \right] \right\}.
\end{aligned}
$$

$\square$

In the following proposition, we consider the influence of gradients on the prediction of $\alpha_{c^*}$. Let $\mathbf{w}_{c^*}$ denote the vector of weight parameters in the last linear layer that influences the prediction of the true-class evidence $c^*$. Let $\mathbf{s}$ denote the logits and $\mathbf{z}$ be the input to the last linear layer. The prediction of $\alpha_{c^*}$ has the following form:

$$
c^* = \sigma_{\text{ReLU}} \left( \mathbf{w}_{c^*}^T \begin{bmatrix} \mathbf{z} \\ 1 \end{bmatrix} \right),
$$

where the last weight in $\mathbf{w}_{c^*}$ is related to the intercept.

**Proposition 2.** *Comparing $\mathcal{L}^{UFCE}$ and $\mathcal{L}^{UCE}$ with numerical analysis on the gradient of the parameters $\mathbf{w}_{c^*}$ in the last linear layer, we have,*

$$
\left\| \frac{\partial \mathcal{L}^{UFCE}}{\partial \mathbf{w}_{c^*}} \right\| - \left\| \frac{\partial \mathcal{L}^{UCE}}{\partial \mathbf{w}_{c^*}} \right\| \begin{cases} \geq 0 & \textit{if } \bar{p}_{c^*} \leq \tau_1(\alpha_{c^*}, \gamma) \\ < 0 & \textit{if } \bar{p}_{c^*} > \tau_2(\alpha_{c^*}, \gamma) \end{cases},
$$

*where $\tau_1(\alpha_{c^*}, \gamma)$ and $\tau_2(\alpha_{c^*}, \gamma)$ are two thresholds within $(\frac{1}{\alpha_0}, 1 - \frac{1}{\alpha_0})$. respectively.*

*Proof.* Define $\mathbf{w}_{c^*}$ as the model parameter of the last linear layer. Using the chain rule, we can easily derive the gradient for the last linear layer's parameter:

$$
\begin{aligned}
\frac{\partial \mathcal{L}^{\text{UFCE}}}{\partial \mathbf{w}_{c^*}} &= \left( \frac{\partial \mathbf{s}}{\partial \mathbf{w}_{c^*}} \right) \left( \frac{\partial \alpha_c}{\partial \mathbf{s}} \right) \left( \frac{\partial \mathcal{L}^{\text{UFCE}}}{\partial \alpha_{c^*}} \right), \\
\frac{\partial \mathcal{L}^{\text{UCE}}}{\partial \mathbf{w}_{c^*}} &= \left( \frac{\partial \mathbf{s}}{\partial \mathbf{w}_{c^*}} \right) \left( \frac{\partial \alpha_c}{\partial \mathbf{s}} \right) \left( \frac{\partial \mathcal{L}^{\text{UCE}}}{\partial \alpha_{c^*}} \right).
\end{aligned}
\qquad (31)
$$

This establishes a connection between the gradient of the last layer's weight and the gradient of the loss functions with respect to $\alpha_{c^*}$. Thus, $\left\| \frac{\partial \mathcal{L}^{\text{UFCE}}}{\partial \alpha_c} \right\| - \left\| \frac{\partial \mathcal{L}^{\text{UCE}}}{\partial \alpha_c} \right\|$ implies $\left\| \frac{\partial \mathcal{L}^{\text{UFCE}}}{\partial \mathbf{w}_{c^*}} \right\| - \left\| \frac{\partial \mathcal{L}^{\text{UCE}}}{\partial \mathbf{w}_{c^*}} \right\|$.

The uncertainty cross entropy loss is given by:

$$
\mathcal{L}^{\text{UCE}} = \mathbb{E}_{\mathbf{p} \sim \text{Dir}(\boldsymbol{\alpha})} \left[ -\sum_{c=1}^{C} y_c \log p_c \right] = \sum_{c=1}^{C} y_c [\psi(\alpha_0) - \psi(\alpha_c)] = \psi(\alpha_0) - \psi(\alpha_{c^*}) \qquad (32)
$$

Its gradient is given by the difference of trigamma functions:

$$
\frac{\partial \mathcal{L}^{\text{UCE}}}{\partial \alpha_{c^*}} = \psi_1(\alpha_0) - \psi_1(\alpha_{c^*}). \qquad (33)
$$

According to Proposition 5, we have:

$$\frac{\partial \mathcal{L}^{\text{UFCE}}}{\partial \alpha_{c^*}} = \frac{\Gamma(\alpha_0)\Gamma(\alpha_0 - \alpha_{c^*} + \gamma)}{\Gamma(\alpha_0 + \gamma)\Gamma(\alpha_0 - \alpha_{c^*})} \left\{ [\psi(\alpha_0) - \psi(\alpha_0 + \gamma)] [\psi(\alpha_0 + \gamma) - \psi(\alpha_{c^*})] + [\psi_1(\alpha_0 + \gamma) - \psi_1(\alpha_{c^*})] \right\}.$$

The difference between the l1-norm of gradients is:

$$\left| \frac{\partial \mathcal{L}^{\text{UFCE}}}{\partial \alpha_0} \right| - \left| \frac{\partial \mathcal{L}^{\text{UCE}}}{\partial \alpha_0} \right| = -\frac{\Gamma(\alpha_0)\Gamma(\alpha_0 - \alpha_{c^*} + \gamma)}{\Gamma(\alpha_0 + \gamma)\Gamma(\alpha_0 - \alpha_{c^*})} \Big\{ [\psi(\alpha_0) - \psi(\alpha_0 + \gamma)] [\psi(\alpha_0 + \gamma) - \psi(\alpha_{c^*})] +$$

$$[\psi_1(\alpha_0 + \gamma) - \psi_1(\alpha_{c^*})] \Big\} + (\psi_1(\alpha_0) - \psi_1(\alpha_{c^*})), \quad (34)$$

where $\frac{\partial \mathcal{L}^{\text{UFCE}}}{\partial \alpha_0}$ and $\frac{\partial \mathcal{L}^{\text{UCE}}}{\partial \alpha_0}$ can be shown to be negative for all $\alpha_0 > 2$, $\alpha_{c^*} > 1$, and $\gamma \in [0, 5]$.

Let $\bar{p}_{c^*} = \frac{\alpha_{c^*}}{\alpha_0}$ and $\alpha_{c^*} = \bar{p}_{c^*} \alpha_0$. We now analyze the relation between the difference and the projected class probability term $\bar{p}_{c^*}$. Rewriting the difference based on $\bar{p}_{c^*}$, we denote the resulting form as $f(\bar{p}_{c^*}, \alpha_0, \gamma)$:

$$f(\bar{p}_{c^*}, \alpha_0, \gamma) = -\frac{\Gamma(\alpha_0)\Gamma(\alpha_0 - \bar{p}_{c^*}\alpha_0 + \gamma)}{\Gamma(\alpha_0 + \gamma)\Gamma(\alpha_0 - \bar{p}_{c^*}\alpha_0)} \Big\{ [\psi(\alpha_0) - \psi(\alpha_0 + \gamma)] [\psi(\alpha_0 + \gamma) - \psi(\bar{p}_{c^*}\alpha_0)] +$$

$$[\psi_1(\alpha_0 + \gamma) - \psi_1(\bar{p}_{c^*}\alpha_0)] \Big\} + (\psi_1(\alpha_0) - \psi_1(\bar{p}_{c^*}\alpha_0)).$$

Next, we derive the gradient of this difference function with respect to $\bar{p}_{c^*}$. Let:

$$A(\bar{p}_{c^*}) = \frac{\Gamma(\alpha_0)\Gamma(\alpha_0 - \bar{p}_{c^*}\alpha_0 + \gamma)}{\Gamma(\alpha_0 + \gamma)\Gamma(\alpha_0 - \bar{p}_{c^*}\alpha_0)},$$

$$B(\bar{p}_{c^*}) = [\psi(\alpha_0) - \psi(\alpha_0 + \gamma)] [\psi(\alpha_0 + \gamma) - \psi(\bar{p}_{c^*}\alpha_0)] + [\psi_1(\alpha_0 + \gamma) - \psi_1(\bar{p}_{c^*}\alpha_0)].$$

Then,

$$f(\bar{p}_{c^*}; \alpha_0, \gamma) = -A(\bar{p}_{c^*})B(\bar{p}_{c^*}) + (\psi_1(\alpha_0) - \psi_1(\bar{p}_{c^*}\alpha_0)).$$

First, we find $\frac{\partial A(\bar{p}_{c^*})}{\partial \bar{p}_{c^*}}$. Recall that the derivative of the Gamma function with respect to its argument is: $\frac{d}{dz}\Gamma(z) = \Gamma(z)\psi(z)$. Using the product rule and the chain rule:

$$\frac{\partial A(\bar{p}_{c^*})}{\partial \bar{p}_{c^*}} = \frac{\Gamma(\alpha_0)}{\Gamma(\alpha_0 + \gamma)} \left[ \frac{d}{d\bar{p}_{c^*}} \left( \frac{\Gamma(\alpha_0 - \bar{p}_{c^*}\alpha_0 + \gamma)}{\Gamma(\alpha_0 - \bar{p}_{c^*}\alpha_0)} \right) \right]$$

$$= \frac{\Gamma(\alpha_0)}{\Gamma(\alpha_0 + \gamma)} \left[ \frac{d}{d\bar{p}_{c^*}} \frac{\Gamma(\alpha_0 - \bar{p}_{c^*}\alpha_0 + \gamma) \left[ -\alpha_0\psi(\alpha_0 - \bar{p}_{c^*}\alpha_0 + \gamma) + \alpha_0\psi(\alpha_0 - \bar{p}_{c^*}\alpha_0) \right]}{\Gamma(\alpha_0 - \bar{p}_{c^*}\alpha_0)} \right]$$

$$= \frac{\Gamma(\alpha_0 - \bar{p}_{c^*}\alpha_0 + \gamma)}{\Gamma(\alpha_0 - \bar{p}_{c^*}\alpha_0)} \left[ -\alpha_0\psi(\alpha_0 - \bar{p}_{c^*}\alpha_0 + \gamma) + \alpha_0\psi(\alpha_0 - \bar{p}_{c^*}\alpha_0) \right]$$

$$= A(\bar{p}_{c^*}) \left[ -\alpha_0\psi(\alpha_0 - \bar{p}_{c^*}\alpha_0 + \gamma) + \alpha_0\psi(\alpha_0 - \bar{p}_{c^*}\alpha_0) \right].$$

$$(35)$$

Next, we find $\frac{\partial B(p)}{\partial \bar{p}_{c^*}}$:

$$\frac{\partial B(\bar{p}_{c^*})}{\partial \bar{p}_{c^*}} = [\psi(\alpha_0) - \psi(\alpha_0 + \gamma)] [-\alpha_0\psi_1(\bar{p}_{c^*}\alpha_0)] + [-\alpha_0\psi_2(\bar{p}_{c^*}\alpha_0)]. \quad (36)$$

Then, we find the total derivative of $f(\bar{p}_{c^*}; \alpha_0, \gamma)$:

$$\frac{\partial f(\bar{p}_{c^*}; \alpha_0, \gamma)}{\partial \bar{p}_{c^*}} = -\frac{\partial A(\bar{p}_{c^*})}{\partial \bar{p}_{c^*}} B(\bar{p}_{c^*}) - A(\bar{p}_{c^*})\frac{\partial B(\bar{p}_{c^*})}{\partial \bar{p}_{c^*}} + (-\alpha_0\psi_2(\bar{p}_{c^*}\alpha_0))$$

$$= -A(\bar{p}_{c^*}) \left[ -\alpha_0\psi(\alpha_0 - \bar{p}_{c^*}\alpha_0 + \gamma) + \alpha_0\psi(\alpha_0 - \bar{p}_{c^*}\alpha_0) \right] B(\bar{p}_{c^*})$$

$$- A(\bar{p}_{c^*}) \left[ [\psi(\alpha_0) - \psi(\alpha_0 + \gamma)] [-\alpha_0\psi_1(\bar{p}_{c^*}\alpha_0)] + [-\alpha_0\psi_2(\bar{p}_{c^*}\alpha_0)] \right]$$

$$- \alpha_0\psi_2(\bar{p}_{c^*}\alpha_0)$$

$$= A(\bar{p}_{c^*})\alpha_0 \left[ \psi(\alpha_0 - \bar{p}_{c^*}\alpha_0 + \gamma) - \psi(\alpha_0 - \bar{p}_{c^*}\alpha_0) \right] B(\bar{p}_{c^*})$$

$$- A(\bar{p}_{c^*})\alpha_0 \left[ [\psi(\alpha_0) - \psi(\alpha_0 + \gamma)] \psi_1(\bar{p}_{c^*}\alpha_0) + \psi_2(\bar{p}_{c^*}\alpha_0) \right] - \alpha_0\psi_2(\bar{p}_{c^*}\alpha_0).$$

$$(37)$$

Noting that this derivative involves polynomial terms of polygamma functions, which are differentiable on $(0, +\infty)$, this verifies $f(\cdot)$ is also differentiable on $(0, +\infty)$. Given that $\alpha_0 \geq \alpha_{c^*} + 1$ and $\alpha_{c^*} \geq 1$, the feature range of $\bar{p}_{c^*}$ is $\left(\frac{1}{\alpha_0}, 1 - \frac{1}{\alpha_0}\right)$, $f(\cdot)$ remains differentiable in this interval. We demonstrate that $\frac{\partial f(\bar{p}_{c^*};\alpha_0,\gamma)}{\partial \bar{p}_{c^*}}$ is positive and negative at the bounding points of this range, respectively. According to the intermediate-value theorem, there exists at least one configuration of $\bar{p}_{c^*}$ in this range such that $\frac{\partial f(\bar{p}_{c^*};\alpha_0,\gamma)}{\partial \bar{p}_{c^*}} = 0$. Therefore, there exist two thresholds $\tau_1(\alpha_{c^*}, \gamma)$ and $\tau_2(\alpha_{c^*}, \gamma)$ within $\left(\frac{1}{\alpha_0}, 1 - \frac{1}{\alpha_0}\right)$, such that $\left|\frac{\partial \mathcal{L}^{\text{UFCE}}}{\partial \alpha_{c^*}}\right| > \left|\frac{\partial \mathcal{L}^{\text{UCE}}}{\partial \alpha_{c^*}}\right|$ when $\bar{p}_{c^*} < \tau_1(\alpha_{c^*}, \gamma)$, and $\left|\frac{\partial \mathcal{L}^{\text{UFCE}}}{\partial \alpha_{c^*}}\right| < \left|\frac{\partial \mathcal{L}^{\text{UCE}}}{\partial \alpha_{c^*}}\right|$ when $\bar{p}_{c^*} > \tau_2(\alpha_{c^*}, \gamma)$.

First, substitute $\bar{p}_{c^*} = \frac{1}{\alpha_0}$:

$$
\begin{aligned}
f\left(\frac{1}{\alpha_0}; \alpha_0, \gamma\right) = &-\frac{\Gamma(\alpha_0)\Gamma(\alpha_0 - 1 + \gamma)}{\Gamma(\alpha_0 + \gamma)\Gamma(\alpha_0 - 1)}\Big\{ [\psi(\alpha_0) - \psi(\alpha_0 + \gamma)][\psi(\alpha_0 + \gamma) - \psi(1)] \\
&+ [\psi_1(\alpha_0 + \gamma) - \psi_1(1)] \Big\} + (\psi_1(\alpha_0) - \psi_1(1)).
\end{aligned}
\tag{38}
$$

Using the properties of the Gamma function, $\Gamma(\alpha_0 - 1) = \frac{\Gamma(\alpha_0)}{\alpha_0 - 1}$ and $\psi(1) = -\gamma_E$ and $\psi_1(1) = \frac{\pi^2}{6}$, where $\gamma_E$ is the Euler-Mascheroni constant, the expression simplifies to:

$$
\begin{aligned}
f\left(\frac{1}{\alpha_0}; \alpha_0, \gamma\right) = &-\frac{\alpha_0 - 1}{\alpha_0 + \gamma - 1}\Big\{ [\psi(\alpha_0) - \psi(\alpha_0 + \gamma)][\psi(\alpha_0 + \gamma) - \psi(1)] \\
&+ [\psi_1(\alpha_0 + \gamma) - \psi_1(1)] \Big\} + (\psi_1(\alpha_0) - \psi_1(1)) \\
= &-\frac{\alpha_0 - 1}{\alpha_0 + \gamma - 1}\Big\{ [\psi(\alpha_0) - \psi(\alpha_0 + \gamma)][\psi(\alpha_0 + \gamma) + \gamma_E] \\
&+ \left[\psi_1(\alpha_0 + \gamma) - \frac{\pi^2}{6}\right] \Big\} + \left(\psi_1(\alpha_0) - \frac{\pi^2}{6}\right).
\end{aligned}
\tag{39}
$$

It is evident from numerical analysis that $f\left(\frac{1}{\alpha_0}; \alpha_0, \gamma\right) > 0$ for all $\gamma \in [0, 5]$ and $\alpha_0 > 2$.

Substituting $\bar{p}_{c^*} = 1 - \frac{1}{\alpha_0}$, we have

$$
\begin{aligned}
f\left(1 - \frac{1}{\alpha_0}; \alpha_0, \gamma\right) = &-\frac{\Gamma(\alpha_0)\Gamma(1 + \gamma)}{\Gamma(\alpha_0 + \gamma)\Gamma(1)}\Big\{ [\psi(\alpha_0) - \psi(\alpha_0 + \gamma)][\psi(\alpha_0 + \gamma) - \psi(\alpha_0 - 1)] \\
&+ [\psi_1(\alpha_0 + \gamma) - \psi_1(\alpha_0 - 1)] \Big\} + (\psi_1(\alpha_0) - \psi_1(\alpha_0 - 1)).
\end{aligned}
\tag{40}
$$

It is evident from numerical analysis that $f\left(1 - \frac{1}{\alpha_0}; \alpha_0, \gamma\right) < 0$ for all $\gamma \in [0, 5]$ and $\alpha_0 > 2$.

$\square$

## A.2 RELATED WORK

### A.2.1 VISION-BASED BIRD'S EYE VIEW SEMANTIC SEGMENTATION

Bird's-eye view (BEV) serves as an effective representation for fusing information from multiple cameras, making it a central component in autonomous driving systems. However, transforming camera images into BEV maps presents significant challenges, primarily due to the complexity of depth estimation and 3D geometric transformations. The key component of BEVSS is the 2D-to-3D lifting strategy, which employs techniques such as depth-weighted splitting (Philion & Fidler, 2020; Hu et al., 2021), attention mechanisms (Zhou & Krähenbühl, 2022), and bilinear sampling (Harley et al., 2023).

In this paper, we use three representative methods, Lift splat shoot (LSS) (Philion & Fidler, 2020), Cross-View Transformer (CVT) (Zhou & Krähenbühl, 2022), and Simple-BEV (Harley et al., 2023).

Figure 2: Numerical analysis of $\left\|\frac{\partial \mathcal{L}^{\mathrm{Ufce}}}{\partial \mathbf{w}_{c*}}\right\| - \left\|\frac{\partial \mathcal{L}^{\mathrm{Uce}}}{\partial \mathbf{w}_{c*}}\right\|$ for different composition of $\alpha_{c*}$ and $\gamma$

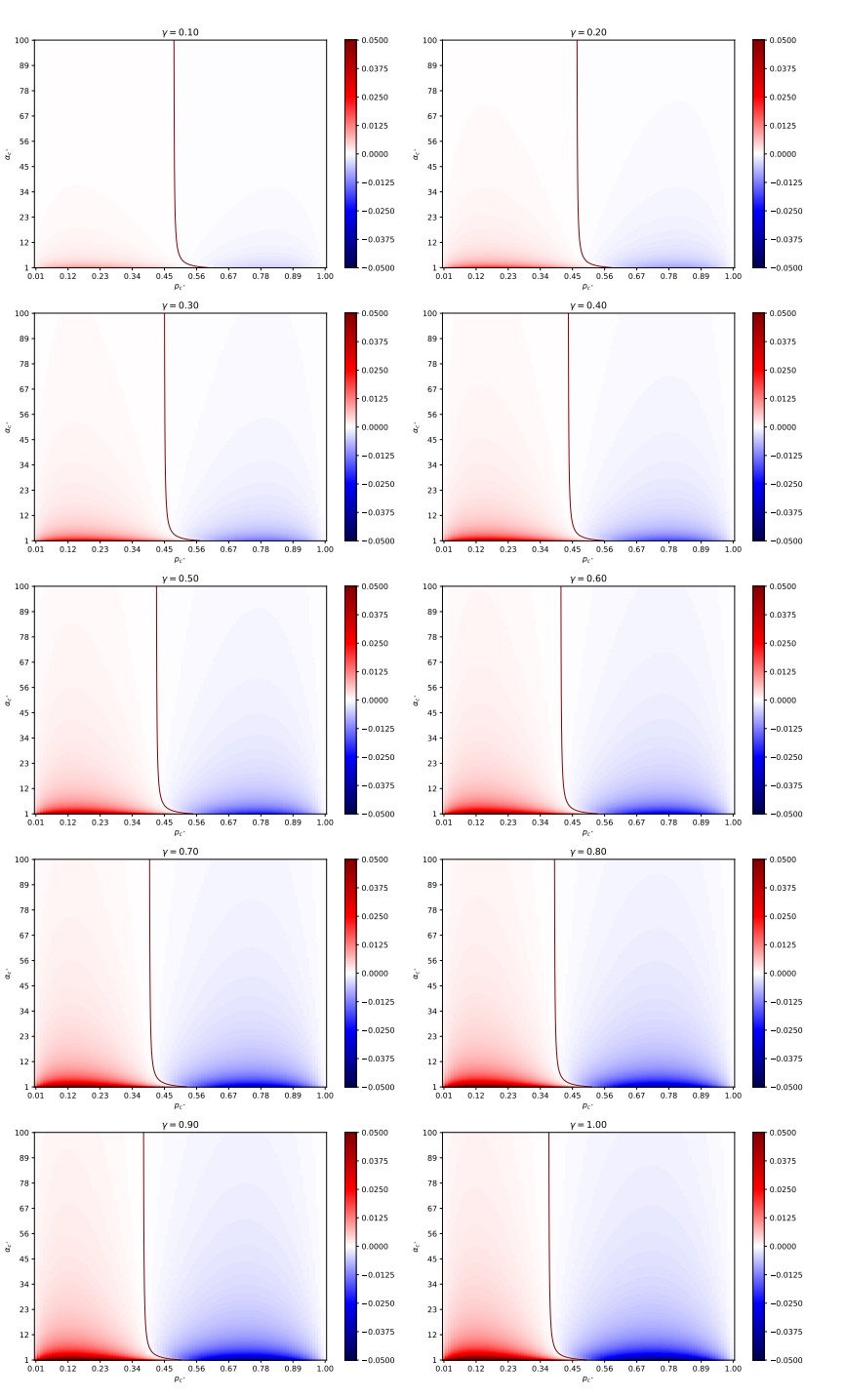

Figure 3: Implicit weight regularization impact by UFCE with $\alpha_{c^*} = 10$

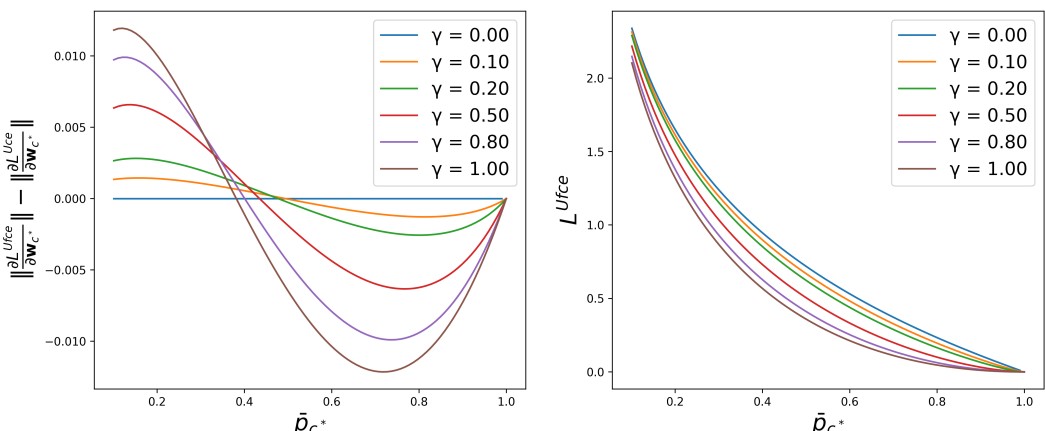

**LSS** leverages raw pixel inputs from multiple surrounding cameras and "lifts" each image individually into a frustum of features. Initially, it predicts a categorical distribution over a predefined set of possible depths. Subsequently, the frustum of features is generated by multiplying the features with their predicted depth probability. By utilizing known camera calibration matrices for each camera, a point cloud of features in the ego coordinate space can be obtained. LSS then "splats" all the frustums into a rasterized bird's-eye-view grid using a PointPillar (Lang et al., 2018) model. These splatted features are then fed into a decoder module to predict BEVSS. The concept of transforming from camera pixels to 3D point clouds and subsequently to BEV pixels has inspired several subsequent models, such as BEVDet (Huang et al., 2021) and FIERY (Hu et al., 2021).

**CVT** takes a distinct approach by leveraging transformer architecture and cross-attention mechanism. CVT begins by extracting features from multiple surrounding camera images using a pre-trained EfficientNet-B4(Tan & Le, 2019) model. These extracted features serve as the attention values in the subsequent cross-attention step. To create the attention keys, the features are concatenated with the camera-aware positional embedding. This positional embedding is constructed using known camera pose and intrinsic information, enabling the model to account for the specific characteristics of each camera. The positional encoding of the BEV space serves as the queries during the cross attention process.

**Simple-BEV** simplifies the process of generating BEV representations by avoiding the use of estimated depth maps. Instead, it defines a 3D coordinate volume over the BEV plane and projects each coordinate into the corresponding camera images. Image features are then sampled from the surrounding regions of the projected locations. While the resulting features are not precisely aligned in the BEV space and are distributed across potential locations, this approach significantly enhances efficiency and robustness to projection errors. PointBEV (Chambon et al., 2024) further improves the 'feature pulling' strategy with the sparse representations.

### A.2.2 Uncertainty quantification on Bird's eye view semantic segmentation

To the best of our knowledge, there is no pioneer work on the uncertainty quantification on the BEVSS task. We introduce the most relative literature in the section.

**Uncertainty quantification on input-dependent data**: The uncertainty quantification methods used in this work are primarily designed for input-independent data, such as images for classification. However, developing uncertainty quantification techniques tailored for input-dependent data remains underexplored but necessary. Zhao et al. (2020) applied evidential deep learning to node-level graph tasks, while Hart et al. (2023); Stadler et al. (2021) incorporated graph topology relationships. He et al. (2023) explored uncertainty in text classification through conservative learning, and He et al. (2024) investigated evidential deep learning for sequential data. We consider this an important direction for future research.

**Uncertainty quantification in camera view's semantic segmentation**. Uncertainty quantification in pixel-level camera semantic segmentation is the most relevant research task. Similarly, uncertainty in this task also arises from two primary sources: aleatoric uncertainty, which reflects inherent ambiguity in the data (e.g., sensor noise or occlusions), and epistemic uncertainty, which represents the model's lack of knowledge due to limited or biased training data. Mukhoti & Gal (2018) evaluated Bayesian deep learning methods, including MC Dropout and Concrete Dropout, applied to semantic segmentation and introduced new patch-based evaluation metrics. However, these metrics focused on the relationship between accuracy and uncertainty, limited to aleatoric uncertainty. In addition, Bayesian-based approaches require multiple forward passes, making it impractical for real-time applications such as autonomous driving. Mukhoti et al. (2021) proposed single-pass uncertainty estimation by extending deep deterministic uncertainty to semantic segmentation tasks. They used Gaussian Discriminant Analysis (GDA) to model feature space means and covariance matrices per class, enabling the quantification of epistemic uncertainty through feature densities, while aleatoric uncertainty was estimated via the softmax distribution. However, this work did not provide a comprehensive quantitative evaluation of the quality of the estimated epistemic uncertainty. Ancha et al. (2024) advanced uncertainty estimation with a model based on the evidential deep learning framework. Their approach is built on a natural posterior network that estimates a pixel-level Dirichlet distribution with a density estimator based on a GMM-enhanced normalizing flow and linear classifier. To mitigate feature collapsing issues, they introduced a decoder that reconstructs image patches from pixel-level latent features. The proposed model is evaluated from both calibration view and OOD detection-like tasks. However, the heavy modification on the network architecture makes it hard to apply to the BEVSS task. Yu et al. (2024) constructed a graph over image pixels and incorporated graph topology into uncertainty estimation. However, this method relies on high-dimensional pixel features from hyperspectral images, making it unsuitable for RGB images.

**Fusing camera view uncertainties to BEV space: Challenges and Opportunities**. Uncertainty quantification in BEV semantic segmentation (BEVSS) lacks established literature, and extending methods from the camera space to BEV is non-trivial. To illustrate this challenge, we conducted a simple experiment. Using the CARLA dataset, we trained an Evidential Neural Network (ENN) model based on DeepLabV3 to quantify pixel-level uncertainty in camera images. The predicted uncertainty was then mapped from the camera space to the BEV space using a ground truth mapping function, with animals included as the OOD category. Experimental results showed an average AUPR of 47% in the camera space, but only 7% in the BEV space.

This discrepancy arises possibly because the predicted OOD regions are often larger than the actual objects. When mapped to BEV, these over-predicted regions expand further due to perspective distortion. Conversely, correctly predicted regions are restricted to the visible parts of the object, resulting in significant inaccuracies and poor performance in BEV. Finally, we highlight that incorporating camera-view uncertainties can enhance the quality of uncertainty quantification in BEV, offering a promising direction for future research.

**Robutness in bird's eye view semantic segmentation**. OOD robustness in BEV semantic segmentation focuses on maintaining model performance under novel and unexpected conditions, such as varying weather, lighting changes, or sensor malfunctions. RoboBEV Xie et al. (2024) provides an extensive benchmark for evaluating the robustness of BEV perception models in autonomous driving, testing their performance under various natural corruptions. For instance, they demonstrate significant performance drops in segmentation tasks when evaluating the CVT model trained on clean datasets against corrupted datasets.

We further clarify the distinction between OOD robustness and OOD detection within our experimental context. OOD robustness (or generalization) aims to ensure that models maintain high performance on OOD samples with domain shifts. In contrast, OOD detection emphasizes model reliability by identifying samples with semantic shifts—cases where the model cannot or should not generalize. Importantly, these concepts can be complementary: an uncertainty-aware model should also be robust to domain shifts, meaning it must maintain effective OOD detection even under conditions of domain shift.

**Calibration in bird's eye view semantic segmentation**. There indeed several works focus primarily on calibration while neglecting challenges such as out-of-distribution data and dataset shifts, which require epistemic uncertainty estimation. Kängsepp & Kull (2022) focus on overconfident probability estimates in semantic segmentation issue by applying isotonic regression for pixel-wise calibration

and beta calibration for object-wise calibration. This method improves confidence reliability but does not explicitly address epistemic uncertainty. MapPrior (Zhu et al., 2023) integrates a generative prior with traditional discriminative BEV models to enhance BEV semantic segmentation performance while focusing on calibration, particularly aleatoric uncertainty. It combines an initial noisy BEV layout estimate from a predictive model with a generative refinement stage that samples diverse outputs from a transformer-based latent space. However, MapPrior's approach is limited to aleatoric uncertainty and relies on an additional model architecture, making it less adaptable to other BEV semantic segmentation models.

**Uncertainty quantification on downstream tasks**. Several works have explored uncertainty in downstream tasks that use bird's-eye view (BEV) maps as intermediate representations. However, these work are also limited to calibration. UAP (Dewangan et al., 2023) targets motion planning and trajectory optimization by building an uncertainty-aware occupancy grid map. It estimates collision probabilities based on sampled distances to nearby occupied cells and optimizes trajectories using a sampling-based approach, incorporating uncertainty into the planning process. Fervers et al. (2023) focus on metric cross-view geolocalization (CVGL), which aims to localize ground-based vehicles relative to aerial map images. Their approach matches BEV representations to aerial images by predicting soft probability distributions over possible vehicle poses. The uncertainty derived from these probability distributions is integrated into a Kalman filter, improving the accuracy and robustness of trajectory tracking over time.

### A.2.3 UNCERTAINTY QUANTIFICATION BASELINES

**Softmax-based (Hendrycks & Gimpel, 2016)** Softmax entropy is one of the most commonly used metrics for uncertainty (Hendrycks & Gimpel, 2016). It is the entropy ($\mathbb{H}(p(\mathbf{Y}_{i,j}|\mathbf{X};\boldsymbol{\theta}))$) of softmax distribution $p(\mathbf{Y}_{i,j}|\mathbf{X};\boldsymbol{\theta})$.

$$\mathbb{H}(p(\mathbf{Y}_{i,j}|\mathbf{X};\boldsymbol{\theta})) = \sum_{c=1}^{C} p_{c,i,j} \log p_{c,i,j}. \tag{41}$$

This metric is known to capture aleatoric uncertainty, but can not capture epistemic uncertainty reliably.

**Energy-based (Liu et al., 2020)** The energy-based model is designed to distinguish OOD data from ID data. Energy scores, which are theoretically aligned with the input's probability density, exhibit reduced susceptibility to overconfidence issues and can be considered as epistemic uncertainty. As a post-hoc method applied to predicted logits, energy scores can be flexibly utilized as a scoring function for any pre-trained neural classifier.

$$u_{i,j}^{epis} = u_{i,j}^{energy} = -T \cdot \log \sum_{c=1}^{C} \exp^{l_{c,i,j}} / T \tag{42}$$

where $l_{c,i,j}$ represents the predicted logits for pixel $(i, j)$ associated with class $c$, where logits are defined as the output of the final layer prior to the softmax activation function in a standard classification model, i.e. $\mathbf{L} = f(\mathbf{X};\boldsymbol{\theta})$. $T$ is the temperature scaling. The energy score will be employed to quantify epistemic uncertainty. There is no aleatoric uncertainty explicitly defined for energy-based models, so we use the softmax confidence score to measure aleatoric uncertainty.

$$u_{i,j}^{alea} = u_{i,j}^{conf} = -\max_c p_{c,i,j} \tag{43}$$

**Ensembles-based (Lakshminarayanan et al., 2017).** Deep Ensembles-based method learn $M$ different versions of network weights $\{\boldsymbol{\theta}^{(1)}, \cdots, \boldsymbol{\theta}^{(M)}\}$ and aggregates the predictions of these versions. The aleatoric uncertainty is measured by the softmax entropy of the mean of the predictions from different network weights. The epistemic uncertainty is measured by the variance between the model predictions.

$$
\begin{aligned}
u_{i,j}^{alea} &= \sum_{c=1}^{C} \left( \frac{1}{M} \sum_{m=1}^{M} p_{c,i,j}^{(m)} \right) \log \left( \frac{1}{M} \sum_{m=1}^{M} p_{c,i,j}^{(m)} \right) \\
u_{i,j}^{epis} &= \text{var}(\{p_{\hat{c},i,j}^{(m)}\}_{m=1}^{M}), \quad \hat{c} = \text{argmax} \left( \frac{1}{M} \sum_{m=1}^{M} \boldsymbol{p}_{i,j}^{(m)} \right)
\end{aligned}
\tag{44}
$$

where $p_{i,j}^{(m)} \in [0,1]^C$ refers to the predictions of BEV network based on the network weights $\boldsymbol{\theta}^{(m)}$. We use $M = 3$ for experiments.

**Dropout-based** (Gal & Ghahramani, 2016). Dropout-based methods approximate Bayesian inference based on activated dropout layers. It conducts multiple stochastic forward passes with active dropout layers at test time. Similar to Deep ensembles, the entropy of expected softmax probability and variance of multiple predictions are used as aleatoric and epistemic uncertainty scores, respectively. We use $M = 10$ for experiments.

**Energy-bounded Learning (EB).** For a fair comparison, we also include OOD-exposure for an energy-based model. With the same strategy as Liu et al. (2020), we consider energy-bounded learning for OOD detection. This approach entails fine-tuning the neural network by assigning lower energy levels to ID data and higher energy levels to OOD data. Specifically, the classification model is trained using the following objective function:

$$\mathcal{L}^{\text{CeEb}} = \mathbb{E}_{(\boldsymbol{X},\boldsymbol{y})\sim\mathcal{D}_{in}^{train}} \mathbb{H}(\boldsymbol{p},\boldsymbol{y}) + \eta \cdot \mathcal{L}^{\text{Eb}} \tag{45}$$

where $\boldsymbol{p}$ is the predicted class probabilities and $\mathcal{L}^{\text{Eb}}$ is the regularization loss defined in terms of energy:

$$\mathcal{L}^{\text{Eb}} = \mathbb{E}_{(\boldsymbol{X}_{in},\boldsymbol{y})\sim\mathcal{D}_{in}^{train}} \left( \max\left(0 \ , \ u_{\boldsymbol{X}_{in}}^{energy} - m_{in}\right)\right)^2 \tag{46}$$

$$= \mathbb{E}_{(\boldsymbol{X}_{out},\boldsymbol{y})\sim\mathcal{D}_{out}^{train}} \left( \max\left(0 \ , \ m_{out} - u_{\boldsymbol{X}_{out}}^{energy}\right)\right)^2 \tag{47}$$

where the $m_{in}, m_{out}$ are two bounded hyperparamters. Details can be found in Liu et al. (2020). We can also replace cross entropy loss with focal loss for better calibration:

$$\mathcal{L}^{\text{FceEb}} = \mathbb{E}_{(\boldsymbol{X},\boldsymbol{y})\sim\mathcal{D}_{in}^{train}} \text{Focal}(\boldsymbol{p},\boldsymbol{y}) + \eta \cdot \mathcal{L}^{\text{Eb}} \tag{48}$$

where $\text{Focal}(\boldsymbol{p},\boldsymbol{y}) = -\sum_{c=1}^{C} y_c (1-p_c)^{\gamma} \log p_c$.

## A.3 IMPLEMENTATION DETAILS

For the model training, we will use equation 49 for the original ENN model optimized by Bayesian loss and equation 50 for our proposed UFCE-based models.

$$\mathcal{L}^{\text{UCE-ENT-ER}} = \mathcal{L}^{\text{UCE-ENT}} + \lambda \mathcal{L}^{\text{ER}} \tag{49}$$

$$\mathcal{L}^{\text{UFCE-ER}} = \mathcal{L}^{\text{UFCE}} + \lambda \mathcal{L}^{\text{ER}} \tag{50}$$

It is important to highlight that $\mathcal{L}^{\text{ER}}$ and $\mathcal{L}^{\text{ENT}}$ represent distinct concepts within our framework. $\mathcal{L}^{\text{ER}}$, identified as a *target loss*, assumes that the level-2 ground truth for pseudo OOD samples corresponds to a flat Dirichlet distribution, reflecting the actual discrepancy between predictions and ground truth. Conversely, $\mathcal{L}^{\text{ENT}}$ is characterized as a *surrogate loss*, fulfilling an auxiliary role. Through the employment of the ENT regularizer, the model is implicitly encouraged to predict a smoother distribution for all training samples.

### A.3.1 HYPERPARAMETERS

**Fixed hyperparamters**: We set the weight of the entropy regularization term in the UCE loss ($\beta$ in Equation 5) to 0.001. The energy-bounded regularization weight in the energy model ($\eta$ in Equations 46 and 48) is set to 0.0001. For all models where "vehicle" is considered the positive class, we assign a positive class weight of 2. In scenarios where "drivable region" is a positive class, the class weight is set to 1. We employ learning rates of $4 \times 10^{-3}$ for focal loss variants and $1 \times 10^{-3}$ for cross-entropy variants, using the Adam optimizer with a weight decay of $1 \times 10^{-7}$. The batch size is set to 32 across all experimental scenarios.

**Hyperparameters tuning strategy**: We tune three regularization weights related to our proposed model: $\lambda$, $\xi$, and $\gamma$. Due to computational constraints, we did not perform a full grid search over these hyperparameters. Instead, we conducted a step-by-step search based on the pseudo-OOD detection task, specifically optimizing for the AUPR metric evaluated on the validation set. First, we performed a coarse grid search over $\lambda \in \{0.001, 0.005, 0.01, 0.05, 0.1\}$ with $\xi = 0$ and $\gamma = 0$, finding that

$\lambda = 0.01$ generally provided good performance across most scenarios. Next, with $\lambda$ fixed at 0.01, we tuned $\xi \in \{8, 16, 32, 64, 128\}$ with $\gamma = 0$, and selected $\xi = 64$ for the remaining experiments. Finally, we fine-tuned $\gamma \in \{0.05, 0.5, 1.5\}$ while keeping $\lambda = 0.01$ and $\xi = 64$ fixed. The detailed hyperparameters used in our reported results are listed in Table 6.

Table 6: Hyperparameter Settings for Different Datasets and Methods

| Dataset | Method | $\gamma$ | $\lambda$ | $\xi$ |
|---------|--------|----------|-----------|-------|
| CARLA | LSS | 0.5 | 0.01 | 64 |
| | CVT | 0.05 | 0.01 | 64 |
| | Simple-BEV | 0.05 | 0.01 | 64 |
| nuScenes | LSS | 1 | 0.01 | 64 |
| | CVT | 0.05 | 0.01 | 64 |
| | Simple-BEV | 0.05 | 0.01 | 64 |
| Lyft | LSS | 1 | 0.01 | 64 |
| | CVT | 0.05 | 0.01 | 64 |
| | Simple-BEV | 0.05 | 0.01 | 64 |

**Hyperparameters Sensitivity Analysis**: To evaluate the robustness of our model with respect to its hyperparameters, we conducted a comprehensive sensitivity analysis on the nuScenes dataset with vehicle segmentation task using the LSS model as the backbone.

We first analyze the impact of the hyperparameter $\lambda$ on model performance, with results presented in Table 7. Intuitively, a larger $\lambda$ corresponds to a higher pseudo-OOD regularization weight, encouraging the model to predict a uniform Dirichlet distribution for pseudo-OOD pixels. However, this may also negatively affect the original task performance. Experimentally, we observe that larger $\lambda$ values lead to a decline in pure segmentation and calibration performance. Due to the observed significant variations in segmentation metrics and misclassification results are closely tied to segmentation performance, we ignore the misclassification detection performance here. For OOD detection, we emphasize the AUPR metric as a key indicator. With increasing $\lambda$, we initially observe an improvement in AUPR (accompanied by higher AUROC and lower FPR95), followed by a decline as $\lambda$ becomes too large. This indicates that while moderate pseudo-OOD regularization enhances OOD detection, overly strong regularization can degrade overall performance.

Table 7: Hyperpameter sensitivity analysis for varying $\lambda$. We conduct the vehicle segmentation with LSS as the backbone on nuScenes with $\xi = 64$, $\gamma = 0.5$.

| $\lambda$ | Pure Classification | | Misclassification | | | OOD Detection | | |
|-----------|--------------------|------|-------------------|------|---------|---------------|------|---------|
| | IoU ↑ | ECE ↓ | AUROC ↑ | AUPR ↑ | FPR95 ↓ | AUROC ↑ | AUPR ↑ | FPR95 ↓ |
| 0.001 | 0.361 | 0.00335 | 0.895 | 0.320 | 0.192 | 0.801 | 0.224 | 0.410 |
| 0.005 | 0.357 | 0.00327 | 0.890 | 0.318 | 0.199 | 0.833 | 0.225 | 0.340 |
| 0.01 | 0.343 | 0.00035 | 0.925 | 0.324 | 0.191 | 0.743 | 0.165 | 0.383 |
| 0.05 | 0.304 | 0.01090 | 0.936 | 0.300 | 0.233 | 0.840 | 0.115 | 0.276 |
| 0.1 | 0.259 | 0.01740 | 0.919 | 0.276 | 0.288 | 0.860 | 0.0537 | 0.230 |

We then analyze the impact of the hyperparameter $\gamma$ on model performance, with results presented in Table 8. We observe that varying $\gamma$ does not significantly affect segmentation performance. However, selecting an appropriate value for $\gamma$ leads to better calibration performance, as evidenced by the lower Expected Calibration Error (ECE) score, and improved aleatoric uncertainty prediction, indicated by better misclassification detection performance. This demonstrates that the UFCE loss can enhance both calibration and misclassification detection performance. Besides, we observe that OOD detection performance is quite sensitive to the selection of $\gamma$.

Finally, we investigate the effect of hyperparameter $\xi$ on model performance, with results presented in Table 9. First, We observe that increasing $\xi$ led to a slightly higher IoU), indicating improved segmentation accuracy (then we omit the misclassification detection performance considering the correlation between these two tasks). We noted that higher values of $\xi$ resulted in increased AUROC on OOD detection performance in general.

Table 8: Hyperpameter sensitivity analysis for varying $\gamma$. We conduct the vehicle segmentation with LSS as the backbone on nuScenes with $\xi = 0$, $\lambda = 0.1$.

| $\gamma$ | Pure Classification | | Misclassification | | | OOD Detection | | |
|---|---|---|---|---|---|---|---|---|
| | IoU ↑ | ECE↓ | AUROC↑ | AUPR↑ | FPR95↓ | AUROC↑ | AUPR↑ | FPR95↓ |
| 0.05 | 0.349 | 0.00378 | 0.913 | 0.321 | 0.195 | 0.876 | 0.184 | 0.278 |
| 0.5 | 0.348 | 0.00166 | 0.901 | 0.328 | 0.187 | 0.774 | 0.341 | 0.358 |
| 1.5 | 0.349 | 0.01170 | 0.924 | 0.331 | 0.183 | 0.634 | 0.334 | 0.473 |

Table 9: Hyperpameter sensitivity analysis for varying $\xi$. We conduct the vehicle segmentation with LSS as the backbone on nuScenes with $\gamma = 0.5$, $\lambda = 0.01$.

| $\xi$ | Pure Classification | | Misclassification | | | OOD Detection | | |
|---|---|---|---|---|---|---|---|---|
| | IoU ↑ | ECE ↓ | AUROC ↑ | AUPR ↑ | FPR95 ↓ | AUROC ↑ | AUPR ↑ | FPR95 ↓ |
| 0 | 0.348 | 0.00166 | 0.901 | 0.328 | 0.187 | 0.774 | 0.341 | 0.358 |
| 8 | 0.349 | 0.00079 | 0.913 | 0.327 | 0.186 | 0.785 | 0.273 | 0.348 |
| 16 | 0.351 | 0.00089 | 0.910 | 0.321 | 0.186 | 0.747 | 0.328 | 0.390 |
| 32 | 0.350 | 0.00154 | 0.900 | 0.324 | 0.192 | 0.822 | 0.321 | 0.315 |
| 64 | 0.351 | 0.00133 | 0.904 | 0.321 | 0.187 | 0.797 | 0.315 | 0.364 |
| 128 | 0.353 | 0.00137 | 0.893 | 0.321 | 0.192 | 0.818 | 0.334 | 0.355 |

## A.4 DATASET DETAILS

**Datasets.** We consider both synthetic and real-world datasets for our experiments and details are presented in Table 10. For synthetic data, we utilize the widely recognized CARLA simulator (Dosovitskiy et al., 2017) to collect our dataset, which will be made available upon request due to its large size. Our simulated CARLA dataset features five towns with varied layouts and diverse weather conditions to enhance dataset diversity. In terms of real-world data, we employ the nuScenes (Caesar et al., 2020) and Lyft (Kesten et al., 2019) dataset, which is also used in the evaluation of the two segmentation backbones used in our paper: LSS and CVT, as well as an updated leaderboard. We choose not to use the KITTI (Behley et al., 2019) dataset for several reasons. KITTY primarily features suburban streets with low traffic density and simpler traffic scenarios, with annotations limited to the front camera view instead of a full 360-degree perspective. It also lacks radar data and is designated for non-commercial use only. In contrast, nuScenes aims to enhance these features by providing dense data from both urban and suburban environments in Singapore and Boston.

Below, we provide detailed descriptions of each dataset: (1) nuScenes. This dataset comprises of 35661 samples. It offers a 360° view around the ego-vehicle through six camera perspectives, with each view providing both intrinsic and extrinsic details. We resize the camera images to 224x480 pixels, and produce Bird's-Eye-View (BEV) labels of 200x200 pixels for analysis. (2) Lyft. This dataset comprises of 22,888 samples. It offers a 360° view around the ego-vehicle through multiple camera perspectives, providing both intrinsic and extrinsic details for each view. We resize the camera images to 224x480 pixels and generate Bird's-Eye-View (BEV) labels at a resolution of 200x200 pixels for analysis. (3) CARLA. In this simulated environment, six cameras are installed at 60-degree intervals around the ego vehicle, emulating the setup found in nuScenes. The simulation includes various weather conditions (e.g., Clear Noon, Cloudy Noon, Wet Noon) and urban layouts (e.g., Town10, Town03) to enrich the dataset's diversity. For each camera, we capture and record intrinsic and extrinsic information. The dataset consists of 224x480 pixel camera images and 200x200 pixel BEV labels, similar to nuScenes. In total, 40,000 frames are collected for training, with an additional 10,000 frames designated for validation.

**True OOD and pseudo OOD setting**. We evaluate the quality of the predicted epistemic uncertainty through an out-of-distribution (OOD) detection task. In-Distribution (ID) pixels are those that belong to the segmentation task with clear labels, such as "vehicle" and "background". **True-OOD** pixels represent a semantic shift from the training data; they only appear in the test dataset and were not seen during training. The OOD detection performance reported in this paper identifies pixels as either ID or true OOD. **Pseudo-OOD** pixels are artificially designated as OOD during training to regularize the model, helping it learn to distinguish between ID and OOD pixels. These pseudo OOD pixels exist in the training and validation sets, and hyperparameter tuning is based on the pseudo-OOD

Table 10: Dataset details with diversity descriptions.

| Dataset | Camera Space | | | BEV Space | | | Diversity Description |
|---|---|---|---|---|---|---|---|
| | Cameras | Positions | Resolution | FOV | Scale | Resolution | |
| CARLA | 6 | 60-degree intervals around the ego vehicle | 224 × 480 | 90 | 100 m × 100 m around vehicle | 200 × 200 | Diverse scenes over 10 weather conditions (e.g., Clear Noon, Cloudy Noon, Wet Noon) and 5 urban layouts (e.g., Town10, Town03) |
| nuScenes | 6 | Front, Front-Left, Front-Right, Back-Left, Back-Right, Back | 224 × 480 | 70,110 | 100 m × 100 m around vehicle, 50 cm resolution | 200 × 200 | 1,000 diverse scenes collected over various weather, time of day, and traffic conditions |
| Lyft Level 5 | 6 | Front, Front-Left, Front-Right, Side-Left, Side-Right, Back | 224 × 480 | 82 | 150 m × 150 m around vehicle | Variable | Diverse urban driving scenes with varying traffic densities and environments in Palo Alto, California |

detection performance on the validation set. The detailed setting we used in this paper is presented in Table 11.

**Criteria used to select the pseudo-OOD data**. We initially adhered to the criteria outlined in the benchmark study by Franchi et al. (2022), a seminal work in the field of autonomous driving, to identify candidate OOD classes. In this context, OOD data typically encompasses less frequently encountered dynamic objects (e.g., motorcycles, bicycles, bears, horses, cows, elephants) and static objects (e.g., food stands, barriers) that are distinct from primary segmentation categories like vehicles, road regions, and pedestrians. These objects were designated as candidate OOD classes. Subsequently, in our experiments, we randomly partitioned these candidate classes into pseudo-OOD and true OOD categories.

It is worth noting that the OOD benchmark dataset MUAD (https://muad-dataset.github.io/) provided by Franchi et al. (2022) was collected using a simulator based on front-camera imagery for image segmentation. However, it does not include a collection of images from multiple cameras or labels for BEV segmentation. To address this, we adopted a similar procedure and generated a BEV segmentation dataset with OOD objects using the well-established CARLA simulator.

Table 11: Dataset configurations for different settings.

| Dataset | Setting 1 (Default) | | | Setting 2 | | | Setting 3 | | |
|---|---|---|---|---|---|---|---|---|---|
| | ID | Pseudo-OOD | True-OOD | ID | Pseudo-OOD | True-OOD | ID | Pseudo-OOD | True-OOD |
| CARLA | vehicle | bears, horses, cows, elephants | deer | vehicle | bears, horses, cows, elephants | kangaroo | n.a. | n.a. | n.a. |
| nuScenes | vehicle | bicycle | motorcycle | vehicle | traffic cones, pushable/pullable objects, motorcycles | barriers | drivable region | bicycle | motorcycle |
| Lyft | vehicle | bicycle | motorcycle | n.a. | n.a. | n.a. | n.a. | n.a. | n.a. |

**Splits for evaluation.** (1) nuScenes: Considering that only public training and validation sets are available, we report results on the validation set, following standard practice in the literature. Notably, we remove all frames that contain true OOD pixels from both the training and validation sets used for training and evaluation, respectively. For models that are not exposed to pseudo-OOD pixels during training, we also remove frames containing pseudo-OOD objects from the training set. (2) Lyft. We use the same split strategy as nuScenes. (3) CARLA. To introduce pseudo and true OOD objects into our synthetically generated dataset, we employ 3D models of specific objects, integrating these models into the CARLA simulator scenes to generate custom objects. We first gather clean datasets for training, validation, and testing separately, where each frame exclusively contains in-distribution objects. Then, we gather separate train-pseudo and val-pseudo datasets, incorporating pseudo-OOD objects into each scene. Finally, we collect a dataset featuring true OOD objects, specifically deer.

In Table 12, we provide the number of frames and OOD pixel ratio for splitter datasets.

Table 12: Dataset split information

| Dataset | train | train-aug | | val | val-aug | | test | |
|---------|-------|-----------|---|-----|---------|---|------|---|
| | No. frames | No. frames | Pseudo-OOD ratio | No. frames | No. frames | Pseudo-OOD ratio | No. frames | True-OOD ratio |
| CARLA | 40,000 | 80,000 | 0.11% | 40,000 | 80,000 | 0.14% | 2,000 | 0.07% |
| nuScenes | 19,208 | 23,831 | 0.017% | 3,878 | 5,082 | 0.012% | 1,204 | 0.02% |
| Lyft | 11,487 | 16,184 | 0.18% | 4,431 | 6,094 | 0.016% | 113 | 0.009% |

Dataset split principle: "train" and "val" datasets only contain ID pixels, "train-aug" and "val-aug" contain the ID and pseudo-OOD pixels, and "test" set contains both ID and true OOD pixels. For models without pseudo-OOD exposure during training, we use the "train" and "val" datasets. For models with pseudo-OOD exposure, we utilize the "train-aug" and "val-aug" datasets. All models are evaluated on the "test" dataset. When calculating pure segmentation and misclassification performance, we mask out the OOD pixels to focus solely on the ID pixels.

**CARLA Data generation process** We first introduce key concepts for dataset generation. These are listed from the bottom up in terms of scale to make everything easy to understand. Then we introduce the main control loop.

- Tick: A tick is a single unit of simulation in the simulator. Every tick, the position of vehicles, states of sensors, weather conditions, and traffic lights are updated. A tick is the smallest denomination that we use.

- Frame: A frame is a single sample of data. Each frame consists of 6 RGB images and a single BEV semantic segmentation image. The RGB images have a resolution of 224x480, and the BEV image has a resolution of 200x200. The BEV image covers an area of 100 meters x 100 meters.

- Scene: We define a scene as a time-frame of N ticks on a specific map. In the beginning of the scene, the map is loaded. Then, 5 vehicles with sensors attached (ego vehicles) are spawned at random locations. These vehicles have 6 RGB cameras attached to them at 60 degree intervals These intervals represent front, front left, front right, back, back let, and back right. We also gather the depth ground truth maps at the same intervals. We place a birds-eye-view segmentation camera above the vehicle that captures an area of 100x100m at a resolution of 200x200 pixels. Then, 50 non-ego vehicles are spawned in random locations. Then, if needed, 40 OOD objects are spawned in random positions. The simulator will run through each of the N ticks one by one in a sequential manner, and will save one frame for each ego vehicle every five ticks.

- Main control loop: The main control loop runs through $M$ scenes, systematically varying weather conditions and urban layouts to generate a diverse dataset.

  - **Weathers Condition**: ClearNoon, CloudyNoon, WetNoon, MidRainyNoon, Soft-RainyNoon, ClearSunset, CloudySunset, WetSunset, WetCloudSunset, SoftRainySunset

  - **Urban Layouts**: Town10, Town07, Town05, Town03, Town02. Specifically, we have
    * Town 02: A small simple town with a mixture of residential and commercial buildings.
    * Town 05: Squared-grid town with cross junctions and a bridge. It has multiple lanes per direction. Useful to perform lane changes.
    * Town 07: A rural environment with narrow roads, corn, barns and hardly any traffic lights.
    * Town 10: A downtown urban environment with skyscrapers, residential buildings and an ocean promenade.

  Within each scene:

  - Weather Variation: Every $N/10$ ticks, the weather changes to the next condition in the set. This approach ensures that each weather condition is represented equally throughout the scene.

  - Urban Layout Selection: The town for each scene is selected in a cyclic manner using the following logic: `town = towns[i]; i = (i + 1) mod 5`. where `towns` is the list of urban layouts, and $i$ is the index that cycles through the towns. The modulo

operation ensures that after reaching the last town, the index wraps around to the first, providing a continuous loop through the available urban layouts.

After each scene: All objects are destroyed, and the town environment is cleared. A new scene begins by loading the next town as determined by the selection process. The number of frames can be calculated with $(N/5) * M * (\text{No. of ego vehicles})$.

## A.5 ADDITIONAL EXPERIMENTS

In this section, we first provide clear definitions and calculations for the evaluation metrics used in our study, ensuring transparency and reproducibility. Then we introduce the additional experimental results.

**Pure segmentation via IoU**: Intersection over Union (IoU) is used to evaluate the segmentation performance of the models. It is calculated as the IOU for positive class, which is defined as:

$$\text{IoU} = \frac{\text{True Positives}}{\text{True Positives} + \text{False Positives} + \text{False Negatives}}$$

Higher IoU values indicate better segmentation performance, as they reflect accurate predictions for both the object and background regions.

**Calibration via ECE**: Expected Calibration Error (ECE) measures how well the predicted probabilities align with the true likelihood of correctness. It is computed by dividing the confidence scores into $M$ bins and calculating the weighted average of the difference between accuracy and confidence for each bin:

$$\text{ECE} = \sum_{m=1}^{M} \frac{|B_m|}{n} |\text{acc}(B_m) - \text{conf}(B_m)|$$

where $B_m$ represents the set of predictions in bin $m$, $|B_m|$ is the number of samples in the bin, $n$ is the total number of samples, $\text{acc}(B_m)$ is the accuracy, and $\text{conf}(B_m)$ is the average confidence. Lower ECE values indicate better calibration. We use $M = 10$ for experiments.

**Misclassification detection**: To evaluate misclassification detection, we treat misclassified pixels as the positive class and do the binary classification task with the aleatoric uncertainty as the score. Metrics such as Area Under the Receiver Operating Characteristic curve (AUROC) and Area Under the Precision-Recall curve (AUPR) are used.

**OOD detection**: For out-of-distribution (OOD) detection, we assess the model's ability to differentiate OOD pixels from in-distribution (ID) pixels. Similar to misclassification detection, AUROC and AUPR are used as evaluation metrics, and OOD pixels are positive classes with epistemic uncertainty as the score.

### A.5.1 FULL EVALUATION ON LYFT

**Results on Lyft dataset.** Table 13 presents the comprehensive results on the Lyft (Kesten et al., 2019) dataset using LSS and CVT as model backbones. Our findings on Lyft align with those from our studies on CARLA and nuScenes. Overall, our proposed model employing the UFCE loss consistently outperforms the UCE loss in terms of semantic segmentation accuracy, calibration, and misclassification detection. Furthermore, incorporating epistemic uncertainty scaling and pseudo-OOD exposure significantly enhances OOD detection performance.

### A.5.2 QUANTITATIVE EVALUATION ON CALRA (FULL)

**Result on calibration/misclassification detection on CARLA**: Table 14 presents the segmentation, calibration, and misclassification detection results on the CARLA dataset using LSS and CVT as model backbones. We observe that our proposed model demonstrates the best OOD detection performance, the second-best calibration performance, and comparable results in segmentation and misclassification detection compared to other models.

### A.5.3 QUANTITATIVE EVALUATION ON NUSCENES (ROAD SEGMENTATION)

**Result on OOD detection on nuScenes dataset for road segmentation**: Table 15 presents the segmentation and calibration performance on the nuScenes dataset, with "road" designated as the

Table 13: Evaluation on Lyft dataset for vehicle segmentation . Best and Runner-up results are highlighted in red and blue.

| Baseline | Loss | LSS | | | | | | CVT | | | | | |
|---|---|---|---|---|---|---|---|---|---|---|---|---|---|
| | | Pure Classification | | Misclassification | | OOD Detection | | Pure Classification | | Misclassification | | OOD Detection | |
| | | IoU ↑ | ECE↓ | AUROC ↑ | AUPR ↑ | AUROC ↑ | AUPR ↑ | IoU ↑ | ECE↓ | AUROC ↑ | AUPR ↑ | AUROC ↑ | AUPR ↑ |
| | | Without pseudo OOD | | | | | | | | | | | |
| Entropy | CE | 0.393 | 0.01530 | 0.876 | 0.316 | 0.555 | 0.00124 | 0.322 | 0.00619 | 0.934 | 0.311 | 0.795 | 0.00259 |
| | Focal | 0.422 | 0.00831 | 0.928 | 0.341 | 0.638 | 0.00147 | 0.329 | 0.00630 | 0.937 | 0.317 | 0.752 | 0.00216 |
| Energy | CE | 0.393 | 0.01530 | 0.876 | 0.316 | 0.594 | 0.00127 | 0.322 | 0.00619 | 0.934 | 0.311 | 0.789 | 0.00266 |
| | Focal | 0.422 | 0.00831 | 0.928 | 0.342 | 0.629 | 0.00133 | 0.329 | 0.00630 | 0.937 | 0.317 | 0.729 | 0.00207 |
| Ensemble | CE | 0.411 | 0.00993 | 0.883 | 0.317 | 0.405 | 0.00057 | 0.351 | 0.00402 | 0.948 | 0.314 | 0.399 | 0.00058 |
| | Focal | 0.446 | 0.00406 | 0.937 | 0.334 | 0.466 | 0.00071 | 0.396 | 0.00335 | 0.957 | 0.334 | 0.535 | 0.00093 |
| Dropout | CE | 0.381 | 0.01470 | 0.870 | 0.317 | 0.295 | 0.00047 | 0.322 | 0.00735 | 0.932 | 0.314 | 0.362 | 0.00056 |
| | Focal | 0.413 | 0.00776 | 0.925 | 0.340 | 0.308 | 0.00048 | 0.318 | 0.00768 | 0.933 | 0.317 | 0.373 | 0.00057 |
| ENN | UCE | 0.396 | 0.00585 | 0.762 | 0.239 | 0.447 | 0.00082 | 0.341 | 0.00997 | 0.879 | 0.283 | 0.717 | 0.00235 |
| | UFCE | 0.427 | 0.00576 | 0.847 | 0.304 | 0.485 | 0.00089 | 0.383 | 0.00344 | 0.914 | 0.332 | 0.665 | 0.00213 |
| | | With pseudo OOD | | | | | | | | | | | |
| Energy | CE | 0.442 | 0.01210 | 0.950 | 0.341 | 0.724 | 0.033 | 0.385 | 0.00430 | 0.957 | 0.335 | 0.802 | 0.019 |
| | Focal | 0.456 | 0.00449 | 0.962 | 0.355 | 0.757 | 0.042 | 0.429 | 0.00516 | 0.964 | 0.349 | 0.761 | 0.024 |
| ENN | UCE | 0.439 | 0.00650 | 0.820 | 0.286 | 0.826 | 0.149 | 0.384 | 0.00688 | 0.905 | 0.306 | 0.913 | 0.060 |
| | Ours | 0.467 | 0.00720 | 0.820 | 0.279 | 0.826 | 0.184 | 0.419 | 0.00088 | 0.934 | 0.340 | 0.936 | 0.145 |

**Observations:** 1. The proposed UFCE loss for the ENN model consistently outperforms the commonly used UCE loss across 20/24 metrics. 2. Our proposed model (last row) shows the best OOD detection performance across LSS and CVT backbone, the specifically on AUPR metric.

Table 14: Calibration and Misclassification detection performance on the CARLA dataset for vehicle segmentation . Best and Runner-up results are highlighted in red and blue.

| Model | Loss | LSS | | | | | CVT | | | | |
|---|---|---|---|---|---|---|---|---|---|---|---|
| | | Pure Classification | | Misclassification | | | Pure Classification | | Misclassification | | |
| | | IoU ↑ | ECE↓ | AUROC ↑ | AUPR ↑ | FPR95 ↓ | IoU ↑ | ECE↓ | AUROC ↑ | AUPR ↑ | FPR95 ↓ |
| | | Without pseudo OOD | | | | | | | | | |
| Entropy | CE | 0.403 | 0.00309 | 0.928 | 0.272 | 0.222 | 0.361 | 0.00183 | 0.952 | 0.246 | 0.215 |
| | Focal | 0.403 | 0.00309 | 0.928 | 0.272 | 0.221 | 0.435 | 0.00146 | 0.970 | 0.265 | 0.148 |
| Energy | CE | 0.403 | 0.00309 | 0.928 | 0.272 | 0.222 | 0.361 | 0.00183 | 0.952 | 0.246 | 0.215 |
| | Focal | 0.403 | 0.00309 | 0.928 | 0.272 | 0.221 | 0.435 | 0.00146 | 0.970 | 0.265 | 0.148 |
| Ensemble | CE | 0.433 | 0.00202 | 0.941 | 0.266 | 0.215 | 0.410 | 0.00160 | 0.961 | 0.222 | 0.227 |
| | Focal | 0.462 | 0.00140 | 0.971 | 0.270 | 0.157 | 0.471 | 0.00135 | 0.973 | 0.260 | 0.154 |
| Dropout | CE | 0.409 | 0.00277 | 0.922 | 0.269 | 0.243 | 0.366 | 0.00183 | 0.944 | 0.238 | 0.235 |
| | Focal | 0.422 | 0.00135 | 0.956 | 0.293 | 0.167 | 0.442 | 0.00142 | 0.966 | 0.264 | 0.167 |
| ENN | UCE | 0.407 | 0.00212 | 0.817 | 0.252 | 0.345 | 0.398 | 0.00098 | 0.917 | 0.253 | 0.192 |
| | UFCE | 0.424 | 0.00033 | 0.913 | 0.283 | 0.153 | 0.422 | 0.00103 | 0.933 | 0.263 | 0.154 |
| | | With pseudo OOD | | | | | | | | | |
| Energy | CE | 0.441 | 0.00249 | 0.975 | 0.314 | 0.110 | 0.425 | 0.00124 | 0.977 | 0.264 | 0.119 |
| | Focal | 0.469 | 0.00588 | 0.979 | 0.277 | 0.094 | 0.465 | 0.00096 | 0.980 | 0.266 | 0.090 |
| ENN | UCE | 0.428 | 0.00131 | 0.894 | 0.300 | 0.193 | 0.432 | 0.00457 | 0.961 | 0.263 | 0.133 |
| | Ours | 0.459 | 0.00040 | 0.950 | 0.303 | 0.099 | 0.468 | 0.00180 | 0.967 | 0.294 | 0.097 |

**Observations:** 1. Involving pseudo-OOD in the training phase does not impact pure segmentation, calibration and misclassification detection. 2. Without pseudo-OOD, the proposed UFCE loss for the ENN model consistently outperforms the commonly used UCE loss across 9 out of 10 metrics. 3. No single model consistently performs best across all metrics, but Focal consistently performs better than CE.

positive class for segmentation. To verify the calibration improvements brought by the proposed UFocal loss, all evaluations are conducted on the clean validation dataset. We observe that our proposed UFocal loss consistently achieves lower ECE and higher IoU scores, indicating better calibration performance and segmentation accuracy compared to the UCE loss. Additionally, the misclassification detection performance is also superior to that of the UCE loss. Compared to other baselines, our model demonstrates top-tier performance.

A.5.4    QUANTITATIVE EVALUATION ON SIMPLE-BEV ON NUSCENES

**Full Results with SimpleBEV backbone on nuScenes dataset for vehicle segmentation**. Table 16 presents the full results with the SimpleBEV (Harley et al., 2023) as the backbone. While the original

Table 15: Segmentation, Calibration and Misclassification detection on road segmentation for nuScenes. Best and Runner-up results are highlighted in red and blue.

| Model | Loss | LSS | | | | CVT | | | |
|---|---|---|---|---|---|---|---|---|---|
| | | Pure Classification | | Misclassification | | Pure Classification | | Misclassification | |
| | | Road IoU↑ | ECE↓ | AUROC↑ | AUPR↑ | Road IoU↑ | ECE↓ | AUROC↑ | AUPR↑ |
| Entropy | CE | 0.756 | 0.0448 | 0.870 | 0.330 | 0.637 | 0.0495 | 0.835 | 0.354 |
| | Focal | 0.763 | 0.0266 | 0.886 | 0.341 | 0.678 | 0.0458 | 0.848 | 0.345 |
| Ensemble | CE | 0.776 | 0.0156 | 0.883 | 0.309 | 0.656 | 0.0269 | 0.840 | 0.324 |
| | Focal | 0.769 | 0.0308 | 0.882 | 0.314 | 0.698 | 0.0263 | 0.853 | 0.324 |
| Dropout | CE | 0.744 | 0.0455 | 0.867 | 0.329 | 0.624 | 0.0573 | 0.831 | 0.351 |
| | Focal | 0.752 | 0.0300 | 0.883 | 0.337 | 0.670 | 0.0511 | 0.844 | 0.343 |
| ENN | UCE | 0.752 | 0.0417 | 0.783 | 0.291 | 0.639 | 0.0455 | 0.806 | 0.335 |
| | UFCE | 0.760 | 0.0250 | 0.839 | 0.319 | 0.679 | 0.0438 | 0.828 | 0.336 |

**Observations:** 1. The proposed UFCE loss for the ENN model consistently outperforms the commonly used UCE loss across all metrics, showing improvements of 2.6% in segmentation IoU, 0.9 in calibration ECE, 3.9% in misclassification AUROC, and 1.5% in AUPR across LSS and CVT backbone. 2. For road detection, there is no clear evidence showing that Focal loss performs better than classic CE loss in terms of segmentation or aleatoric uncertainty prediction.

paper reported a vehicle segmentation IoU of 44.7 on nuScenes, we report 38.2. This discrepancy arises because we adopt a consistent experimental setup across all backbones, including LSS and CVT. Specifically, we exclude frames containing true OOD objects from the training set and use camera images with a resolution of $224 \times 480$ without any image augmentations. Additionally, we employ bilinear sampling as the lifting strategy, EfficientNet-B4 as the network backbone, and batch size of 16.

We observe that (1).The proposed model achieves the second-best performance in pure segmentation, with a gap of only 0.1% from the best baseline, while outperforming all models in calibration. (2). All models demonstrate comparable performance in misclassification detection. (3). For OOD detection, the proposed model achieves the best performance in AUROC and AUPR, with a 12% improvement in AUPR over the second-best model and FPR95 is only 0.008 higher than the best baseline.

Table 16: Simple-BEV backbone: Segmentation, Calibration, Misclassification detection, OOD detection performance for vehicle segmentation on nuScenes . Best and Runner-up results are highlighted in red and blue.

| model | loss | Pure Classification | | Misclassification | | | OOD Detection | | |
|---|---|---|---|---|---|---|---|---|---|
| | | IoU ↑ | ECE↓ | AUROC↑ | AUPR↑ | FPR95↓ | AUROC↑ | AUPR↑ | FPR95↓ |
| | | Without pseudo OOD | | | | | | | |
| Entropy | CE | 0.352 | 0.00721 | 0.936 | 0.326 | 0.196 | 0.684 | 0.001 | 0.816 |
| | Focal | 0.366 | 0.00697 | 0.942 | 0.322 | 0.193 | 0.661 | 0.001 | 0.841 |
| Energy | CE | 0.352 | 0.00721 | 0.936 | 0.326 | 0.196 | 0.638 | 0.000 | 0.819 |
| | Focal | 0.366 | 0.00698 | 0.942 | 0.322 | 0.193 | 0.614 | 0.000 | 0.835 |
| Ensemble | CE | 0.373 | 0.00456 | 0.946 | 0.322 | 0.197 | 0.534 | 0.000 | 0.920 |
| | Focal | 0.388 | 0.00453 | 0.946 | 0.322 | 0.185 | 0.504 | 0.000 | 0.937 |
| Dropout | CE | 0.350 | 0.00617 | 0.934 | 0.325 | 0.207 | 0.458 | 0.000 | 0.964 |
| | Focal | 0.364 | 0.00623 | 0.940 | 0.323 | 0.200 | 0.523 | 0.000 | 0.931 |
| ENN | UCE | 0.361 | 0.00647 | 0.844 | 0.285 | 0.286 | 0.657 | 0.000 | 0.832 |
| | UFCE | 0.370 | 0.00296 | 0.848 | 0.294 | 0.279 | 0.604 | 0.000 | 0.831 |
| | | With pseudo OOD | | | | | | | |
| Energy | CE | 0.360 | 0.00586 | 0.946 | 0.332 | 0.183 | 0.820 | 0.071 | 0.326 |
| | Focal | 0.372 | 0.00606 | 0.947 | 0.332 | 0.186 | 0.838 | 0.058 | 0.325 |
| ENN | UCE | 0.364 | 0.00455 | 0.882 | 0.314 | 0.207 | 0.914 | 0.215 | 0.272 |
| | Ours | 0.372 | 0.00315 | 0.896 | 0.322 | 0.187 | 0.845 | 0.319 | 0.356 |

A.5.5 ROBUTNESS ON SELECTION OF PSEUDO-OOD

**Robustness to the selection of pseudo-OOD (cont.)**. We investigate how the similarity between pseudo-OOD and true OOD affects epistemic uncertainty predictions. For this, we use two pseudo-OOD and true-OOD pairs for nuScenes and CARLA, with detailed settings presented in Table 11. Generalization results for nuScenes are shown in Table 4, with the main discussion provided in Section 4.2. Generalization results for CARLA are presented in Table 17, where we evaluate OOD detection performance using "kangaroo" as the true OOD and "bears, horses, cows, elephants" as pseudo-OODs. For this true/pseudo-OOD pair, our proposed framework consistently outperforms others across all six metrics, achieving an average AUPR improvement of 3.2% over the runner-up, sharing the same observation on nuScenes.

Table 17: Robustness Analysis (selection of pseudo-OOD): OOD detection performance for vehicle segmentation on CARLA ("kangaroo" as true OOD). Best and Runner-up results are highlighted in red and blue.

| Model | Loss | LSS | | | CVT | | |
|---|---|---|---|---|---|---|---|
| | | AUROC ↑ | AUPR ↑ | FPR95 ↓ | AUROC ↑ | AUPR ↑ | FPR95 ↓ |
| Without pseudo OOD | | | | | | | |
| Entropy | CE | 0.665 | 0.005 | 0.768 | 0.767 | 0.007 | 0.691 |
| | Focal | 0.725 | 0.006 | 0.727 | 0.783 | 0.007 | 0.675 |
| Energy | CE | 0.672 | 0.005 | 0.742 | 0.749 | 0.006 | 0.69 |
| | Focal | 0.717 | 0.006 | 0.726 | 0.755 | 0.007 | 0.675 |
| Ensemble | CE | 0.485 | 0.001 | 0.963 | 0.487 | 0.001 | 0.962 |
| | Focal | 0.449 | 0.001 | 0.955 | 0.504 | 0.001 | 0.951 |
| Dropout | CE | 0.432 | 0.001 | 0.966 | 0.388 | 0.001 | 0.962 |
| | Focal | 0.401 | 0.001 | 0.967 | 0.342 | 0.001 | 0.974 |
| ENN | UCE | 0.623 | 0.004 | 0.779 | 0.68 | 0.005 | 0.784 |
| | UFCE | 0.593 | 0.004 | 0.794 | 0.727 | 0.006 | 0.712 |
| With pseudo OOD | | | | | | | |
| Energy | CE | 0.746 | 0.045 | 0.556 | 0.746 | 0.034 | 0.442 |
| | Focal | 0.727 | 0.049 | 0.489 | 0.786 | 0.051 | 0.425 |
| ENN | UCE | 0.818 | 0.097 | 0.417 | 0.911 | 0.077 | 0.353 |
| | Ours | 0.882 | 0.111 | 0.368 | 0.914 | 0.127 | 0.322 |

Table 18 presents the OOD detection performance across various pseudo/true OOD pairs. The results support our intuition that higher similarity between true and pseudo-OOD pairs leads to improved OOD detection performance, suggesting that identifying representative pseudo-OODs is a promising direction for advancing OOD detection tasks. Notably, our proposed model consistently outperforms the best baseline methods across all eight scenarios, achieving improvements of up to 12.6% in AUPR. This highlights the robustness of our approach, even in settings with less similar OOD pairs.

Table 18: OOD detection performance (AUPR ↑) for similar and dissimilar OOD pairs across datasets and backbones, comparing our model to the best baselines and its ablated variants.

| Dataset | Similarity between True/Pseudo OOD | LSS | | CVT | |
|---|---|---|---|---|---|
| | | Best Baseline | Ours | Best Baseline | Ours |
| nuScenes | Similar | 31.5 | 33.5 | 21.2 | 26.9 |
| | Dissimilar | 20.8 | 21.5 | 11.7 | 15.3 |
| CARLA | Similar | 14.7 | 20.4 | 11.1 | 23.7 |
| | Dissimilar | 9.7 | 11.0 | 7.7 | 12.7 |

A.5.6 ROBUTNESS ON CORRUPTED DATASET

**Robustness on nuScenes-C**. nuScenes-C(Xie et al., 2024) introduces various corruptions to the validation set of the nuScenes dataset, comprising eight types of corruption, each with three levels of severity: easy, mid, and hard. *Brightness, Dark, Fog, and Snow* represent external environmental

dynamics, such as illumination changes or extreme weather conditions. *Motion Blur* and *Color Quant* simulate effects caused by high-speed motion and image quantization, respectively. *Camera Crash* and *Frame Lost* model camera malfunctions.

Using a model trained on the clean nuScenes training dataset (excluding frames containing true OOD pixels), we evaluate its performance on the corrupted validation set to assess its robustness to domain shifts. We compare our model against the most relevant baseline, "ENN-UCE," and the model that generally performs best in OOD detection tasks, "Energy-Focal". The results are presented in Table 19.

For misclassification detection, our model achieves the highest AUROC in 14 out of 21 scenarios and the second highest in the remaining 7 scenarios. Our model archives the highest AUPR in 13 out of 21 scenarios and the second highest in the remaining 8 scenarios. Our model consistently outperforms the ENN-UCE baseline across all 21 scenarios on both AUROC and AUPR. For OOD detection, our model achieves the best AUPR in 16 out of 21 scenarios and the second-best AUPR for the other 5 scenarios. Our model archives the best AUROC in 8 out of 21 scenarios and second-best in the remaining 13 scenarios. We note that AUROC and AUPR offer different perspectives to measure the quality of a ranking on data points (BEV pixels in our context) for separating positives and negatives. A relatively lower AUROC but higher AUPR for our method in some scenarios suggests our model identifies more true positives among top-ranked pixels than the ENN-UCE model, while ENN-UCE model better separates true positives and negatives among lower-ranked BEV pixels. It indicates that data corruptions in these scenarios may affect the quality of epistemic uncertainty quantification for some lowly ranked pixels, but not for the top ranked ones. However, high AUPR is particularly important in applications where human experts manually verify top-ranked misclassified instances or anomalies. Given the high cost of manual verification, ensuring a high rate of true positives among the top-ranked data points makes AUPR a more suitable metric in such cases.

We observe a greater performance drop under domain shifts with higher corruption severity, consistent with the findings in Xie et al. (2024), where significant performance degradation was noted in the segmentation task using CVT on nuScenes compared to the clean dataset. Additionally, sensor-driven distortions, such as color quantization and motion blur, have the least impact, while snow and camera crashes cause the most severe performance degradation. These experiments highlight the need for robust models or optimization strategies, which we plan to explore as a future direction.

### A.5.7    ROBUTNESS ON DIVERSE CONDITIONS (CARLA)

We evaluate our models on diverse subsets of the test sets, considering variations in urban layouts and weather conditions. In the CARLA dataset, we include four towns across 10 weather conditions, with details on dataset generation provided in Appendix A.4.

Evaluation results on different towns are shown in Table 20. Towns 2 and 5 represent denser urban layouts compared to Towns 7 and 10. We observe consistent performance across each subset, aligning with the evaluations conducted on the full CARLA dataset. Specifically, our model consistently outperforms all baselines in OOD detection tasks across all towns, while also achieving better segmentation, calibration, and misclassification detection compared to the standard UCE loss. Furthermore, we observe performance gaps across urban layouts, highlighting the need to improve model robustness in diverse environments.

Evaluation results on varying weather conditions are shown in table 21, 22, 23. While the results align with the conclusions drawn from evaluations on the full dataset when comparing our proposed model to the baselines, we also observe performance variations across different weather conditions. Interestingly, the results challenge the intuition that the best performance would occur under "ClearNoon" conditions. Notably, OOD detection performance is positively correlated with segmentation performance, which may be attributed to clearer environments improving segmentation quality and, consequently, OOD detection.

### A.5.8    ROBUTNESS ON MODEL INITIALIZATIONS

We report the model variance for our proposed model in Table 24 and the evidential UCE baseline. We randomly initialize the model three times and the variance is within 3%.

Table 19: Robustness Analysis (data corruption): evaluation on estimated uncertainty with CVT as the backbone on nuScenes-C.

| Corruption Type | Severity | Model | Misclassification | | | OOD Detection | | |
|---|---|---|---|---|---|---|---|---|
| | | | AUROC ↑ | AUPR ↑ | FPR95 ↓ | AUROC ↑ | AUPR ↑ | FPR95 ↓ |
| None | Clean | Energy-Focal | 0.955 | 0.321 | 0.196 | 0.860 | 0.024 | 0.319 |
| | | ENN-UCE | 0.919 | 0.313 | 0.227 | 0.921 | 0.212 | 0.306 |
| | | Ours | 0.934 | 0.321 | 0.196 | 0.928 | 0.269 | 0.244 |
| CameraCrash | Easy | Energy-Focal | 0.863 | 0.226 | 0.436 | 0.710 | 0.016 | 0.581 |
| | | ENN-UCE | 0.823 | 0.195 | 0.478 | 0.859 | 0.052 | 0.459 |
| | | Ours | 0.878 | 0.318 | 0.365 | 0.787 | 0.109 | 0.525 |
| | Mid | Energy-Focal | 0.785 | 0.140 | 0.644 | 0.615 | 0.011 | 0.742 |
| | | ENN-UCE | 0.727 | 0.117 | 0.659 | 0.723 | 0.008 | 0.734 |
| | | Ours | 0.867 | 0.351 | 0.538 | 0.651 | 0.067 | 0.700 |
| | Hard | Energy-Focal | 0.796 | 0.151 | 0.605 | 0.623 | 0.004 | 0.809 |
| | | ENN-UCE | 0.751 | 0.140 | 0.611 | 0.791 | 0.003 | 0.756 |
| | | Ours | 0.813 | 0.251 | 0.440 | 0.663 | 0.027 | 0.773 |
| FrameLost | Easy | Energy-Focal | 0.880 | 0.235 | 0.411 | 0.712 | 0.021 | 0.536 |
| | | ENN-UCE | 0.846 | 0.227 | 0.423 | 0.862 | 0.118 | 0.483 |
| | | Ours | 0.880 | 0.291 | 0.364 | 0.805 | 0.148 | 0.533 |
| | Mid | Energy-Focal | 0.783 | 0.132 | 0.654 | 0.516 | 0.003 | 0.792 |
| | | ENN-UCE | 0.727 | 0.117 | 0.673 | 0.743 | 0.044 | 0.790 |
| | | Ours | 0.848 | 0.290 | 0.503 | 0.605 | 0.038 | 0.843 |
| | Hard | Energy-Focal | 0.728 | 0.112 | 0.723 | 0.499 | 0.002 | 0.871 |
| | | ENN-UCE | 0.657 | 0.074 | 0.785 | 0.674 | 0.032 | 0.799 |
| | | Ours | 0.826 | 0.280 | 0.572 | 0.559 | 0.024 | 0.898 |
| ColorQuant | Easy | Energy-Focal | 0.949 | 0.315 | 0.209 | 0.850 | 0.025 | 0.354 |
| | | ENN-UCE | 0.905 | 0.307 | 0.246 | 0.926 | 0.174 | 0.301 |
| | | Ours | 0.925 | 0.310 | 0.219 | 0.923 | 0.243 | 0.275 |
| | Mid | Energy-Focal | 0.929 | 0.297 | 0.263 | 0.825 | 0.028 | 0.407 |
| | | ENN-UCE | 0.866 | 0.273 | 0.302 | 0.915 | 0.100 | 0.322 |
| | | Ours | 0.902 | 0.287 | 0.289 | 0.889 | 0.144 | 0.367 |
| | Hard | Energy-Focal | 0.866 | 0.223 | 0.436 | 0.777 | 0.025 | 0.579 |
| | | ENN-UCE | 0.775 | 0.197 | 0.462 | 0.843 | 0.008 | 0.507 |
| | | Ours | 0.831 | 0.227 | 0.445 | 0.866 | 0.008 | 0.431 |
| MotionBlur | Easy | Energy-Focal | 0.947 | 0.309 | 0.229 | 0.831 | 0.024 | 0.357 |
| | | ENN-UCE | 0.899 | 0.284 | 0.279 | 0.921 | 0.194 | 0.384 |
| | | Ours | 0.934 | 0.307 | 0.228 | 0.931 | 0.244 | 0.266 |
| | Mid | Energy-Focal | 0.900 | 0.249 | 0.374 | 0.748 | 0.020 | 0.573 |
| | | ENN-UCE | 0.820 | 0.225 | 0.387 | 0.853 | 0.149 | 0.443 |
| | | Ours | 0.898 | 0.260 | 0.342 | 0.857 | 0.155 | 0.417 |
| | Hard | Energy-Focal | 0.864 | 0.214 | 0.443 | 0.739 | 0.020 | 0.616 |
| | | ENN-UCE | 0.772 | 0.195 | 0.451 | 0.836 | 0.138 | 0.494 |
| | | Ours | 0.867 | 0.235 | 0.402 | 0.833 | 0.131 | 0.463 |
| Brightness | Easy | Energy-Focal | 0.939 | 0.300 | 0.253 | 0.875 | 0.049 | 0.314 |
| | | ENN-UCE | 0.881 | 0.278 | 0.278 | 0.915 | 0.176 | 0.346 |
| | | Ours | 0.911 | 0.288 | 0.254 | 0.898 | 0.172 | 0.329 |
| | Mid | Energy-Focal | 0.909 | 0.261 | 0.342 | 0.852 | 0.038 | 0.375 |
| | | ENN-UCE | 0.833 | 0.227 | 0.379 | 0.860 | 0.057 | 0.434 |
| | | Ours | 0.881 | 0.248 | 0.360 | 0.883 | 0.075 | 0.324 |
| | Hard | Energy-Focal | 0.887 | 0.239 | 0.400 | 0.803 | 0.008 | 0.461 |
| | | ENN-UCE | 0.792 | 0.202 | 0.448 | 0.792 | 0.010 | 0.494 |
| | | Ours | 0.862 | 0.221 | 0.442 | 0.870 | 0.037 | 0.420 |
| Snow | Easy | Energy-Focal | 0.890 | 0.233 | 0.387 | 0.809 | 0.017 | 0.619 |
| | | ENN-UCE | 0.741 | 0.183 | 0.482 | 0.750 | 0.057 | 0.600 |
| | | Ours | 0.841 | 0.237 | 0.384 | 0.830 | 0.104 | 0.480 |
| | Mid | Energy-Focal | 0.839 | 0.169 | 0.523 | 0.785 | 0.006 | 0.636 |
| | | ENN-UCE | 0.613 | 0.080 | 0.732 | 0.679 | 0.004 | 0.762 |
| | | Ours | 0.798 | 0.176 | 0.512 | 0.724 | 0.053 | 0.626 |
| | Hard | Energy-Focal | 0.826 | 0.158 | 0.547 | 0.792 | 0.005 | 0.685 |
| | | ENN-UCE | 0.580 | 0.063 | 0.797 | 0.604 | 0.001 | 0.809 |
| | | Ours | 0.754 | 0.137 | 0.604 | 0.676 | 0.067 | 0.709 |
| Fog | Easy | Energy-Focal | 0.920 | 0.280 | 0.310 | 0.873 | 0.029 | 0.341 |
| | | ENN-UCE | 0.857 | 0.249 | 0.352 | 0.840 | 0.134 | 0.485 |
| | | Ours | 0.906 | 0.271 | 0.318 | 0.880 | 0.173 | 0.353 |
| | Mid | Energy-Focal | 0.899 | 0.250 | 0.376 | 0.849 | 0.019 | 0.416 |
| | | ENN-UCE | 0.858 | 0.248 | 0.372 | 0.829 | 0.114 | 0.498 |
| | | Ours | 0.892 | 0.251 | 0.368 | 0.856 | 0.160 | 0.426 |
| | Hard | Energy-Focal | 0.878 | 0.227 | 0.426 | 0.803 | 0.014 | 0.427 |
| | | ENN-UCE | 0.851 | 0.239 | 0.395 | 0.838 | 0.061 | 0.500 |
| | | Ours | 0.890 | 0.241 | 0.388 | 0.823 | 0.090 | 0.432 |

Table 20: Robustness Analysis (urban layouts): evaluation with LSS backbone on CARLA in diverse towns. Best results are highlighted in red.

| pseudo OOD | model | loss | Pure Classification | | Misclassification | | | OOD Detection | | |
|---|---|---|---|---|---|---|---|---|---|---|
| | | | IoU↑ | ECE↓ | AUROC↑ | AUPR↑ | FPR95↓ | AUROC↑ | AUPR↑ | FPR95↓ |
| | | | Town 10 | | | | | | | |
| No | Baseline | CE | 0.377 | 0.00331 | 0.931 | 0.270 | 0.228 | 0.688 | 0.003 | 0.805 |
| | | Focal | 0.392 | 0.00140 | 0.956 | 0.291 | 0.177 | 0.755 | 0.003 | 0.749 |
| | Energy | CE | 0.377 | 0.00331 | 0.931 | 0.270 | 0.228 | 0.667 | 0.002 | 0.784 |
| | | Focal | 0.392 | 0.00140 | 0.956 | 0.291 | 0.177 | 0.748 | 0.003 | 0.754 |
| | Ensemble | CE | 0.409 | 0.00218 | 0.942 | 0.257 | 0.228 | 0.491 | 0.001 | 0.963 |
| | | Focal | 0.431 | 0.00165 | 0.964 | 0.262 | 0.194 | 0.453 | 0.001 | 0.961 |
| | Dropout | CE | 0.381 | 0.00299 | 0.921 | 0.264 | 0.252 | 0.462 | 0.001 | 0.960 |
| | | Focal | 0.398 | 0.00151 | 0.941 | 0.276 | 0.213 | 0.391 | 0.001 | 0.967 |
| | Evidential | UCE | 0.384 | 0.00231 | 0.813 | 0.245 | 0.352 | 0.604 | 0.002 | 0.816 |
| | | UFCE | 0.390 | 0.00031 | 0.905 | 0.274 | 0.167 | 0.578 | 0.003 | 0.820 |
| Yes | Energy | CE | 0.413 | 0.00263 | 0.969 | 0.297 | 0.141 | 0.857 | 0.058 | 0.358 |
| | | Focal | 0.427 | 0.00586 | 0.976 | 0.278 | 0.111 | 0.861 | 0.061 | 0.284 |
| | Evidential | UCE | 0.403 | 0.00141 | 0.876 | 0.276 | 0.228 | 0.866 | 0.106 | 0.338 |
| | | Ours | 0.427 | 0.00064 | 0.939 | 0.286 | 0.133 | 0.946 | 0.180 | 0.235 |
| | | | Town 5 | | | | | | | |
| No | Baseline | CE | 0.407 | 0.00329 | 0.930 | 0.289 | 0.208 | 0.677 | 0.003 | 0.784 |
| | | Focal | 0.414 | 0.00158 | 0.960 | 0.303 | 0.158 | 0.765 | 0.003 | 0.729 |
| | Energy | CE | 0.407 | 0.00329 | 0.930 | 0.289 | 0.208 | 0.678 | 0.003 | 0.763 |
| | | Focal | 0.414 | 0.00158 | 0.960 | 0.303 | 0.158 | 0.766 | 0.003 | 0.719 |
| | Ensemble | CE | 0.438 | 0.00207 | 0.942 | 0.278 | 0.201 | 0.493 | 0.001 | 0.961 |
| | | Focal | 0.457 | 0.00139 | 0.969 | 0.283 | 0.158 | 0.442 | 0.001 | 0.960 |
| | Dropout | CE | 0.413 | 0.00294 | 0.919 | 0.276 | 0.235 | 0.437 | 0.001 | 0.963 |
| | | Focal | 0.420 | 0.00170 | 0.948 | 0.291 | 0.186 | 0.401 | 0.001 | 0.964 |
| | Evidential | UCE | 0.413 | 0.00229 | 0.822 | 0.256 | 0.334 | 0.630 | 0.002 | 0.798 |
| | | UFCE | 0.416 | 0.00007 | 0.916 | 0.291 | 0.147 | 0.520 | 0.002 | 0.832 |
| Yes | Energy | CE | 0.443 | 0.00267 | 0.975 | 0.315 | 0.117 | 0.878 | 0.071 | 0.298 |
| | | Focal | 0.456 | 0.00609 | 0.979 | 0.282 | 0.106 | 0.892 | 0.086 | 0.214 |
| | Evidential | UCE | 0.441 | 0.00136 | 0.893 | 0.294 | 0.196 | 0.891 | 0.144 | 0.256 |
| | | Ours | 0.455 | 0.00049 | 0.945 | 0.306 | 0.111 | 0.957 | 0.213 | 0.192 |
| | | | Town 7 | | | | | | | |
| No | Baseline | CE | 0.376 | 0.00348 | 0.935 | 0.283 | 0.206 | 0.599 | 0.001 | 0.877 |
| | | Focal | 0.393 | 0.00182 | 0.967 | 0.311 | 0.126 | 0.725 | 0.002 | 0.821 |
| | Energy | CE | 0.376 | 0.00348 | 0.935 | 0.283 | 0.206 | 0.604 | 0.001 | 0.849 |
| | | Focal | 0.393 | 0.00182 | 0.967 | 0.311 | 0.126 | 0.706 | 0.002 | 0.820 |
| | Ensemble | CE | 0.400 | 0.00224 | 0.951 | 0.276 | 0.181 | 0.482 | 0.001 | 0.953 |
| | | Focal | 0.425 | 0.00132 | 0.975 | 0.295 | 0.134 | 0.460 | 0.001 | 0.957 |
| | Dropout | CE | 0.374 | 0.00322 | 0.929 | 0.279 | 0.225 | 0.509 | 0.001 | 0.951 |
| | | Focal | 0.387 | 0.00182 | 0.960 | 0.308 | 0.149 | 0.461 | 0.001 | 0.952 |
| | Evidential | UCE | 0.384 | 0.00244 | 0.822 | 0.259 | 0.333 | 0.532 | 0.001 | 0.868 |
| | | UFCE | 0.394 | 0.00017 | 0.922 | 0.304 | 0.136 | 0.550 | 0.001 | 0.839 |
| Yes | Energy | CE | 0.418 | 0.00282 | 0.975 | 0.325 | 0.097 | 0.845 | 0.059 | 0.349 |
| | | Focal | 0.420 | 0.00478 | 0.982 | 0.311 | 0.078 | 0.888 | 0.076 | 0.236 |
| | Evidential | UCE | 0.410 | 0.00156 | 0.899 | 0.308 | 0.184 | 0.858 | 0.126 | 0.326 |
| | | Ours | 0.438 | 0.00035 | 0.957 | 0.320 | 0.079 | 0.949 | 0.230 | 0.227 |
| | | | Town 2 | | | | | | | |
| No | Baseline | CE | 0.407 | 0.00320 | 0.943 | 0.294 | 0.185 | 0.696 | 0.002 | 0.788 |
| | | Focal | 0.437 | 0.00115 | 0.971 | 0.310 | 0.114 | 0.759 | 0.002 | 0.754 |
| | Energy | CE | 0.407 | 0.00320 | 0.943 | 0.294 | 0.185 | 0.697 | 0.002 | 0.753 |
| | | Focal | 0.437 | 0.00115 | 0.971 | 0.310 | 0.114 | 0.757 | 0.002 | 0.750 |
| | Ensemble | CE | 0.445 | 0.00195 | 0.954 | 0.287 | 0.174 | 0.490 | 0.001 | 0.964 |
| | | Focal | 0.478 | 0.00125 | 0.978 | 0.292 | 0.117 | 0.461 | 0.001 | 0.954 |
| | Dropout | CE | 0.411 | 0.00291 | 0.935 | 0.287 | 0.204 | 0.436 | 0.001 | 0.967 |
| | | Focal | 0.440 | 0.00133 | 0.964 | 0.299 | 0.136 | 0.397 | 0.001 | 0.962 |
| | Evidential | UCE | 0.422 | 0.00203 | 0.842 | 0.275 | 0.297 | 0.632 | 0.002 | 0.800 |
| | | UFCE | 0.435 | 0.00029 | 0.925 | 0.306 | 0.130 | 0.540 | 0.001 | 0.870 |
| Yes | Energy | CE | 0.458 | 0.00246 | 0.979 | 0.321 | 0.101 | 0.894 | 0.057 | 0.261 |
| | | Focal | 0.473 | 0.00571 | 0.985 | 0.308 | 0.067 | 0.893 | 0.062 | 0.225 |
| | Evidential | UCE | 0.444 | 0.00114 | 0.907 | 0.309 | 0.168 | 0.896 | 0.093 | 0.271 |
| | | Ours | 0.479 | 0.00022 | 0.963 | 0.319 | 0.074 | 0.964 | 0.156 | 0.161 |

Table 21: Robustness Analysis (weather conditions - part 1): evaluation with LSS backbone on CARLA in diverse weather conditions. Best results are highlighted in red.

| pseudo OOD | model | loss | Pure Classification | | Misclassification | | | OOD Detection | | |
|---|---|---|---|---|---|---|---|---|---|---|
| | | | IoU↑ | ECE↓ | AUROC↑ | AUPR↑ | FPR95↓ | AUROC↑ | AUPR↑ | FPR95↓ |
| ClearNoon | | | | | | | | | | |
| No | Baseline | CE | 0.401 | 0.00318 | 0.927 | 0.275 | 0.217 | 0.654 | 0.002 | 0.839 |
| | | Focal | 0.420 | 0.00159 | 0.959 | 0.295 | 0.156 | 0.737 | 0.002 | 0.761 |
| | Energy | CE | 0.401 | 0.00318 | 0.927 | 0.275 | 0.217 | 0.648 | 0.002 | 0.809 |
| | | Focal | 0.420 | 0.00159 | 0.959 | 0.296 | 0.156 | 0.740 | 0.002 | 0.757 |
| | Ensemble | CE | 0.433 | 0.00207 | 0.941 | 0.261 | 0.200 | 0.493 | 0.001 | 0.958 |
| | | Focal | 0.460 | 0.00146 | 0.969 | 0.271 | 0.160 | 0.454 | 0.001 | 0.953 |
| | Dropout | CE | 0.398 | 0.00292 | 0.920 | 0.270 | 0.235 | 0.484 | 0.001 | 0.947 |
| | | Focal | 0.419 | 0.00154 | 0.952 | 0.289 | 0.178 | 0.411 | 0.001 | 0.957 |
| | Evidential | UCE | 0.411 | 0.00220 | 0.815 | 0.244 | 0.348 | 0.573 | 0.002 | 0.831 |
| | | UFCE | 0.416 | 0.00015 | 0.912 | 0.287 | 0.157 | 0.543 | 0.002 | 0.834 |
| Yes | Energy | CE | 0.449 | 0.00255 | 0.975 | 0.317 | 0.111 | 0.852 | 0.051 | 0.347 |
| | | Focal | 0.449 | 0.00493 | 0.980 | 0.299 | 0.087 | 0.862 | 0.064 | 0.278 |
| | Evidential | UCE | 0.443 | 0.00133 | 0.889 | 0.292 | 0.203 | 0.866 | 0.098 | 0.299 |
| | | Ours | 0.455 | 0.00038 | 0.950 | 0.310 | 0.091 | 0.945 | 0.159 | 0.255 |
| CloudyNoon | | | | | | | | | | |
| No | Baseline | CE | 0.434 | 0.00300 | 0.926 | 0.278 | 0.213 | 0.679 | 0.002 | 0.801 |
| | | Focal | 0.428 | 0.00136 | 0.969 | 0.307 | 0.130 | 0.731 | 0.002 | 0.760 |
| | Energy | CE | 0.387 | 0.00343 | 0.940 | 0.284 | 0.206 | 0.664 | 0.002 | 0.783 |
| | | Focal | 0.413 | 0.00133 | 0.963 | 0.300 | 0.145 | 0.731 | 0.002 | 0.764 |
| | Ensemble | CE | 0.419 | 0.00219 | 0.947 | 0.277 | 0.202 | 0.507 | 0.001 | 0.948 |
| | | Focal | 0.460 | 0.00118 | 0.975 | 0.304 | 0.132 | 0.463 | 0.001 | 0.956 |
| | Dropout | CE | 0.415 | 0.00290 | 0.933 | 0.286 | 0.220 | 0.485 | 0.001 | 0.959 |
| | | Focal | 0.417 | 0.00163 | 0.955 | 0.303 | 0.164 | 0.428 | 0.001 | 0.951 |
| | Evidential | UCE | 0.418 | 0.00222 | 0.833 | 0.273 | 0.313 | 0.587 | 0.002 | 0.836 |
| | | UFCE | 0.423 | 0.00016 | 0.923 | 0.307 | 0.136 | 0.560 | 0.002 | 0.833 |
| Yes | Energy | CE | 0.454 | 0.00256 | 0.981 | 0.331 | 0.082 | 0.913 | 0.068 | 0.231 |
| | | Focal | 0.487 | 0.00415 | 0.985 | 0.306 | 0.070 | 0.922 | 0.090 | 0.158 |
| | Evidential | UCE | 0.451 | 0.00120 | 0.913 | 0.316 | 0.158 | 0.897 | 0.124 | 0.236 |
| | | Ours | 0.471 | 0.00010 | 0.965 | 0.332 | 0.059 | 0.975 | 0.229 | 0.104 |
| WetNoon | | | | | | | | | | |
| No | Baseline | CE | 0.415 | 0.00316 | 0.940 | 0.294 | 0.189 | 0.643 | 0.002 | 0.832 |
| | | Focal | 0.418 | 0.00159 | 0.967 | 0.319 | 0.128 | 0.754 | 0.003 | 0.739 |
| | Energy | CE | 0.415 | 0.00316 | 0.940 | 0.294 | 0.189 | 0.654 | 0.002 | 0.799 |
| | | Focal | 0.418 | 0.00159 | 0.967 | 0.319 | 0.128 | 0.755 | 0.003 | 0.739 |
| | Ensemble | CE | 0.439 | 0.00199 | 0.956 | 0.300 | 0.162 | 0.501 | 0.001 | 0.954 |
| | | Focal | 0.460 | 0.00118 | 0.975 | 0.303 | 0.132 | 0.463 | 0.001 | 0.957 |
| | Dropout | CE | 0.415 | 0.00290 | 0.933 | 0.286 | 0.220 | 0.485 | 0.001 | 0.959 |
| | | Focal | 0.417 | 0.00163 | 0.955 | 0.303 | 0.164 | 0.428 | 0.001 | 0.951 |
| | Evidential | UCE | 0.418 | 0.00222 | 0.833 | 0.272 | 0.313 | 0.587 | 0.002 | 0.835 |
| | | UFCE | 0.423 | 0.00015 | 0.923 | 0.307 | 0.136 | 0.560 | 0.002 | 0.833 |
| Yes | Energy | CE | 0.454 | 0.00256 | 0.981 | 0.330 | 0.082 | 0.913 | 0.068 | 0.231 |
| | | Focal | 0.467 | 0.00549 | 0.987 | 0.308 | 0.052 | 0.911 | 0.078 | 0.177 |
| | Evidential | UCE | 0.451 | 0.00120 | 0.913 | 0.317 | 0.158 | 0.897 | 0.124 | 0.236 |
| | | Ours | 0.471 | 0.00010 | 0.965 | 0.333 | 0.059 | 0.975 | 0.229 | 0.104 |
| MidRainyNoon | | | | | | | | | | |
| No | Baseline | CE | 0.410 | 0.00319 | 0.938 | 0.302 | 0.192 | 0.657 | 0.002 | 0.838 |
| | | Focal | 0.428 | 0.00136 | 0.969 | 0.306 | 0.130 | 0.731 | 0.002 | 0.760 |
| | Energy | CE | 0.410 | 0.00319 | 0.938 | 0.302 | 0.192 | 0.672 | 0.002 | 0.798 |
| | | Focal | 0.428 | 0.00135 | 0.969 | 0.306 | 0.130 | 0.735 | 0.002 | 0.765 |
| | Ensemble | CE | 0.445 | 0.00196 | 0.953 | 0.283 | 0.173 | 0.474 | 0.001 | 0.969 |
| | | Focal | 0.466 | 0.00129 | 0.978 | 0.297 | 0.121 | 0.457 | 0.001 | 0.963 |
| | Dropout | CE | 0.418 | 0.00283 | 0.927 | 0.286 | 0.220 | 0.475 | 0.001 | 0.957 |
| | | Focal | 0.432 | 0.00142 | 0.956 | 0.296 | 0.167 | 0.442 | 0.001 | 0.955 |
| | Evidential | UCE | 0.420 | 0.00211 | 0.834 | 0.264 | 0.311 | 0.595 | 0.002 | 0.817 |
| | | UFCE | 0.429 | 0.00014 | 0.926 | 0.297 | 0.128 | 0.515 | 0.001 | 0.871 |
| Yes | Energy | CE | 0.453 | 0.00255 | 0.981 | 0.327 | 0.083 | 0.904 | 0.085 | 0.249 |
| | | Focal | 0.462 | 0.00564 | 0.986 | 0.308 | 0.058 | 0.909 | 0.089 | 0.189 |
| | Evidential | UCE | 0.447 | 0.00118 | 0.910 | 0.310 | 0.164 | 0.901 | 0.143 | 0.259 |
| | | Ours | 0.469 | 0.00034 | 0.963 | 0.312 | 0.068 | 0.965 | 0.268 | 0.148 |

Table 22: Robustness Analysis (weather conditions - part 2): evaluation with LSS backbone on CARLA in diverse weather conditions. Best results are highlighted in red.

| pseudo OOD | model | loss | Pure Classification | | Misclassification | | | OOD Detection | | |
|---|---|---|---|---|---|---|---|---|---|---|
| | | | IoU ↑ | ECE↓ | AUROC↑ | AUPR↑ | FPR95↓ | AUROC↑ | AUPR↑ | FPR95↓ |
| | | | | | SoftRainNoon | | | | | |
| No | Baseline | CE | 0.395 | 0.00333 | 0.942 | 0.293 | 0.197 | 0.655 | 0.002 | 0.842 |
| | | Focal | 0.413 | 0.00133 | 0.963 | 0.299 | 0.145 | 0.734 | 0.002 | 0.758 |
| | Energy | CE | 0.387 | 0.00343 | 0.940 | 0.284 | 0.206 | 0.664 | 0.002 | 0.783 |
| | | Focal | 0.413 | 0.00133 | 0.963 | 0.300 | 0.145 | 0.731 | 0.002 | 0.764 |
| | Ensemble | CE | 0.439 | 0.00199 | 0.956 | 0.300 | 0.162 | 0.501 | 0.001 | 0.954 |
| | | Focal | 0.460 | 0.00118 | 0.975 | 0.303 | 0.132 | 0.462 | 0.001 | 0.957 |
| | Dropout | CE | 0.415 | 0.00290 | 0.933 | 0.286 | 0.220 | 0.485 | 0.001 | 0.959 |
| | | Focal | 0.417 | 0.00163 | 0.955 | 0.303 | 0.164 | 0.428 | 0.001 | 0.951 |
| | Evidential | UCE | 0.418 | 0.00222 | 0.833 | 0.272 | 0.313 | 0.587 | 0.002 | 0.836 |
| | | UFCE | 0.423 | 0.00016 | 0.923 | 0.308 | 0.136 | 0.560 | 0.002 | 0.833 |
| Yes | Energy | CE | 0.454 | 0.00256 | 0.981 | 0.330 | 0.082 | 0.913 | 0.068 | 0.231 |
| | | Focal | 0.441 | 0.00603 | 0.983 | 0.297 | 0.076 | 0.902 | 0.071 | 0.209 |
| | Evidential | UCE | 0.451 | 0.00120 | 0.913 | 0.317 | 0.158 | 0.897 | 0.124 | 0.236 |
| | | Ours | 0.471 | 0.00010 | 0.965 | 0.332 | 0.059 | 0.975 | 0.229 | 0.104 |
| | | | | | ClearSunset | | | | | |
| No | Baseline | CE | 0.387 | 0.00343 | 0.940 | 0.283 | 0.206 | 0.668 | 0.002 | 0.804 |
| | | Focal | 0.413 | 0.00134 | 0.963 | 0.300 | 0.145 | 0.734 | 0.002 | 0.758 |
| | Energy | CE | 0.387 | 0.00343 | 0.940 | 0.283 | 0.206 | 0.664 | 0.002 | 0.783 |
| | | Focal | 0.418 | 0.00159 | 0.967 | 0.318 | 0.128 | 0.755 | 0.003 | 0.739 |
| | Ensemble | CE | 0.439 | 0.00199 | 0.956 | 0.300 | 0.162 | 0.501 | 0.001 | 0.954 |
| | | Focal | 0.460 | 0.00118 | 0.975 | 0.303 | 0.132 | 0.463 | 0.001 | 0.956 |
| | Dropout | CE | 0.415 | 0.00290 | 0.933 | 0.286 | 0.220 | 0.485 | 0.001 | 0.959 |
| | | Focal | 0.417 | 0.00163 | 0.955 | 0.303 | 0.164 | 0.428 | 0.001 | 0.951 |
| | Evidential | UCE | 0.418 | 0.00222 | 0.833 | 0.272 | 0.313 | 0.587 | 0.002 | 0.835 |
| | | UFCE | 0.423 | 0.00015 | 0.923 | 0.307 | 0.136 | 0.560 | 0.002 | 0.833 |
| Yes | Energy | CE | 0.454 | 0.00256 | 0.981 | 0.330 | 0.082 | 0.913 | 0.068 | 0.231 |
| | | Focal | 0.445 | 0.00588 | 0.977 | 0.278 | 0.106 | 0.868 | 0.073 | 0.261 |
| | Evidential | UCE | 0.451 | 0.00120 | 0.913 | 0.316 | 0.158 | 0.897 | 0.124 | 0.236 |
| | | Ours | 0.471 | 0.00010 | 0.965 | 0.333 | 0.059 | 0.975 | 0.229 | 0.104 |
| | | | | | CloudySunset | | | | | |
| No | Baseline | CE | 0.374 | 0.00349 | 0.933 | 0.281 | 0.218 | 0.678 | 0.002 | 0.790 |
| | | Focal | 0.380 | 0.00145 | 0.966 | 0.307 | 0.143 | 0.771 | 0.003 | 0.771 |
| | Energy | CE | 0.373 | 0.00342 | 0.938 | 0.279 | 0.209 | 0.658 | 0.002 | 0.799 |
| | | Focal | 0.391 | 0.00133 | 0.966 | 0.300 | 0.141 | 0.766 | 0.003 | 0.717 |
| | Ensemble | CE | 0.401 | 0.00215 | 0.950 | 0.277 | 0.190 | 0.489 | 0.001 | 0.957 |
| | | Focal | 0.427 | 0.00143 | 0.973 | 0.282 | 0.157 | 0.425 | 0.001 | 0.964 |
| | Dropout | CE | 0.376 | 0.00317 | 0.930 | 0.267 | 0.235 | 0.448 | 0.001 | 0.959 |
| | | Focal | 0.398 | 0.00162 | 0.953 | 0.289 | 0.176 | 0.383 | 0.001 | 0.952 |
| | Evidential | UCE | 0.370 | 0.00246 | 0.834 | 0.265 | 0.310 | 0.612 | 0.002 | 0.791 |
| | | UFCE | 0.387 | 0.00030 | 0.919 | 0.298 | 0.148 | 0.538 | 0.002 | 0.803 |
| Yes | Energy | CE | 0.404 | 0.00277 | 0.974 | 0.319 | 0.121 | 0.865 | 0.059 | 0.318 |
| | | Focal | 0.427 | 0.00613 | 0.979 | 0.289 | 0.113 | 0.889 | 0.065 | 0.213 |
| | Evidential | UCE | 0.401 | 0.00147 | 0.885 | 0.288 | 0.209 | 0.871 | 0.104 | 0.328 |
| | | Ours | 0.431 | 0.00062 | 0.947 | 0.297 | 0.117 | 0.959 | 0.182 | 0.217 |
| | | | | | WetSunset | | | | | |
| No | Baseline | CE | 0.371 | 0.00344 | 0.933 | 0.283 | 0.213 | 0.684 | 0.002 | 0.780 |
| | | Focal | 0.391 | 0.00133 | 0.966 | 0.299 | 0.141 | 0.772 | 0.003 | 0.717 |
| | Energy | CE | 0.373 | 0.00342 | 0.938 | 0.278 | 0.209 | 0.658 | 0.002 | 0.799 |
| | | Focal | 0.391 | 0.00133 | 0.966 | 0.299 | 0.141 | 0.766 | 0.003 | 0.717 |
| | Ensemble | CE | 0.402 | 0.00214 | 0.950 | 0.277 | 0.190 | 0.489 | 0.001 | 0.957 |
| | | Focal | 0.427 | 0.00143 | 0.973 | 0.282 | 0.157 | 0.425 | 0.001 | 0.964 |
| | Dropout | CE | 0.376 | 0.00317 | 0.930 | 0.267 | 0.235 | 0.448 | 0.001 | 0.960 |
| | | Focal | 0.398 | 0.00162 | 0.953 | 0.289 | 0.176 | 0.383 | 0.001 | 0.953 |
| | Evidential | UCE | 0.370 | 0.00246 | 0.834 | 0.265 | 0.310 | 0.612 | 0.002 | 0.791 |
| | | UFCE | 0.387 | 0.00030 | 0.919 | 0.299 | 0.148 | 0.538 | 0.002 | 0.803 |
| Yes | Energy | CE | 0.404 | 0.00277 | 0.974 | 0.319 | 0.121 | 0.865 | 0.059 | 0.318 |
| | | Focal | 0.427 | 0.00613 | 0.979 | 0.288 | 0.113 | 0.889 | 0.065 | 0.213 |
| | Evidential | UCE | 0.401 | 0.00147 | 0.885 | 0.288 | 0.209 | 0.871 | 0.104 | 0.328 |
| | | Ours | 0.431 | 0.00062 | 0.947 | 0.296 | 0.117 | 0.959 | 0.182 | 0.217 |

Table 23: Robustness Analysis (weather conditions - part 3): evaluation with LSS backbone on CARLA in diverse weather conditions. Best results are highlighted in red.

| pseudo OOD | model | loss | Pure Classification | | Misclassification | | | OOD Detection | | |
|---|---|---|---|---|---|---|---|---|---|---|
| | | | IoU ↑ | ECE↓ | AUROC↑ | AUPR↑ | FPR95↓ | AUROC↑ | AUPR↑ | FPR95↓ |
| | | | | | WetCloudySunset | | | | | |
| No | Baseline | CE | 0.361 | 0.00353 | 0.932 | 0.279 | 0.214 | 0.680 | 0.002 | 0.788 |
| | | Focal | 0.391 | 0.00133 | 0.966 | 0.299 | 0.141 | 0.772 | 0.003 | 0.718 |
| | Energy | CE | 0.373 | 0.00342 | 0.938 | 0.279 | 0.209 | 0.658 | 0.002 | 0.799 |
| | | Focal | 0.391 | 0.00133 | 0.966 | 0.299 | 0.141 | 0.766 | 0.003 | 0.717 |
| | Ensemble | CE | 0.401 | 0.00215 | 0.950 | 0.277 | 0.190 | 0.489 | 0.001 | 0.957 |
| | | Focal | 0.427 | 0.00143 | 0.973 | 0.281 | 0.157 | 0.425 | 0.001 | 0.964 |
| | Dropout | CE | 0.376 | 0.00317 | 0.930 | 0.267 | 0.235 | 0.448 | 0.001 | 0.960 |
| | | Focal | 0.398 | 0.00162 | 0.953 | 0.289 | 0.176 | 0.383 | 0.001 | 0.953 |
| | Evidential | UCE | 0.370 | 0.00246 | 0.834 | 0.266 | 0.310 | 0.612 | 0.002 | 0.791 |
| | | UFCE | 0.387 | 0.00030 | 0.920 | 0.299 | 0.148 | 0.538 | 0.002 | 0.803 |
| Yes | Energy | CE | 0.404 | 0.00277 | 0.974 | 0.319 | 0.121 | 0.865 | 0.059 | 0.318 |
| | | Focal | 0.427 | 0.00613 | 0.979 | 0.288 | 0.113 | 0.889 | 0.065 | 0.213 |
| | Evidential | UCE | 0.401 | 0.00147 | 0.885 | 0.288 | 0.209 | 0.871 | 0.104 | 0.328 |
| | | Ours | 0.431 | 0.00062 | 0.947 | 0.297 | 0.117 | 0.959 | 0.182 | 0.217 |
| | | | | | SoftRainSunset | | | | | |
| No | Baseline | CE | 0.373 | 0.00342 | 0.938 | 0.279 | 0.209 | 0.681 | 0.002 | 0.810 |
| | | Focal | 0.448 | 0.00170 | 0.958 | 0.313 | 0.141 | 0.737 | 0.003 | 0.711 |
| | Energy | CE | 0.434 | 0.00300 | 0.926 | 0.277 | 0.213 | 0.674 | 0.002 | 0.770 |
| | | Focal | 0.448 | 0.00170 | 0.958 | 0.313 | 0.141 | 0.736 | 0.003 | 0.708 |
| | Ensemble | CE | 0.463 | 0.00194 | 0.944 | 0.276 | 0.187 | 0.504 | 0.001 | 0.958 |
| | | Focal | 0.490 | 0.00127 | 0.970 | 0.292 | 0.144 | 0.463 | 0.001 | 0.960 |
| | Dropout | CE | 0.431 | 0.00272 | 0.924 | 0.277 | 0.219 | 0.478 | 0.001 | 0.955 |
| | | Focal | 0.442 | 0.00140 | 0.952 | 0.311 | 0.154 | 0.428 | 0.001 | 0.953 |
| | Evidential | UCE | 0.443 | 0.00206 | 0.816 | 0.258 | 0.347 | 0.595 | 0.002 | 0.806 |
| | | UFCE | 0.441 | 0.00025 | 0.914 | 0.301 | 0.153 | 0.553 | 0.002 | 0.826 |
| Yes | Energy | CE | 0.490 | 0.00238 | 0.979 | 0.324 | 0.084 | 0.906 | 0.072 | 0.250 |
| | | Focal | 0.487 | 0.00415 | 0.985 | 0.305 | 0.070 | 0.922 | 0.090 | 0.158 |
| | Evidential | UCE | 0.482 | 0.00112 | 0.893 | 0.306 | 0.197 | 0.913 | 0.170 | 0.195 |
| | | Ours | 0.494 | 0.00029 | 0.956 | 0.325 | 0.072 | 0.975 | 0.276 | 0.105 |

Table 24: Variance for CVT on nuScenes.

| Num. Models | Model | Pure Classification | | Misclassification | | OOD | |
|---|---|---|---|---|---|---|---|
| | | IoU ↑ | ECE ↓ | AUROC ↑ | AUPR ↑ | AUROC ↑ | AUPR ↑ |
| 3 | UFCE-EUS-ER | $34.4 \pm 0.0013$ | $0.0793 \pm 0.000016$ | $94.4 \pm 0.0009$ | $32.8 \pm 0.0021$ | $92.4 \pm 0.021$ | $28.7 \pm 0.29$ |
| | UCE-ENT-EUS-ER | $31.3 \pm 0.00053$ | $0.353 \pm 0.00000511$ | $92.7 \pm 0.0000023$ | $31.8 \pm 0.001$ | $87.5 \pm 0.31$ | $25.4 \pm 0.00063$ |

### A.5.9 QUALITATIVE EVALUATIONS FOR MODEL COMPARISON

We qualitatively analyze the model performance with pixel-level prediction when viewed from a bird's eye perspective, and show the effectiveness of our proposed model.

We first present the semantic segmentation predictions in Figure 4. Our model demonstrates comparable segmentation performance to energy-based, dropout, and ensemble models. Compared to ENN-UCE, our model produces tighter object boundary predictions.

Figure 4: Comparison of Semantic Segmentation Performance: Each row represents an example, with the first column showing the ground truth labels, where the yellow regions indicate the positive class ("vehicle" in these examples). We visualize the predicted probabilities for the positive class generated by our model and four baselines. Brighter regions correspond to higher probability values.

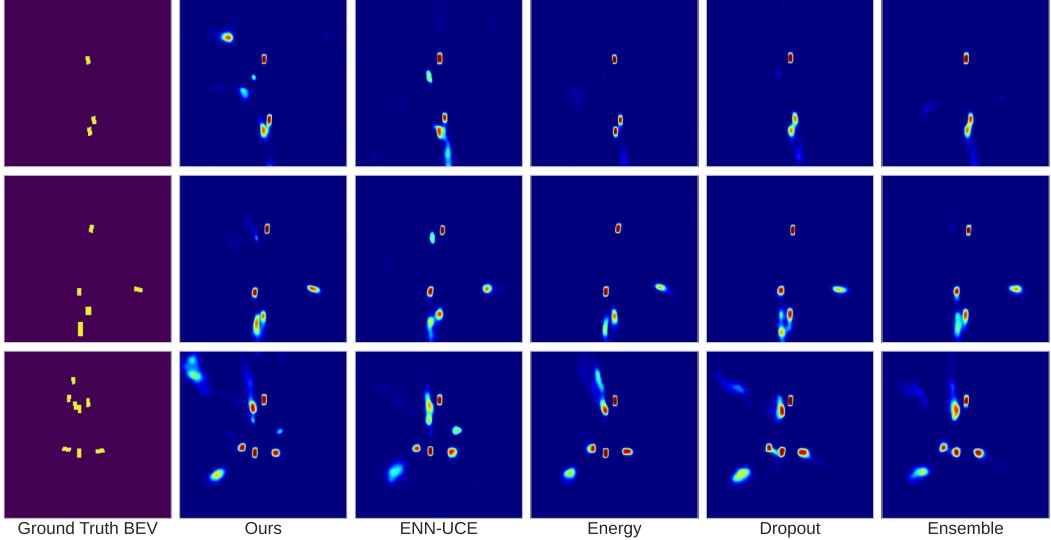

Ground Truth BEV     Ours     ENN-UCE     Energy     Dropout     Ensemble

We present the predicted aleatoric uncertainty in Figure 5. We anticipate that correctly classified pixels will exhibit low aleatoric uncertainty, while misclassified pixels will display high aleatoric uncertainty. Analyzing misclassification detection is complex because the ground truth varies across different model predictions. Based on these three frames, we cannot see a large performance difference between the various model variants.

Figure 6 presents the predicted epistemic uncertainty. Ideally, in-distribution (ID) pixels should exhibit low epistemic uncertainty, while out-of-distribution (OOD) pixels should show high uncertainty. Our focus is on the relative uncertainty levels between ID and OOD pixels rather than the absolute uncertainty scale. The results demonstrate that our proposed model achieves the most accurate identification of OOD pixels, indicating the best-predicted epistemic uncertainties. In contrast, the ENN-UCE baseline shows a significant increase in false positives. Even with the assistance of pseudo-OOD, the energy model fails to accurately locate OOD pixels. Dropout and ensemble models, unable to leverage pseudo-OOD, exhibit the worst performance.

Figure 5: Comparison of Predicted Aleatoric Uncertainty for Misclassification Detection: Each row represents an example, with each pair of columns corresponding to one model. The left column shows the misclassified labels, where yellow indicates misclassified pixels, while the right column visualizes the predicted aleatoric uncertainty for the same model, with brighter regions representing higher uncertainty values.

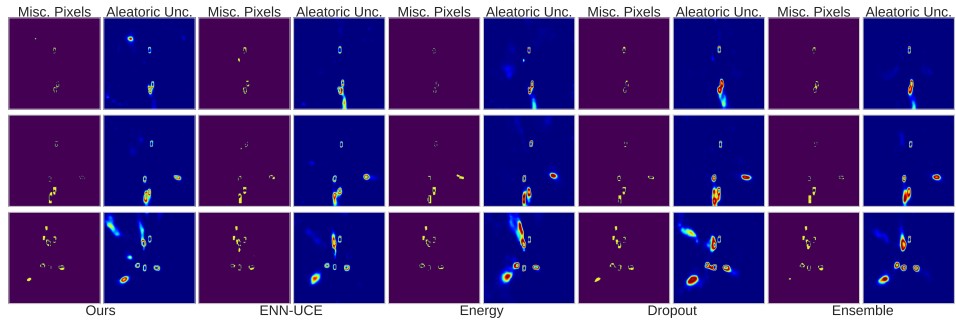

Figure 6: Comparison of Predicted Epistemic Uncertainty for OOD Detection: Each row represents an example, with the first column displaying the ground truth labels, where yellow regions indicate OOD pixels ("motorcycle" in these examples). The predicted epistemic uncertainty is visualized, with brighter regions indicating higher uncertainty values.

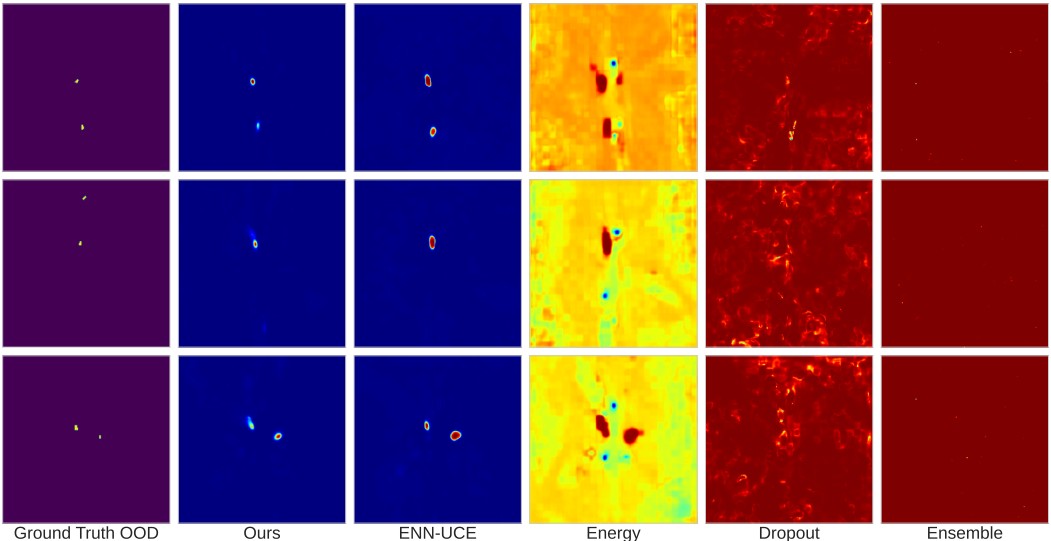

