# OpenReview forum: "Predictive Uncertainty Quantification for Bird's Eye View Segmentation: A Benchmark and Novel Loss Function"
_ICLR.cc/2025/Conference — ICLR 2025 Poster_

### Official Review · Reviewer_CuPe · 2024-10-24

**Soundness:** 2
**Presentation:** 2
**Contribution:** 3
**Rating:** 6
**Confidence:** 2

**Summary:**

This manuscript introduces a new benchmark and a new approach for predictive uncertainty quantification in Bird’s Eye View Semantic Segmentation (BEVSS) tasks.

BEV segmentation plays a key role in understanding the vehicle’s surrounding environment by creating a top-down representation using sensor data (e.g., cameras, LiDAR, radar). However, neural networks used for these tasks often produce overconfident predictions on unseen data and underconfident predictions in noisy data. The paper focuses on the two types of uncertainty—aleatoric (caused by noise in data) and epistemic (caused by model uncertainty due to lack of knowledge)—and their impact on safety-critical tasks like autonomous driving.

The authors propose a new loss function called Uncertainty-Focal-Cross-Entropy (UFCE), which mitigates under- and overfitting in BEV segmentation models by integrating regularization on uncertainty.

Additionally, the paper introduces a benchmark for evaluating five uncertainty quantification methods across three large autonomous driving datasets (CARLA, nuScenes, and Lyft) using two popular BEV segmentation backbones: Lift-Splat-Shoot and Cross-View Transformer. The primary goal is to quantify how well these methods detect misclassified pixels, improve model calibration, and identify out-of-distribution (OOD) data.

Experiments demonstrate that the UFCE loss improves both aleatoric and epistemic uncertainty prediction compared to the existing softmax and ensemble-based approaches. The paper reports a 4.758% improvement in OOD detection performance and gains in misclassification detection. The benchmark serves as a good resource for future work in predictive uncertainty quantification for BEVSS.

**Strengths:**

(+) The focus on uncertainty quantification for BEV segmentation is highly relevant to real-world applications in autonomous vehicles, where handling uncertain and dynamic environments is crucial for decision-making.

(+) The manuscript provides some good evaluations of uncertainty quantification techniques, analyzing their performance across various autonomous driving datasets and network backbones.

(+) The introduction of the UFCE loss function is a good contribution, effectively enhancing the calibration and uncertainty prediction of BEVSS models. It mitigates the shortcomings of traditional loss functions in handling aleatoric and epistemic uncertainties.

**Weaknesses:**

(-) While the manuscript provides a thorough benchmark for predictive uncertainty quantification, the individual uncertainty quantification techniques themselves are not particularly novel. Many of these approaches, such as deep ensembles and MC-Dropout, are well-established in the literature, and the work lacks an in-depth exploration of more recent or advanced uncertainty quantification methods.

(-) Although the paper addresses both types of uncertainty (aleatoric and epistemic), the explanations regarding how the proposed method specifically handles these uncertainties are not well developed. More technical insights on how the UFCE loss function balances both types of uncertainty and why it is better than existing methods are needed.

(-) The benchmark, while extensive, is focused primarily on the CARLA, nuScenes, and Lyft datasets. These datasets are useful but do not encompass the full range of scenarios an autonomous vehicle might encounter, particularly when it comes to highly dynamic environments like dense urban settings or environments with poor weather conditions. Evaluating on a wider variety of datasets, including those that feature more diverse sensor modalities or environments, would strengthen the paper’s claims. It would also be interesting to see how the benchmarked models perform on out-of-distribution data, such as those corruption sets from RoboBEV [R1], MapBench [R2], and Robo3D [R3].

(-) The manuscript lacks clarity in some sections, particularly in the technical description of the UFCE loss function and the details of the ablation studies. Some important concepts, such as the handling of noise and sparsity in the LiDAR point clouds, are not adequately explained, which might make it difficult for readers to fully grasp the method’s contributions.


**References:**
- [R1] S. Xie, et al. “Benchmarking and Improving Bird's Eye View Perception Robustness in Autonomous Driving,” arXiv, 2024.

- [R2] X. Hao, et al. “Is Your HD Map Constructor Reliable under Sensor Corruptions?” arXiv, 2024.

- [R3] L. Kong, et al. “Robo3D: Towards Robust and Reliable 3D Perception against Corruptions,” ICCV, 2023.

---

### Justification of Rating
The manuscript makes a contribution by focusing on uncertainty quantification in Bird’s Eye View Semantic Segmentation (BEVSS) for autonomous driving tasks.

The introduction of the Uncertainty-Focal-Cross-Entropy (UFCE) loss function is an interesting attempt to improve model calibration and uncertainty prediction, addressing a relevant real-world problem. The inclusion of a benchmark to evaluate different uncertainty quantification techniques across three datasets adds practical value.

However, I have limited experience in the area of OOD detection and am not fully familiar with the theoretical derivations behind the bounds of the UFCE loss function. As such, I leave more room for the other reviewers to assess the technical novelty and rigor of this contribution.

While the results are compelling, the work could have benefitted from more in-depth explanations of key components, such as handling aleatoric and epistemic uncertainties and how the proposed loss function truly differentiates itself from previous works.

Additionally, although the benchmark evaluation is thorough, it could be further expanded to include more diverse datasets and dynamic environments. The manuscript also lacks clarity in some technical sections, which slightly hampers the overall readability. For these reasons, I believe that while the manuscript has merits, it leaves some gaps that need to be addressed before it reaches its full potential.

**Questions:**

- **Q1:** Could the authors provide more detailed insights into how the UFCE loss function handles both aleatoric and epistemic uncertainty in BEVSS? Specifically, what mechanisms within the loss function enable it to balance both types of uncertainty effectively?

---

- **Q2:** How does the proposed UFCE loss function compare to other more recent uncertainty quantification techniques? Are there any specific examples where the UFCE loss provides a distinct advantage over methods such as deep ensembles or MC-Dropout?

---

- **Q3:** The paper discusses its benchmark evaluation on the CARLA, nuScenes, and Lyft datasets. Could the authors provide more information on how well the model performs in more complex or dynamic environments, such as dense urban settings or environments with adverse weather conditions?

---

- **Q4:** Could the authors explain the potential computational overhead of the UFCE loss function and other uncertainty quantification techniques? Specifically, how feasible is the proposed method for real-time applications in autonomous driving, considering resource-constrained environments?

---

- **Q5:** The manuscript lacks comparisons with recent benchmarks such as RoboBEV, MapBench, and Robo3D for out-of-distribution detection. Could the authors conduct additional experiments on these datasets to strengthen the claims about the robustness of their model?

---

> ### Comment · Reviewer_CuPe · 2024-11-26
>
> Since the authors didn't attempt to provide a rebuttal, I have to keep the current rating for this submission.

---

> > ### Author Response · Authors · 2024-11-26
> >
> > Dear Reviewer CuPe,
> >
> > Thank you for your valuable feedback and insightful suggestions. We are actively conducting related experiments to demonstrate the effectiveness and robustness of our work across additional benchmarks and backbones. As some of these experiments are time-intensive, we finalized the collection of all empirical results yesterday. We plan to submit the revised PDF tomorrow, including clarifications and the empirical results from the requested experiments, along with our detailed rebuttal addressing all reviewers' concerns. We look forward to further discussions to ensure we have thoroughly addressed your feedback.

---

> ### Author Response · Authors · 2024-11-29
> **Response to Question 1**
>
> **Q1: Mechanisms of UFCE Loss for Balancing Aleatoric and Epistemic Uncertainty**
>
> We appreciate the reviewer’s feedback and would like to clarify that our model leverages evidential deep learning (EDL), which theoretically enables the disentanglement of aleatoric and epistemic uncertainties. Specifically, in EDL, each pixel’s classification follows a Categorical distribution over predefined categories, and this Categorical distribution itself follows a Dirichlet distribution. The parameters of this Dirichlet distribution are predicted by the neural network, allowing for second-order uncertainty modeling in a single forward pass. This efficient approach facilitates the decomposition of uncertainty into distinct sources.
>
> In EDL, the predicted Dirichlet parameters represent "evidence," which quantifies the support provided by the training data that a sample belongs to a specific class. When the total evidence across all predefined classes is large, it indicates the presence of sufficient similar samples in the training set, corresponding to low epistemic uncertainty. Conversely, a lack of evidence suggests high epistemic uncertainty. On the other hand, significant conflicting evidence across multiple classes signifies high aleatoric uncertainty, arising from inherent ambiguity in the data.
>
> Our proposed framework consists of the following components. The first term is related to the enhancement of aleatoric uncertainty quantification and the last two are related to epistemic uncertainty quantification.
>
> 1. **UFCE Loss:** This component helps avoid overly peaked Dirichlet distribution predictions (where evidence is excessively high for the ground truth class and nearly zero for others, resulting in unrealistically low aleatoric and epistemic uncertainties). By mitigating this peaking effect, UFCE loss enhances the quality of aleatoric uncertainty predictions for in-distribution (ID) pixels, leading to improved calibration and misclassification detection.
> 2. **Evidence Regularization (ER):** This term incorporates pseudo-OOD data during training, improving the quality of epistemic uncertainty predictions by guiding the model to better distinguish ID and OOD samples.
> 3. **Epistemic Uncertainty Scaling (EUS):** This term focuses the model’s training efforts on samples with high epistemic uncertainty, helping to reduce false positives in OOD detection by emphasizing challenging samples.
>
> Together, these components enable the framework to effectively handle both types of uncertainty, improving calibration, misclassification detection, and OOD detection performance.
>
> **Effectiveness of the UFCE Loss to improve the aleatoric uncertainty of EDL in highly imbalanced data.**
>
> Our proposed **Uncertainty-Focal-Cross-Entropy (UFCE) loss** improves upon the Uncertainty-Cross-Entropy (UCE) loss, traditionally used in evidential neural networks (ENNs), particularly in addressing aleatoric uncertainty and calibration. The UCE loss optimizes the expected cross-entropy between the predicted Categorical distribution and the ground truth. However, as demonstrated in **Proposition 1**, minimizing UCE pushes the predicted Dirichlet distribution toward a one-hot target evidence distribution while simultaneously peaking it, resulting in overconfidence. This peaking is particularly problematic in scenarios with highly imbalanced data, such as BEVSS, where pixels of rare classes (e.g., vehicles) are overwhelmed by those of dominant classes (e.g., the background).
>
> The UFCE loss mitigates this overconfidence by introducing a focal-like mechanism, which reduces the model's confidence for correctly classified samples. As shown in **Equation 9** (Page 5), UFCE provides a trade-off through an upper bound formulation, which maximizes a term that implicitly regularizes the model to be less confident in evidence predictions for correctly classified samples. This behavior is crucial in BEVSS, where the imbalance between foreground and background pixels can otherwise bias the model toward overconfident predictions for dominant classes.
>
> **Gradient-Based Insights:** In addition to the theoretical intuition provided by the upper bound, **Proposition 2** further illustrates the relationship between the gradient norms of the last linear layer when optimizing UFCE versus UCE under the same network architecture. This relationship shows that the UFCE loss initially increases the evidence for the ground truth class faster than UCE, focusing more on less confident scenarios. This gradient behavior acts as an implicit weight regularizer, enabling the model to prioritize underrepresented or uncertain samples effectively.
>
> We provide supporting visual evidence in **Figure 1**, demonstrating how UFCE achieves better regularization in imbalanced scenarios. Further analysis in **Figures 2 and 3** elaborates on this behavior, highlighting UFCE’s robustness and its superior handling of both aleatoric and epistemic uncertainties.

---

> ### Author Response · Authors · 2024-11-29
> **Response to Question 2**
>
> **Q2: Comparison of UFCE Loss with Recent Uncertainty Quantification Techniques.**
>
> Thanks for your question related to more advanced uncertainty quantification techniques. To the best of our knowledge, there is no prior work specifically addressing epistemic and aleatoric uncertainty quantification in the BEVSS task. We expand on this in Appendix A.2, where we review related literature in closely aligned research domains.
>
> We begin by introducing uncertainty quantification in semantic segmentation for front camera images, a closely related domain with similar data-dependent properties. Traditional uncertainty quantification approaches, such as dropout and deep ensembles, and more recent or advanced approaches, such as evidential neural networks (ENN), have been explored for image segmentation segmentation tasks. The ENN based methods we studied in our work are more recent and representative uncertainty quantification techniques in the literature [1].
>
> However, these traditional and recent methods have not been explored for BEV-view semantic segmentation tasks in the literature. BEV segmentation is significantly different from image-based semantic segmentation, as it involves unique challenges stemming from the BEV view being a virtual view around the ego vehicle, generated based on the fusion of multiple camera images.
>
> For example, one of the unique challenges in BEV-view segmentation is the need to learn a mapping from pixels in different camera reviews to pixels in the BEV view.  In our updated PDF, we considered three representative BEVSS backbones, including **LSS**, **CVT**, and **Simpble-BEV** (a recent backbone added during the rebuttal as suggested by one of the reviewers), that rely on three different ways to learn the mapping, respectively. In particular,  **LSS** converts raw camera inputs into BEV representations by predicting depth distributions, constructing feature frustums, and rasterizing them onto a BEV grid, while **CVT** employs a transformer-based approach with cross-attention and camera-aware positional embeddings to align features into the BEV space. In contrast, **Simple-BEV** bypasses depth estimation entirely, projecting 3D coordinate volumes onto camera images to sample features, emphasizing efficiency and robustness to projection errors. Detailed descriptions can be found in Appendix A.2.1. Due to the different ways of learning the mapping, their neural network architectures are significantly different, accordingly.
>
> One of our main research objectives is to assess whether the aleatoric and epistemic uncertainties estimated by different uncertainty quantification methods are reliable across these three backbones.
>
> We introduce the first benchmark for evaluating uncertainty quantification methods in BEVSS, analyzing five representative approaches (softmax entropy, energy, deep ensemble, dropout, and evidential) across four datasets (CARLA (Dosovitskiy et al.,2017), nuScenes (Caesar et al., 2020), Lyft (Kesten et al., 2019), and a corrupted dataset, nuScenes-C (Xie et al. 2024)) using the three BEVSS backbones. Among these methods, energy-based and evidential neural networks (ENN)-based are more recent and state-of-the-art uncertainty quantification methods.
>
> We use ENN-UCE to denote the classical EDL model learned based on the UCE loss. We qualitatively compare our proposed model with baseline approaches, including MC-Dropout, deep ensembles, and the classic ENN-UCE model. The comparisons are illustrated in Figures 4–6:
>
> * **Figure 4:** Semantic segmentation predictions.
> * **Figure 5:** Aleatoric uncertainty predictions with respect to misclassified pixels.
> * **Figure 6:** Epistemic uncertainty predictions for the OOD detection task.
>
> **Epistemic Uncertainty:** Our model significantly reduces false positives near true OOD objects compared to ENN-UCE. The energy model, even with pseudo-OOD assistance, struggles to detect OOD pixels effectively, assigning high uncertainty to background regions and low uncertainty to ID pixels. Dropout and ensemble models also fail to identify OOD pixels effectively.
>
> **Semantic Segmentation and Aleatoric Uncertainty:** Our model predicts tighter object boundaries than ENN-UCE but shows no major differences in overall segmentation accuracy. Aleatoric uncertainty predictions are comparable across models, with our approach slightly better-aligning uncertainty with misclassified pixels.
>
> References:
>
> [1] Ancha, Siddharth, Philip R. Osteen, and Nicholas Roy. "Deep Evidential Uncertainty Estimation for Semantic Segmentation under Out-Of-Distribution Obstacles." In Proc. IEEE Int. Conf. Robot. Autom. 2024\.

---

> ### Author Response · Authors · 2024-11-29
> **Response to Question 5**
>
> **Q5: Evaluating Model Robustness on RoboBEV, MapBench, and Robo3D for OOD Detection.**
>
> In the updated PDF, we have evaluated our model on the **nuScenes-C** dataset introduced in RoboBEV, which applies various corruptions to the nuScenes validation set. While RoboBEV directly aligns with our Bird’s Eye View (BEV) semantic segmentation task, the other two benchmarks, MapBench and Robo3D, focus on tasks that are less related and were therefore not included in this study. For consistency with RoboBEV, which reports results for CVT-based models, we used CVT as the backbone in our experiments. Performance was tested across all seven types of corruption (Camera Crash, Frame Lost, Color Quantization, Motion Blur, Brightness, Snow, and Fog) at three levels of severity (Easy, Medium, Hard).
>
> The results are presented in **Table 19**, with a detailed analysis in **Section A.5.6**. Each combination of corruption type and severity level constitutes a unique scenario, resulting in a total of 21 scenarios. The following are key insights:
>
> **Loss Function Design and Its Role**
>
> Our ENN model is trained using a loss function comprising three components: the **UFCE loss**, **Evidence Regularization (ER)**, and **Epistemic Uncertainty Scaling (EUS)**.
>
> * The **UFCE loss** replaces the traditional UCE loss to enhance aleatoric uncertainty quantification.
> * The **ER** and **EUS** terms are specifically designed to improve epistemic uncertainty quantification. We use aleatoric uncertainty estimates for misclassification detection and epistemic uncertainty estimates for OOD detection.
>
> **Performance on RoboBEV (nuScenes-C)**
>
> 1. **Misclassification Detection**:
>    * Our model achieved the highest AUROC in **14 out of 21 scenarios** and the second highest in the remaining 7 scenarios.
>    * It achieved the highest AUPR in **13 out of 21 scenarios** and the second highest in the remaining 8 scenarios.
>    * Our model consistently outperformed the ENN-UCE baseline across all 21 scenarios for both AUROC and AUPR.
>    * These results validate the effectiveness of the proposed UFCE loss in improving aleatoric uncertainty quantification.
> 2. **OOD Detection**:
>    * Our model achieved the best AUPR in **16 out of 21 scenarios** and the second best in the remaining 5 scenarios.
>    * For AUROC, our model achieved the best performance in **8 scenarios** and the second best in the remaining **13 scenarios**.
> 3. **AUROC vs. AUPR Trade-Off**:
>    * AUROC and AUPR provide complementary perspectives on ranking quality for distinguishing positive and negative BEV pixels. AUPR is particularly sensitive to the quality of the top-ranked BEV pixels, focusing on how well the model prioritizes true positives. In contrast, AUROC reflects the overall ranking quality across all BEV pixels, capturing the model's ability to separate positives from negatives throughout the entire dataset.
>    * While our model occasionally shows slightly lower AUROC than ENN-UCE in some scenarios, it consistently achieves higher AUPR. This indicates that our model ranks true positives higher, which is particularly valuable in applications where human experts verify top-ranked instances (e.g., misclassified pixels or anomalies). High AUPR ensures a higher rate of true positives among the top-ranked predictions, making it a more suitable metric in cost-sensitive environments.
>
>
> **Insights on Epistemic Uncertainty and Corruption Severity**
>
> * Our model demonstrated robust performance in OOD detection across CARLA, nuScenes, and Lyft datasets, achieving the best results for both AUROC and AUPR.
> * On nuScenes-C, while our model showed the best AUPR, it was second-best on AUROC in certain scenarios for OOD detection. This suggests that data corruptions in these scenarios may slightly degrade the quality of epistemic uncertainty quantification for lower-ranked pixels, though top-ranked pixels remain unaffected. We plan to address this in future work to enhance robustness for lower-ranked predictions.
> * The impact of corruption severity was significant, with performance degrading under higher severity levels. This aligns with findings from the RoboBEV study and highlights the challenge of domain shifts in OOD scenarios.
>
> To summarize, the results from the nuScenes-C dataset demonstrate the robustness of our approach across diverse environmental conditions and domain shifts. Our model outperforms the ENN-UCE baseline in most scenarios, achieving state-of-the-art performance in misclassification and OOD detection tasks. While the results highlight the strengths of our method, they also identify areas for future improvements, particularly in handling lower-ranked predictions under severe domain shifts.

---

> ### Author Response · Authors · 2024-11-29
> **Response to Question 3, 4 & Weakness**
>
> **Q3: Evaluating Model Performance in Complex and Dynamic Environments Beyond Standard Benchmarks**
>
> Thank you for the suggestion. We have conducted supplementary experiments to address this point.
>
> For the CARLA dataset, we included four towns representing diverse urban layouts: Town 10 simulates a dense downtown urban environment, while Town 7 represents a rural setting. We also considered 10 weather conditions, such as varying brightness (e.g., noon and sunset) and weather types (e.g., clear, cloudy, wet, mid-rainy, and soft-rain). We evaluated our model under these conditions separately, and the results are presented in Tables 20–23 (Pages 35 to 38). Our analysis reveals varying model performance depending on the layout and weather conditions. However, our model consistently achieved the best performance in OOD detection across all baselines and demonstrated better calibration and misclassification detection than UCE models, regardless of weather or layout variations. Details on dataset generation and statistics are provided in Appendix A.4, with full results and analysis in Appendix A.5.7.
>
> **Q4: Computational Efficiency of UFCE Loss in Real-Time Autonomous Driving Applications.**
>
> We have expanded the complexity analysis in Table 3 to include an evaluation of the ENN model with different loss functions. Our analysis shows no significant computational difference between the proposed UFCE loss and the original UCE loss. Specifically, our model requires a similar training time as standard segmentation models and performs a single forward pass during inference. This computational efficiency ensures that our approach is equally suitable for real-time tasks in autonomous driving, comparable to traditional segmentation models.
>
> **Our response to the weaknesses:**
>
> Our answers to the above questions have covered the first three weaknesses. About the fourth weakness about clarity in the technical description of the UFCE loss function and the details of the ablation studies. We have revised the manuscript to provide a more detailed technical explanation of the UFCE loss function. Specifically, the proposed UFCE loss is formally defined in Equation 8, with its derivation detailed in Appendix A.1. The motivation for addressing the limitations of the previous UCE loss is discussed in Section 3.1, and the verification of how the UFCE loss improves calibration is presented in Section 3.2.
>
> We have also expanded the descriptions of the ablation studies in Section 4.2 to analyze the effects of the three primary components of our proposed model. The results of these investigations are presented in Table 5, with detailed observations provided in the corresponding section.
>
> Our model relies solely on camera RGB images as input and does not incorporate LiDAR data. Therefore, handling noise and sparsity in LiDAR point clouds falls outside the scope of this work.

---

> > ### Comment · Reviewer_CuPe · 2024-12-02
> >
> > Dear Authors,
> >
> > Thanks for the comprehensive and thoughtful responses to the review comments, although it's a bit late and seems too long :)
> >
> > I have read the responses, as well as the comments from other reviewers.
> >
> > I appreciate the effort you put into addressing the concerns raised during the review process, particularly in clarifying the technical aspects of your proposed UFCE loss function and expanding the experimental analysis.
> >
> > The responses have addressed most of my concerns, including:
> > - The clarification on the UFCE loss function, where the visual evidence and gradient-based analysis offer some support for your claims and help illustrate the improvements over traditional UCE loss.
> > - The expanded benchmark and comparisons.
> > - The performance in complex and dynamic environments.
> > - The analysis of the computational efficiency.
> >
> > Given the improvements made, I am inclined to upgrade the rating from 5 to 6 for this submission. However, since I am not directly working on this specific topic, I have decided to lower my confidence score (from 3 to 2) and will defer to other reviewers to assess the broader contributions and technical novelty of the work.
> >
> > Thanks again for your responses and for addressing the feedback constructively.
> >
> > Sincerely,
> >
> > Reviewer CuPe

---

> > > ### Author Response · Authors · 2024-12-02
> > >
> > > Dear reviewer CuPe,
> > >
> > > We appreciate your encouraging and constructive feedback. We have included most of the responses in our revised PDF and will ensure all the issues are clarified in our future revised version.
> > >
> > > Sincerely,
> > >
> > > Authors

---

### Official Review · Reviewer_Uxrf · 2024-10-27

**Soundness:** 3
**Presentation:** 2
**Contribution:** 3
**Rating:** 6
**Confidence:** 3

**Summary:**

The task tackled in this paper is Bird’s Eye View (BEV) semantic segmentation. The authors present  a comprehensive benchmark for predictive uncertainty quantification and introduce a novel loss function, i.e., Uncertainty-Focal-Cross-Entropy
(UFCE). The studies focus on model calibration, uncertainty quantification (detecting misclassified pixels) and out-of-distribution (OOD) detection. The methods are evaluated across three popular datasets and two representative backbones.

**Strengths:**

- The problems addressed in the paper are very important in the area of safety in neural networks.
- The new loss function presented is well thought out and mathematically motivated.
- The very well-known uncertainty methods are compared with each other. Different datasets are used, both synthetic and real-world.
- The methods are briefly explained in the main part, and details are provided in the appendix. It is a good balance between information and details in the method section.
- The implementation details are very well documented, also the dataset details in terms of creation and splitting etc.
- The complexity analysis and the ablation studies show a good comparison.

**Weaknesses:**

- A figure for BEVSS and uncertainty estimation at the beginning of the paper would have an explanatory effect.
- The experiments section is very confusing, constantly going back and forth between results in the main section and in the appendix. Differences between the datasets are not emphasized enough. For example, you could focus on one dataset and one network in the main part and have all the other experiments in the appendix. If the results for the datasets/networks differ, a paragraph with the most important findings could be written.
- The appendix on page 24 is also a bit mixed up with the headings printed, but the tables are in different places.
- I am missing a related work section on the papers that have already dealt with uncertainty estimation and OOD detection for the BEVSS task. This makes it difficult for me to assess the added value of this paper.

**Questions:**

- The two selected networks seem to be representative, but not SOTA, which would be expected for benchmark paper. What were the reasons for selecting these two networks and how does this choice affect the generalizability of the benchmark results?
- The in-distribution results between the uncertainty methods are very similar and provide little information. What is this supposed to show?

---

> ### Author Response · Authors · 2024-11-29
> **Response to Questions**
>
> **Q1: Why LSS and CVT?**
>
> We appreciate the reviewer’s feedback on the selected network backbones. BEV-view semantic segmentation tasks in the literature are significantly different from image semantic segmentation tasks, where traditional (e.g, drop-out, deep ensembles) and recent (e.g., energy-based, evidential deep learning) uncertainty quantification methods have been explored. BEV view is a virtual view around the ego vehicle generated based on the fusion of multiple camera images. One of the unique challenges in BEV-view segmentation is the need to learn a mapping from pixels in different camera reviews to pixels in the BEV view.
>
> The **LSS** and **CVT** are two representative BEVSS (BEV semantic segmentation) backbones that rely on two representative ways to learn the mapping. In particular,  **LSS** converts raw camera inputs into BEV representations by predicting depth distributions, constructing feature frustums, and rasterizing them onto a BEV grid, while **CVT** employs a transformer-based approach with cross-attention and camera-aware positional embeddings to align features into the BEV space. Due to the different ways in learning the mapping, their neural network architecture is significantly different, accordingly.
>
> In our updated PDF, we have incorporated new experiments using a more recent BEVSS backbone, **Simple-BEV** [1] (ICRA 2023). In contrast to LSS and CVT, **Simple-BEV** bypasses depth estimation entirely, projecting 3D coordinate volumes onto camera images to sample features, emphasizing efficiency and robustness to projection errors. Detailed descriptions can be found in Appendix A.2.1. We selected Simple-BEV over the state-of-the-art model PointBEV [2] (CVPR 2024\) with public code due to its higher adoption and reliability: Simple-BEV has 90 citations and 496 GitHub stars, compared to PointBEV’s 6 citations and 94 stars. This made Simple-BEV a safer and more reliable choice for obtaining robust results within our limited time frame. Furthermore, as shown in Table 1 of the PointBEV paper, Simple-BEV achieves runner-up performance, outperforming other recent backbones, such as BEVFormer [3] (ECCV 2022\) and BAEFormer [4] (CVPR 2023), further supporting its inclusion in our study.
>
> The results based on Simple-BEV are presented in Table 16 and demonstrate consistent performance when compared to LSS and CVT as backbones. While our model with Simple-BEV as backbone delivers better segmentation performance compared to LSS and CVT as backbones, no significant improvement in OOD detection performance was observed. For our proposed model, we observed consistent results regardless of whether LSS, CVT, or Simple-BEV were used as the backbone. Our model consistently achieves the best performance in calibration and OOD detection while maintaining comparable performance in misclassification detection.
>
> References:
>
> [1]: Harley, Adam W., et al. "Simple-bev: What really matters for multi-sensor bev perception?." *2023 IEEE International Conference on Robotics and Automation (ICRA)*. IEEE, 2023\.
>
> [2]: Chambon, Loick, et al. "PointBeV: A Sparse Approach for BeV Predictions." *Proceedings of the IEEE/CVF Conference on Computer Vision and Pattern Recognition*. 2024\.
>
> [3]: Zhiqi Li, Wenhai Wang, Hongyang Li, Enze Xie, Chonghao Sima, Tong Lu, Yu Qiao, and Jifeng Dai. BEVFormer: Learning bird’s-eye-view representation from multi-camera images via spatiotemporal transformers. In ECCV, 2022\.
>
> [4]: Cong Pan, Yonghao He, Junran Peng, Qian Zhang, Wei Sui, and Zhaoxiang Zhang. BAEFormer: Bi-directional and early interaction transformers for bird’s eye view semantic segmentation. In CVPR, 2023
>
> **Q2: The in-distribution results between the uncertainty methods are very similar and provide little information. What is this supposed to show?**
>
> The ENN architecture replaces the standard softmax activation function in the final layer with a ReLU function to predict the parameters of the Dirichlet distribution for pixel-wise classification. In our proposed approach, the ENN model parameters are estimated using the UFCE loss function alongside two regularization terms: Evidence Regularization (ER) and Epistemic Uncertainty Scaling (EUS), as detailed in Section 3\. Our goal is to present in-distribution results to demonstrate that the trained ENN model achieves segmentation accuracy comparable to other baseline methods.

---

> ### Author Response · Authors · 2024-11-29
> **Response to Weakness**
>
> **W1: Uncertainty-aware BEVSS Framework Visualization**.
>
> Thanks for the suggestion, we plot a figure for the proposed framework and it can be found in [https://drive.google.com/file/d/1cTDk4VGpcXGH9gwhd7MfH-yNu5ymXwo\_/view?usp=sharing](https://drive.google.com/file/d/1cTDk4VGpcXGH9gwhd7MfH-yNu5ymXwo_/view?usp=sharing)
>
> **W2: Prensentation of Experiment Section**.
>
> Thanks for the suggestion. In our revised PDF, we revise the experiment section and organize the result tables in the appendix.
>
> In the main paper, we present results on nuScenes using LSS and CVT as backbones, covering predicted segmentation and aleatoric uncertainty (Table 1), epistemic uncertainty (Table 2), running time (Table 3), robustness analysis (Table 4), and ablation studies (Table 5). In the appendix, we provide results on CARLA and Lyft using all backbones including LSS, CVT and Simple-BEV (Tables 13–16). Additionally, we include hyperparameter analysis (Tables 7–9) and robustness experiments (Tables 17–23), which encompass detailed results on corrupted nuScenes-C, diverse town and weather conditions in CARLA, and pseudo-OOD selections. Qualitative comparisons are provided in Figures 4–6.
>
> **W3: Presentation in Appendix for Additional Experiments**.
>
> Thanks for pointing out the empty subsections in Appendix A.5. We have updated Appendix A.5 in our revised PDF. This section has the following subsections: Full evaluation on Lyft (A.5.1), Quantitative evaluation on CARLA (A.5.2), Quantitative evaluation on nuScenes (Road Segmentation) (A.5.3), Quantitative evaluation on Simpble-BEV on nuScenes (A.5.4), Robustness on the selection of pseudo-OOD (A.5.5), Robustness on corrupted dataset (A.5.6), Robustness on diverse conditions (CARLA) (A.5.7), and Robustness on model initializations (A.5.8), and Qualitative evaluations for model comparison (A.5.9).
>
> **W4:  Related Work Section on Uncertainty Estimation and OOD Detection for BEVSS**.
>
> Thanks for the suggestion. To the best of our knowledge, there is no prior work specifically addressing epistemic and aleatoric uncertainty quantification in the BEVSS task. We expand the literature review in Appendix A2.2 with closely related research domains.
>
> We begin by introducing uncertainty quantification in semantic segmentation for camera views, a closely related domain with similar data-dependent properties. State-of-the-art models in this area adapt traditional Bayesian methods (e.g., dropout, ensembles), feature-density-based approaches, and evidential deep learning to pixel-level segmentation tasks. Our experimental evidence underscores the challenges of directly transferring camera-view uncertainties to the BEV space. Specifically, fusing uncertainties from an ENN model trained on camera-view segmentation into the BEV space results in much lower quality compared to using an ENN model trained directly on BEV segmentation. This highlights the need for tailored approaches to uncertainty quantification in BEVSS.
>
> Next, we discuss robustness and calibration in BEVSS tasks, which are conceptually adjacent to uncertainty quantification. Additionally, we discuss the so-called "uncertainty quantification" in downstream tasks. However, none of these works involve epistemic uncertainty estimation, a critical component for OOD detection.

---

> ### Comment · Reviewer_Uxrf · 2024-12-01
>
> Dear authors, thank you for the reply.
> After reading the other reviews and the responses, I maintain my score and think that this paper is acceptable.
> The new method illustration helps clarity and should also be included in the paper.

---

> > ### Author Response · Authors · 2024-12-02
> >
> > Dear reviewer Uxrf,
> >
> > We appreciate your encouraging feedback. We will include the new method illustration and other related clarifications in our future revised version.

---

### Official Review · Reviewer_6JWb · 2024-10-29

**Soundness:** 2
**Presentation:** 1
**Contribution:** 2
**Rating:** 3
**Confidence:** 3

**Summary:**

The paper addresses the challenge of uncertainty quantification in BEV segmentation, a critical component of autonomous vehicle perception. The authors propose a loss function, called uncertainty-focal-cross-entropy (UFCE), in order to improve uncertainty estimation and calibration of BEV segmentation. The authors conduct a comprehensive benchmark, evaluating various uncertainty quantification methods on popular datasets and backbones.

**Strengths:**

The paper presents an evaluation benchmark for uncertainty quantification in BEV semantic segmentation, which consists of three datasets like CARLA, nuScenes, and Lyft.


The benchmark includes two backbones that are proposed for general BEV semantic segmentation. The benchmark provides insights into potential improvements of previous BEV semantic segmentation models.

The paper introduces a novel loss function, UFCE, which is specifically designed for imbalanced data in BEV segmentation.

**Weaknesses:**

While the paper focuses on uncertainty quantification of the BEV segmentation task. As mentioned in the introduction, it would be beneficial to widely explore the impact of uncertainty on other downstream tasks such as motion planning and decision-making of the advanced driver-assistance systems (ADAS).

A more detailed comparison with other state-of-the-art uncertainty quantification methods would strengthen the finding and prove the effectiveness of the proposed loss function. For example, there is only a comparison between UCE and UCFE.

While the paper focuses on evidential deep learning, a more comprehensive comparison with other uncertainty quantification methods, such as Bayesian neural networks, Monte Carlo dropout, and deep ensembles, would be valuable.

The organization of the results is not easy to follow. For example, Sec. 4.2 presents results from ten tables, but the tables are not well structured. Therefore, the presentation and organization of the experimental results could be improved.

The comparison with other robust BEV methods is missing.

**Questions:**

Why is the evidential neural network selected to conduct uncertainty quantification for BEV semantic segmentation? Is only the last softmax function replaced by using the relu function so as to have the Dirichelt distribution of the pixel-wise classification prediction? How about this method compared to other uncertainty estimation methods?

Is there any comparison between the ENN and other uncertainty measure methods? Or the combination of these methods? For example, the methods mentioned in Sec.2, like Bayesian Neural Networks, Monte Carlo Dropout, and Deep Ensembles?

How do the authors prove that the UCE loss has also the issue that the optimization reduces the NLL value by increasing the probability of the predicted class? Is there any case study about the limitation of UCE loss?

In Sec. 3.2, the authors claim that the UFCE loss improves the calibration over UCE loss. Is there any detail or training results for comparison?

For the comparison of calibration between UCE and UFCE, the authors claim that models with focal-based loss exhibit lower ECE scores, indicating better calibration than cross-entropy based models. But in Table 1, the performance of UFCE is not consistently better than UCE.


The presentation and organization of the experimental results/tables make it difficult to follow. For example, Sec. 4.2 presents the results from Table 1 to 10.  Not all tables need to show comparison with all methods. For instance, some tables can focus on comparing UCE and UFCE.

Are there any visualisation results to show the calibration between UCE and UFCE?

How about the comparison with the most recent BEV methods? Also the comparison with other robust BEV segmentation models?

---

> ### Author Response · Authors · 2024-11-29
> **Response to Question 1**
>
> **Q1: Why Choose Evidential Neural Networks for Bird’s eye view (BEV) Semantic Segmentation?**
>
> We appreciate the reviewer’s feedback on the issues related to the evidential neural networks (ENN). We clarify the issues on two items: (a) Advantage of evidential neural network and (b) ENN vs. Classification Models.
>
> **(a) Advantage of ENN**
>
> In our study, we introduced the first benchmark for evaluating uncertainty quantification methods in BEVSS, analyzing five representative approaches: softmax entropy, energy, deep ensemble, dropout, and evidential neural networks (ENN), across three popular datasets. Among these, we found ENNs to be the most promising in terms of both effectiveness and efficiency for BEVSS tasks.
>
> In the following, we provide more detailed insights into how ENN handles both aleatoric and epistemic uncertainty in BEV semantic segmentation (BEVSS).
>
>
> The ENN model leverages evidential deep learning (EDL), which theoretically enables the disentanglement of aleatoric and epistemic uncertainties. Specifically, in the ENN model, each pixel’s classification follows a Categorical distribution over predefined categories, and this Categorical distribution itself follows a Dirichlet distribution. The parameters of this Dirichlet distribution are predicted by the neural network, allowing for second-order uncertainty modeling in a single forward pass. This efficient approach facilitates the decomposition of uncertainty into distinct sources.
>
> In ENN, the predicted Dirichlet parameters represent "evidence," which quantifies the support provided by the training data that a sample belongs to a specific class. When the total evidence across all predefined classes is large, it indicates the presence of sufficient similar samples in the training set, corresponding to low epistemic uncertainty. Conversely, a lack of evidence suggests high epistemic uncertainty. On the other hand, significant conflicting evidence across multiple classes signifies high aleatoric uncertainty, arising from inherent ambiguity in the data.
>
> ENN has recently demonstrated state-of-the-art performance for uncertainty quantification in semantic segmentation of hyperspectral imaging data and camera imaging data [1, 2].
>
> In this paper, we acknowledge the challenges of training a well-calibrated ENN using the UCE (uncertainty cross-entropy) loss for highly imbalanced BEVSS data. To address these issues, we propose a new optimization loss function UFCE (uncertainty focal cross entropy)-EUS-ER (Equation 12\) specifically tailored to the BEVSS task, improving the quality of predicted uncertainties.
>
> Our answers to Q1 and Q2 of the reviewer CuPe provide more detailed insights into the following related questions: (1) how the UFCE loss function and two evidence-based regularization terms handle both aleatoric and epistemic uncertainty in BEVSS; (2) what mechanisms within the loss function enable it to balance both types of uncertainty effectively; and (3) How does the proposed UFCE loss function compare to other more recent uncertainty quantification techniques.
>
>
>
> **(b) ENN vs. Classification Models**
>
> From the model architecture perspective, the ENN shares the same structure as the backbones, with the main difference being the replacement of the final softmax activation with a non-negative ReLU function. Backbones refer to segmentation models like LSS, CVT, and Simple-BEV used in this work.
>
> From an optimization perspective, the ENN is trained using the UCE loss (often accompanied by an entropy regularization term \[1\]). The UCE loss is defined as the expectation of the cross-entropy loss, where the class probability vector follows the Dirichlet distribution parameterized by the ENN. In our work, we extend this framework by proposing a new loss function to address the limitations of the UCE loss for highly imbalanced BEVSS data.
>
> From the model prediction, traditional classification models predict categorical distributions. In contrast, the ENN predicts the conjugate prior, Dirichlet distribution, which models the distribution over categorical distributions. This allows us to define multidimensional uncertainties, including both aleatoric and epistemic uncertainties, based on the parameters of the predicted Dirichlet distribution.
>
>
> References:
>
> [1] Yu, Linlin, Yifei Lou, and Feng Chen. "Uncertainty-aware Graph-based Hyperspectral Image Classification." The Twelfth International Conference on Learning Representations. 2024\.
>
> [2] Ancha, Siddharth, Philip R. Osteen, and Nicholas Roy. "Deep Evidential Uncertainty Estimation for Semantic Segmentation under Out-Of-Distribution Obstacles." In Proc. IEEE Int. Conf. Robot. Autom. 2024\.

---

> > ### Author Response · Authors · 2024-11-30
> > **Response to Question 3**
> >
> > **Q3 (a) How do the authors prove that the UCE loss has also the issue that the optimization reduces the NLL value by increasing the probability of the predicted class?**
> >
> > **Issue of CE loss:**
> >
> > Overconfidence of deep neural network is frequently correlated with overfitting the negative log-likelihood (NLL), since even with a classification error of zero (indicative of perfect calibration), the NLL can remain positive. The optimization algorithm may continue to reduce this value by increasing the probability of the predicted class.
> >
> > **Issue of UCE loss:**
> >
> > In our original manuscript, we stated: *"We prove that the UCE loss also has this issue."* However, we realized this statement may be unclear or misleading.
> >
> > In the revised manuscript, we clarify this statement as follows: *"We demonstrate that the UCE loss suffers from a similar issue. Specifically, even with a perfect prediction of evidence volume, the UCE loss remains positive, and increasing the evidence for the predicted class further decreases the UCE loss."*
> >
> > To support this claim, we provide both a formal proof and an intuitive explanation.
> >
> > **Proposition:**
> > The UCE loss $\mathcal{L}^{UCE}$ is always positive and monotonically decreasing with respect to the Dirichlet parameter $\alpha_{c^*}$, which corresponds to the ground truth class $c^*$.
> >
> > $
> > \mathcal{L}^{UCE} = \psi\left(\alpha_{c^*} + \sum_{c \neq c^*} \alpha_c\right) - \psi(\alpha_{c^*}),
> > $
> >
> > where $\alpha_{c} \geq 1$ is the Dirichlet parameter corresponding to class $c$, and $\psi(\cdot)$ is the digamma function.
> >
> >
> > **Proof:**
> >
> > 1. **Non-Negativity:**
> >    The digamma function $\psi(x)$ is monotonically increasing with respect to $x$ for $x>0$. Therefore,
> >
> >    $
> >    \psi\left(\alpha_{c^*} + \sum_{c \neq c^*} \alpha_c\right) > \psi(\alpha_{c^*}),
> >    $
> >
> >    since $\alpha_{c^*} + \sum_{c \neq c^*} \alpha_c > \alpha_{c^*}$.
> >    Hence,
> >    $
> >    \mathcal{L}^{UCE} = \psi\left(\alpha_{c^*} + \sum_{c \neq c^*} \alpha_c\right) - \psi(\alpha_{c^*}) > 0.
> >    $
> >
> >
> > 2. **Monotonicity:**
> >    The derivative of $\mathcal{L}^{UCE}$ with respect to $\alpha_{c^*}$ is given by:
> >
> >
> >    $
> >    \frac{\partial \mathcal{L}^{UCE}}{\partial \alpha_{c^*}} = \psi^1\left(\alpha_{c^*} + \sum_{c \neq c^*} \alpha_c\right) - \psi^1(\alpha_{c^*}),
> >    $
> >
> >
> >    where $\psi^1(\cdot)$ is the trigamma function. The trigamma function $\psi^1(x)$ is monotonically decreasing for $x > 0$.
> >
> >    Since $\alpha_{c^*} + \sum_{c \neq c^*} \alpha_c > \alpha_{c^*}$, we have:
> >
> >
> >    $
> >    \psi^1\left(\alpha_{c^*} + \sum_{c \neq c^*} \alpha_c\right) < \psi^1(\alpha_{c^*}).
> >    $
> >
> >
> >    Therefore,
> >
> >
> >    $
> >    \frac{\partial \mathcal{L}^{UCE}}{\partial \alpha_{c^*}} < 0.
> >    $
> >
> >    This proves that $\mathcal{L}^{UCE}$ is monotonically decreasing with respect to $\alpha_{c^*}$.
> >
> > **Intuition**
> >
> > The UCE loss leads the model to predict lower uncertainty values than it should. For instance, even with a perfect uncertainty prediction (e.g., a noisy training example correctly assigned non-zero aleatoric uncertainty), the UCE loss remains positive. Increasing the predicted evidence for the ground truth class further decreases the loss, resulting in overly high evidence for the ground truth class and, consequently, lower uncertainty values. This behavior fosters overconfidence.
> >
> > In Proposition 1, we prove that minimizing the UCE loss peaks the predicted Dirichlet distribution at the ground truth categorical distribution, which is typically a one-hot probability vector. This peaking effect leads to uncertainty predictions that are lower than the ground truth, further exacerbating the overconfidence issue.
> >
> > To address the limitations of the UCE loss, we propose the UFCE loss combined with evidence-regularized learning (ERL). As stated in Proposition 2, the UFCE loss prevents the model from overfitting by discouraging it from overly focusing on examples where it already predicts high projected class probabilities. ERL acts as a pseudo-ground truth for epistemic uncertainty prediction by encouraging the model to predict a uniform Dirichlet distribution for pseudo-OOD samples, thereby improving the quality of epistemic uncertainty estimates.
> >
> > Please refer to our answers to Q1 and Q2 of the reviewer CuPe about the detailed insights of our proposed UFCE loss function.
> >
> > **Q3 (b) Case Study on the Limitation of UCE Loss**
> >
> > We did not conduct a dedicated case study on the limitations of the UCE loss alone, but this will be an area for future work. In our manuscript, we include case studies comparing the proposed UFCE loss and the UCE loss, focusing on how the UFCE loss addresses the overconfidence issue inherent in the UCE loss. For more details, please refer to our responses to Q4 and Q7.

---

> ### Author Response · Authors · 2024-11-29
> **Response to Question 2,4,5,7**
>
> **Q2: Comparing ENN with Bayesian Neural Networks.**
>
> We adapt five classic uncertainty quantification baselines from general deep learning tasks into the uncertainty-aware BEVSS task in this paper: entropy, deep ensemble, Monte Carlo dropout, energy, and ENN. The second and the third methods are traditional Bayesian-based uncertainty quantification methods and the last two are more recent and advanced ones. We evaluate the performance on three tasks with 3 backbones across 3 popular datasets.
>
> We highlight the key conclusions from the three tasks as follows:
>
> 1. Pure Segmentation and Misclassification Detection (Table 1, Table 13 and Table 14):
>    The ENN performs on par with the baselines, with no single model consistently outperforming others across all metrics.
> 2. Calibration (Tables 1, Table 13 and Table 14):
>    Our proposed model achieves the lowest ECE score in 4 out of 6 configurations and the second-lowest in 1 configuration, highlighting its superior calibration performance.
> 3. OOD Detection (Table 2 and Table 13):
>    * Without pseudo-OOD exposure: All models, including Bayesian methods and ENN, exhibit poor OOD detection performance, as reflected by low AUPR values.
>    * With pseudo-OOD exposure: Only energy-based and evidential-based models can leverage pseudo-OOD data. In this setting, our framework achieves the highest OOD detection performance across 9 out of 10 metrics.
>
> In summary, our proposed model demonstrates comparable performance with baselines in segmentation and misclassification detection, superior calibration with the lowest ECE scores in most configurations, and achieves the highest OOD detection performance in 9 out of 10 metrics when leveraging pseudo-OOD data.
>
> **Q4 & Q7: Comparing Calibration Between UCE and UFCE Loss (case study).**
>
> Figures 1-3 (Page 5, 20, 21\) present case studies comparing UCE and UFCE in terms of the gradient norms of the last linear layer.
>
> Specifically, we examine the gradient norm differences between UFCE with varying $\gamma$ and UCE ($\gamma=0$) across different predicted probabilities for the ground truth class, while keeping the evidence supporting the ground truth class constant (testing several values of $\alpha_{c^*}$).
>
> The key observations are as follows:
>
> 1. Confident Samples: For samples with higher predicted probabilities for the ground truth class, the UFCE loss produces lower gradient norms compared to UCE (negative difference). This limits excessive evidence assignment to the ground truth class, reducing the risk of underestimating uncertainty and mitigating network overconfidence, resulting in more balanced uncertainty estimates.
> 2. Less Confident Samples: For samples with lower predicted probabilities, the UFCE loss results in higher gradient norms (positive difference), encouraging the model to focus on learning from these samples and accelerating the correction of underconfidence.
>
> **Q5: Inconsistencies Between UCE and UFCE in Table 1\.**
>
> The calibration performance is assessed using the ECE score, with calculation details provided in Appendix A.5. A lower ECE score reflects better calibration. As shown in Table 1, our proposed model achieves significantly lower ECE scores compared to the UCE loss on both LSS (0.00193 vs. 0.00342) and CVT (0.00019 vs. 0.0036), showing our proposed loss function has better calibration than the UCE loss.

---

> ### Author Response · Authors · 2024-11-29
> **Response to Question 6, 8 (part 1)**
>
> **Q6: Improving the Clarity and Organization of Experimental Results and Tables.**
>
> Thanks for the suggestion. In our revised PDF, we revise the experiment section and organize the result tables in the appendix.
>
> In the main paper, we present results on nuScenes using LSS and CVT as backbones, covering predicted segmentation and aleatoric uncertainty (Table 1), epistemic uncertainty (Table 2), running time (Table 3), robustness analysis (Table 4), and ablation studies (Table 5). In the appendix, we provide results on CARLA and Lyft using all backbones including LSS, CVT and Simple-BEV (Tables 13–16). Additionally, we include hyperparameter analysis (Tables 7–9) and robustness experiments (Tables 17–23), which encompass detailed results on corrupted nuScenes-C, diverse town and weather conditions in CARLA, and pseudo-OOD selections. Qualitative comparisons are provided in Figures 4–6.
>
>
> **Q8 (a): Comparison with the most recent BEV Methods.**
>
> We appreciate the reviewer’s feedback on the selected network backbones. BEV-view semantic segmentation tasks in the literature are significantly different from image semantic segmentation tasks. BEV view is a virtual view around the ego vehicle generated based on the fusion of multiple camera images. One of the unique challenges in BEV-view segmentation is the need to learn a mapping from pixels in different camera reviews to pixels in the BEV view.
>
> The **LSS** and **CVT** are two representative BEVSS (BEV semantic segmentation) backbones that rely on two representative ways to learn the mapping. In particular,  **LSS** converts raw camera inputs into BEV representations by predicting depth distributions, constructing feature frustums, and rasterizing them onto a BEV grid, while **CVT** employs a transformer-based approach with cross-attention and camera-aware positional embeddings to align features into the BEV space. Due to the different ways of learning the mapping, their neural network architectures are significantly different, accordingly.
>
> In our updated PDF, we have incorporated new experiments using a more recent BEVSS backbone, **Simple-BEV** [1] (ICRA 2023). In contrast to LSS and CVT, **Simple-BEV** bypasses depth estimation entirely, projecting 3D coordinate volumes onto camera images to sample features, emphasizing efficiency and robustness to projection errors. Detailed descriptions can be found in Appendix A.2.1. We selected Simple-BEV over the state-of-the-art model PointBEV [2] (CVPR 2024\) with public code due to its higher adoption and reliability: Simple-BEV has 90 citations and 496 GitHub stars, compared to PointBEV’s 6 citations and 94 stars. This made Simple-BEV a safer and more reliable choice for obtaining robust results within our limited time frame. Furthermore, as shown in Table 1 of the PointBEV paper, Simple-BEV achieves runner-up performance, outperforming other recent backbones, such as BEVFormer [3] (ECCV 2022\) and BAEFormer [4] (CVPR 2023), further supporting its inclusion in our study.
>
> In summary, we adopt a recent BEVSS backbone, Simple-BEV [1], into our comparison on the nuScenes dataset. The results are presented in current Table 16\. Based on observations from Table 1, Table 2, and Table 16, we conclude the following:
>
> * While using Simple-BEV as backbone delivers better segmentation performance compared to using LSS or CVT, no significant improvement in OOD detection performance is observed.
> * Our proposed model achieves the best performance in calibration (in 2 out of 3 evaluations) and OOD detection (in 7 out of 9 evaluations) across the three backbones, and presents comparable performance in misclassification detection.
>
> To ensure concision and save space for addressing other important comments, we briefly summarize the main points here and kindly ask that please refer to our detailed response to W3 of Reviewer QCfP for additional details.
>
> References:
>
> [1]: Harley, Adam W., et al. "Simple-bev: What really matters for multi-sensor bev perception?." *2023 IEEE International Conference on Robotics and Automation (ICRA)*. IEEE, 2023\.
>
> [2]: Chambon, Loick, et al. "PointBeV: A Sparse Approach for BeV Predictions." *Proceedings of the IEEE/CVF Conference on Computer Vision and Pattern Recognition*. 2024\.
>
> [3] Zhiqi Li, Wenhai Wang, Hongyang Li, Enze Xie, Chonghao Sima, Tong Lu, Yu Qiao, and Jifeng Dai. BEVFormer: Learning bird’s-eye-view representation from multi-camera images via spatiotemporal transformers. In ECCV, 2022\.
>
> [4] Cong Pan, Yonghao He, Junran Peng, Qian Zhang, Wei Sui, and Zhaoxiang Zhang. BAEFormer: Bi-directional and early interaction transformers for bird’s eye view semantic segmentation. In CVPR, 2023

---

> ### Author Response · Authors · 2024-11-29
> **Response to Question 8 (part 2)**
>
> **Q8 (b) Comparison with other robust BEV segmentation models.**
>
> **We answer this question on the following two items:**
>
> **(1) Robustness evaluation of our model:** (In the original version,) We evaluate the robustness of our model in terms of BEV backbones and pseudo-OOD selections. To provide a more throughout robustness evaluation, we extend the robustness analysis in the revised manuscript to include:
>
> * **Dataset corruption** (Table 19). We test our model on the nuScenes-C dataset [1], which applies seven types of corruption (e.g., environmental dynamics) at three levels of severity to the nuScenes validation set. This dataset is specifically designed for robustness evaluation. We compare our model with the two best-performing baselines in terms of OOD detection performance on the clean nuScenes dataset.
> * **Diverse weather conditions and urban layouts** (Table 20-23). We test our model across four distinct urban layouts, encompassing both dense and sparse scenarios, and under 10 different weather conditions, varying in brightness and wetness.
>
> **(2) Comparing to BEV models designed for robustness improvement:** We would like to clarify the difference in the objective between our proposed model (uncertainty-aware BEVSS model) and traditional BEVSS models (referred to as "backbones"). While traditional BEVSS models output only segmentation predictions, our proposed model provides segmentation predictions paired with multidimensional uncertainties related to the predictions or data. Our evaluation emphasizes both segmentation performance and the quality of the predicted uncertainties, with our primary contribution being effective uncertainty quantification. Consequently, evaluating the robustness of uncertainty prediction, rather than solely segmentation robustness, is more appropriate for our model.
>
> We acknowledge the existence of methods designed for improving the robustness of BEVSS models and note that adapting these strategies to our model is infeasible. For instance, [1] enhances robustness through corruption-based data augmentation and fine-tuning within a CLIP [2] representation space, demonstrating that data augmentation statistically improves robustness. However, applying the same strategy in our case is infeasible, as data augmentation targeting true OOD types is impossible by definition, given that these represent unseen and unknown samples. Instead, our proposed method leverages pseudo-OOD data to enhance the quality of epistemic uncertainty prediction.

---

> ### Author Response · Authors · 2024-11-29
> **Response to weakness**
>
> In the list of weaknesses, we clarify the issues uncovered by the above questions.
>
> **W1:Invesigation of the downstream tasks**
>
> This work focuses on uncertainty quantification in BEVSS, estimating multi-dimensional uncertainties for pixels in the BEV space.
>
> In this context, misclassification detection and Out-of-Distribution (OOD) detection represent key applications of aleatoric and epistemic uncertainty estimation, respectively, in deep learning. Both aspects are thoroughly explored in our experiments.
>
> BEVSS provides spatial and semantic understanding of the driving scene, forming a critical foundation for downstream tasks like motion planning and decision-making. Leveraging uncertainties in the BEV space to enhance the performance and reliability of these downstream tasks is an exciting avenue for future exploration.
>
> Additionally, investigating the uncertainties of the downstream tasks themselves poses valuable research questions. These could involve either fusing uncertainties from the BEV space or estimating uncertainties directly within the downstream tasks, both of which represent promising directions.
>
> Thank you for your insightful suggestion. We agree that exploring the impact of BEVSS uncertainty on downstream autonomous driving tasks is an important direction for future research.
>
>
> **W2\&W3:Uncertainty quantification baselines**
>
> In our study, we introduced the first benchmark for evaluating uncertainty quantification methods in Bird’s Eye View Semantic Segmentation (BEVSS), analyzing five representative approaches: softmax entropy, energy-based methods, deep ensembles, dropout, and evidential neural networks (ENNs) across three popular datasets and three representative BEVSS backbones. Through empirical comparisons, we identified ENNs as the most promising in terms of both effectiveness and efficiency, particularly for Out-of-Distribution (OOD) detection in BEVSS. This finding motivated the development of our proposed model, which builds upon the ENN framework. Please kindly refer our response to Q1 of reviewer CuP for more details.
>
> To improve clarity, we distinguish between two key concepts in the revised Section 4:
>
> 1. Uncertainty Quantification Baselines:
>    As there is no prior work on epistemic and aleatoric uncertainty quantification specifically for BEVSS, we adopt UQ methods from traditional deep learning as baselines for this task:
>    * Entropy (Confidence): A simple yet widely used method in complex applications like the LLM.
>    * Energy Model: A post-hoc processing method applied to predicted logits, known for its adaptability and competitive OOD detection performance.
>    * Bayesian-Based Methods: Includes Ensemble and Dropout, both proven effective in diverse applications. Ensembles involve training multiple independent models, while Dropout requires multiple forward passes during inference.
>    * ENN Model: Recognized for its computational efficiency and interpretability, it generates multinomial uncertainties and serves as the foundation for our proposed UFCE loss.
> 2. BEVSS Backbones:
>    These are the architectures used for bird’s-eye view semantic segmentation, such as LSS, CVT, and SimpleBEV (supplementary experiments during the rebuttal phase). For the entropy, energy, dropout, and ensemble baselines, we use the corresponding backbone architecture and optimization strategy, varying only the uncertainty computation method.
>    For the ENN model, we modify the backbone by replacing the final activation function with ReLU and optimize it using either the UCE or our proposed loss.

---

> ### Comment · Area_Chair_9zvT · 2024-12-01
>
> Dear Reviewer,
>
> Authors apparently did a responsible rebuttal; Could you please look at if your concerns have been addressed?
>
> AC

---

> ### Comment · Reviewer_6JWb · 2024-12-02
>
> Dear authors,
>
> Thank you for providing a detailed response to my previous questions and concerns. I believe that a large effort has been spent on the re-organization of the revised paper, the new experiments, and the analysis. However, there are still some questions that remain unclear.
>
> The authors have added the Simple-BEV method for comparison. However, the comparison with stronger or more recent state-of-the-art methods is still limited. For example, how about the comparison with the MetaBEV (ICCV2023), X-Align (WACV2023), X-Align++ (Journal version), UniTR (ICCV2023), to list a few, under the uncertainty-aware BEVSS task?
>
> A comparison with other loss functions is expected for the RQ2. The claim of the authors that the focal loss function is generally better or consistently performs better than ce loss, is not a very new observation in general segmentation. To support the observation of the loss function analysis, it is better to conduct a detailed comparison with other different loss functions from different perspectives, such as Focal Tversky loss, PolyLoss (ICLR202), or the cross-modal feature alignment (X-FA) loss from the X-Align method.
>
> In my previous question Q5, my concern that the proposed UFCE or focal loss is consistently better than the CE loss is not addressed. For example, w/o pseudo OOD, the runner-up method and the Ensemble with focal have lower ECE scores as compared with the ENN with UFCE.
>
> Besides, there are a few questions for the revised version.
>
> “Without pseudo-OOD, the proposed UFCE loss for the ENN model consistently outperforms the commonly used UCE loss across all metrics, showing average improvements of 0.0014 in calibration ECE” It is mentioned in the caption of Table 1, but how can the improvement score calculated? I just got 0.00429 - 0.00332 = 0.00097. Did I misunderstand the comparison? Also for the AUROC, AUPR, and FPR95.
>
> Apart from the ECE scores, the IoU scores are very similar or the IoU improvements over the baselines are marginal. For example, w/o pseudo OOD, UCE vs. UFCE is 0.341 vs. 0.343 based on LSS, or 0.291 vs. 0.319 based on CVT. Also for the cases with pseudo OOD. How does this minor change in the output results affect the downstream task or applications?

---

> ### Author Response · Authors · 2024-12-02
> **Response to Q1(new) and Q2(new)**
>
> **Q1 (new): Comparison with stronger or more recent state-of-the-art methods is still limited**
>
> Thank you for the suggestion. In our study, we focus on BEV semantic segmentation (BEVSS) tasks using only camera RGB images as input, without incorporating LiDAR sensor data, as described in our problem formulation in Section 2. This is aligned with practical systems like the full self-driving system of Tesla, which relies exclusively on cameras to perceive the environment.  The three suggested models (MetaBEV, X-Align, and UniTR) utilize both camera images and LiDAR point clouds, making their settings fundamentally different from ours.
>
> For comparison, we consider **PointBEV** (We made a typo in our previous response, incorrectly citing it as CVPR 2023 instead of **CVPR 2024**), the most recent state-of-the-art model for BEVSS using only camera images. However, due to time and computational resource constraints, we implemented the runner-up model **Simple-BEV** (ICRA2023) reported in PointBEV, as it has higher recognition with 90 citations and 496 GitHub stars compared to PointBEV's 6 citations and 94 stars.  As a reference, Simple-BEV outperforms other recent backbones, such as **BEVFormer** (ECCV 2022) and **BAEFormer** (CVPR 2023).
>
> It is important to note that our primary objective is not to propose a new BEVSS model to outperform existing methods on segmentation accuracy. Instead, we aim to study **epistemic and aleatoric uncertainty quantification techniques** for existing BEVSS models, which is a critical, underexplored safety-related area in the literature. We selected **LSS**, **CVT**, and **Simple-BEV** as they represent three distinct approaches to solving the unique challenge of mapping pixels from multiple camera views to pixels in the BEV view.
>
> * **LSS** converts raw camera inputs into BEV representations by predicting depth distributions, constructing feature frustums, and rasterizing them onto a BEV grid.
> * **CVT** employs a transformer-based approach with cross-attention and camera-aware positional embeddings to align features into the BEV space.
> * In contrast, **Simple-BEV** bypasses depth estimation entirely by projecting 3D coordinate volumes onto camera images to sample features, prioritizing efficiency and robustness to projection errors.
>
> For additional context, our response to **Reviewer DVo9's Q2** ("Why LSS and CVT as backbones?") provides complementary insights.
>
> Lastly, we appreciate your suggestion and agree that studying uncertainty quantification techniques for BEVSS models utilizing both camera and LiDAR sensors is a promising direction for future work.
>
> **Q2 (new): A comparison with other loss functions is expected for the RQ2**
>
> First, it is important to distinguish **ENNs**—for which our proposed UFCE loss is specifically designed—from traditional **softmax-based models**, for which the original focal loss and many other segmentation loss functions were developed. Softmax-based models produce deterministic class probability vectors through a softmax activation function, which are directly optimized against true class labels. In contrast, ENNs predict a **Dirichlet distribution** over class probability vectors, capturing both **aleatoric** and **epistemic uncertainties**.
>
> This fundamental difference makes loss functions like the original focal loss incompatible with ENN, as they cannot account for the stochasticity introduced by the Dirichlet distribution. Extending other softmax-tailored loss functions (e.g., Focal Tversky Loss) to ENNs and analyzing their theoretical properties is non-trivial. While this is an interesting direction, it falls outside the scope of this work. We agree it warrants further exploration and will address it in future work.
>
> In the existing literature, ENNs are primarily optimized using the **UCE loss**, which minimizes the expected cross-entropy under the predicted Dirichlet distribution. While effective, UCE loss often results in overconfident predictions. To address this limitation, we propose the **UFCE loss**, which integrates the adaptive weighting mechanism of focal loss into the ENN framework. UFCE loss is defined as the expectation of focal loss under the predicted Dirichlet distribution, allowing it to dynamically emphasize challenging or misclassified samples while mitigating the gradient contribution of confidently predicted ones.
>
> In our last response to Reviewer DVo9, we outlined the relationships between four key loss functions—**CE**, **focal loss**, **UCE**, and our proposed **UFCE loss**. We also explained why traditional loss functions, such as CE and focal loss, are incompatible with ENNs due to the stochastic nature of the predicted class probability vector in ENNs. The same reasoning applies to other softmax-based loss functions, such as **Focal Tversky Loss**, **PolyLoss**, and **X-FA loss**, which cannot be directly applied to ENNs without substantial modification.
>
> We will clarify these points in the revised version.

---

> ### Author Response · Authors · 2024-12-02
> **Response to Q3(new), Q4(new), Q5(new)**
>
> **Q3 (new)**: In my previous question Q5, my concern that the proposed UFCE or focal loss is consistently better than the CE loss is not addressed. For example, w/o pseudo OOD, the runner-up method and the Ensemble with focal have lower ECE scores as compared with the ENN with UFCE.
>
> **Our response**:
>
> The statement we made is that Focal-based loss functions have better calibration (lower ECE score) than the CE-based loss functions with respect to the same model (ensemble, dropout…) and backbone (LSS or CVT). Take Table 1 as an example, Ensemble-Focal has a lower ECE (0.00233 for LSS and 0.00243 for CVT) than that of the Ensemble-CE (0.00569 for LSS and 0.00276 for CVT), ENN-UFCE has lower ECE (0.00332 for LSS and 0.00190 for CVT) than that of the ENN-UCE (0.00429 for LSS and 0.00371 for CVT). As CE and Focal loss functions are not valid loss functions for ENNs (as discussed in our response to Q2 (new)), we mainly compare UFCE with UCE to demonstrate the effectiveness of our proposed UFCE loss over the traditional UCE loss for training ENNs for epistemic and aleagoric uncertainty quantification. Thanks for your suggestion in your review, we will provide some tables that focus on comparing UCE and UFCE.
>
> **Q4 (new)**: Without pseudo-OOD, the proposed UFCE loss for the ENN model consistently outperforms the commonly used UCE loss across all metrics, showing average improvements of 0.0014 in calibration ECE” It is mentioned in the caption of Table 1, but how can the improvement score calculated? I just got 0.00429 \- 0.00332 \= 0.00097. Did I misunderstand the comparison? Also for the AUROC, AUPR, and FPR95.
>
> **Our response**:
>
> The **“average improvements”** numbers for ECE, AUROC, AUPR, and FPR95 are based on the average across the LSS and CVT backbones. Specifically, for ECE, the improvement between ENN-UFCE and ENN-UCE without pseudo-OOD is calculated as follows: ((0.00429 \- 0.00332) \+ (0.00371-0.0019))/2 \= 0.0014.
>
> We will clarify that the reported improvements are averaged across these two backbones in our future revised version.
>
> **Q5 (new)**: Apart from the ECE scores, the IoU scores are very similar or the IoU improvements over the baselines are marginal. For example, w/o pseudo OOD, UCE vs. UFCE is 0.341 vs. 0.343 based on LSS, or 0.291 vs. 0.319 based on CVT. Also for the cases with pseudo OOD. How does this minor change in the output results affect the downstream task or applications?
>
> **Our response**:
>
> Thank you for highlighting this observation. The similarity in IoU scores or marginal improvements over the baselines is expected and aligns with our objective. Our primary goal is to demonstrate that the trained ENN model achieves segmentation accuracy comparable to baseline methods, in addition to our enhancements on aleatoric and epistemic uncertianty estimation (the focus of this work).
>
> It is important to note that the ENN architecture replaces the standard softmax activation function in the final layer with a ReLU function to predict the parameters of the **Dirichlet distribution** for pixel-wise classification. In our approach, ENN model parameters are optimized using the **UFCE loss function**, alongside two regularization terms: **Evidence Regularization (ER)** and **Epistemic Uncertainty Scaling (EUS)** (as detailed in Section 3). While the predicted pixel-level class probability vectors in the BEV view are not identical to those of the baselines, they maintain comparable segmentation accuracy, ensuring no degradation in IoU performance.
>
> Traditional BEVSS models output only a probability map in the BEV view. In contrast, our ENN models, built on these BEVSS backbones, generate:
>
> 1. A similar probability map (with comparable IoU scores).
> 2. An **epistemic uncertainty map**.
> 3. An **aleatoric uncertainty map** in the BEV view.
>    This functionality is demonstrated in the accompanying diagram of our framework: [link to diagram](https://drive.google.com/file/d/1cTDk4VGpcXGH9gwhd7MfH-yNu5ymXwo_/view?usp=sharing).
>
> For downstream tasks or applications that rely solely on the probability map, our ENN models provide identical utility with no adverse impact. However, the additional uncertainty maps enable **safety-critical tasks and applications**, such as:
>
> * **Visualization of high-uncertainty areas** to alert users to potential risks.
> * **Routing or decision-making to avoid high-uncertainty regions**, enhancing overall safety and reliability.
>
> These added capabilities, supported by robust uncertainty quantification, represent a key contribution of our work and address an important gap in the current literature.

---

> ### Author Response · Authors · 2024-12-04
> **Response to Q1(new) - Additional**
>
> Dear Reviewer 6JWb,
>
> We sincerely thank you for sharing your questions and concerns. We hope our responses have sufficiently addressed them. We would greatly appreciate it if you could consider updating your rating in light of our new results and clarifications.
>
> Regarding your suggestion to include models that utilize both camera and LiDAR sensors as input, we would like to provide further clarification why these models should belong to a different benchmark from our proposed benchamk for camera-only BEVSS models. Models incorporating both image and LiDAR data face distinct challenges, such as handling the multi-modality of the two sensor types. However, the mapping problem from camera to BEV spaces is generally less challenging for camera-LiDAR fusion models compared to camera-only models, as the LiDAR sensor provides 3D point cloud data that aids in establishing these mappings.
>
> In contrast, BEVSS models relying solely on camera images—such as those studied in our submission—face the unique challenge of learning these mappings without 3D point cloud data. This makes the BEVSS problem more complex for camera-only models. The three backbones (LSS, CVT, Simple-BEV) that we considered represent three representative ways to address the mapping challenge as discussed in our answer to Q8 (a).
>
> Typically, models with different input sensor configurations are benchmarked separately in the literature. For example, the BEV-related benchmarking paper [1], as suggested by Reviewer CuPE, introduces camera-only models and camera-LiDAR fusion models in separate sections: Section 5.2 ("Camera-Only Benchmark") and Section 5.3 ("Camera-LiDAR Fusion Benchmark"). This separation acknowledges the distinct challenges and evaluation scenarios associated with these two categories.
>
> We agree that studying uncertainty quantification techniques for BEVSS models utilizing both camera and LiDAR sensors is an exciting and promising direction for future work. However, in alignment with established benchmarks, we believe such a study should be conducted separately from our current benchmark study. These two categories of models address fundamentally different challenges and require sufficiently distinct network architectures to handle multi-modality effectively.
> Thank you once again for your thoughtful feedback and the opportunity to clarify these points.
>
> References:
>
> [1] Xie, Shaoyuan, Lingdong Kong, Wenwei Zhang, Jiawei Ren, Liang Pan, Kai Chen, and Ziwei Liu. "Benchmarking and Improving Bird's Eye View Perception Robustness in Autonomous Driving." arXiv preprint arXiv:2405.17426 (2024).

---

### Official Review · Reviewer_DVo9 · 2024-11-02

**Soundness:** 2
**Presentation:** 2
**Contribution:** 2
**Rating:** 6
**Confidence:** 3

**Summary:**

Summary:

This paper is motivated to explored the uncertainty estimation in bird’s-eye-view segmentation task, as uncertainty estimation can be utilized to handle out-of-distribution scenarios and adapt to unexpected conditions, especially in dynamic driving environments.. The contribution is two-folds:

1. Testing of Uncertainty Quantification: The paper evaluates predictive uncertainty quantification methods in Bird's Eye View (BEV) semantic segmentation with existing datasets, highlighting the model’s ability to detect misclassified and out-of-distribution (OOD) pixels. This incorporates five existing approaches across multiple backbones.
2. New Loss Function – UFCE: The authors claim a novel Uncertainty-Focal-Cross-Entropy (UFCE) loss function. This loss function is designed to handle the challenges of highly imbalanced data, improving both model calibration and the reliability of uncertainty predictions for BEV segmentation tasks. The UFCE loss, inspired by Focal Loss, enhances the model’s focus on uncertain predictions, leading to better aleatoric (data-related) and epistemic (model-related) uncertainty estimation.
3. Improvement in Model Calibration and OOD Detection: The proposed UFCE loss function, coupled with an uncertainty-scaling regularization term, shows superior results over traditional methods, especially in improving the AUPR (Area Under Precision-Recall curve) for OOD detection by approximately 4.758%. The framework achieves high accuracy in segmentation while also performing well in detecting misclassifications and OOD pixels, thus bolstering both model reliability and safety for autonomous vehicle applications.

**Strengths:**

Strength

1. Effectiveness Across Datasets and Models: Through extensive experiments, the paper demonstrates that the proposed uncertainty quantification framework performs robustly across various datasets (CARLA, nuScenes, Lyft) and model architectures (Lift-Splat-Shoot, Cross-View Transformer), outperforming existing methods on key metrics like AUROC (Area Under ROC Curve) and Expected Calibration Error (ECE).
2. Computational Efficiency: The proposed Evidential Neural Networks (ENN) with UFCE loss achieves comparable or improved results with lower computational complexity compared to methods requiring multiple forward passes, making it practical for real-time applications.
3. Methodical Analysis: The study rigorously evaluates aleatoric and epistemic uncertainties, using metrics like Area Under the ROC Curve (AUROC) and Area Under Precision-Recall Curve (AUPR). Also a runtime efficiency comparison is included as well. This multi-faceted approach allows for a clearer understanding of each method's effectiveness.
4. Real-world Relevance: By focusing on predictive uncertainty in the context of autonomous vehicles, the research addresses a critical area of real-world application where model reliability and safety are paramount.

**Weaknesses:**

Weakness:

1. This paper looks more than a study than a question-motivated paper. The benchmark part utilizes existing dataset as well as existing methods and no clear conclusion is drawn from the study. The proposed new loss and the regularization terms are not quite related to the study in terms of motivation. The author is encouraged to explain more on the logical motivation of the proposed new loss function part.
2. Missing details in experiments part. For different datasets, they have different sensor setting (camera position on the vehicle, FOV, number of camera, number of beams for LiDAR) as well as the scale of bird’s-eye-view grid (0.25m or 0.5m or 1m?). It would be great if the authors could provide a table summarizing the sensor configurations and BEV grid scales for each dataset used in the experiments.
3. Some typos. For example, L49: Aleatoric uncertainty, which arises from inherent randomnesses, such as noisy data and labels. **not sure if it's missing Subject–verb–object or Subject-Link verb-Predicative Structure** L075: We propose the UFCE loss, which we theoretically **demonstrate can** implicitly regularize sample weights, mitigating both under-fitting and over-fitting.

**Questions:**

Question

1. What is the reason of choosing such pseudo-OOD classes in those dataset? Any statistical supporting evidence here? Or any quantitative justification for the choice of pseudo-OOD classes, such as similarity metrics between the chosen classes and true OOD examples?
2. Also the reason of choosing LSS and CVT as baselines. These two methods are back in 2022 and there are many more advanced BEV segmentation models such as BEVSegFormer[1], PETR-V2[2], and so on. It is suggested that authors either include more recent baselines in their comparison or provide a clear justification for why LSS and CVT were chosen despite the availability of newer methods.

[1] Peng, Lang, Zhirong Chen, Zhangjie Fu, Pengpeng Liang, and Erkang Cheng. "Bevsegformer: Bird's eye view semantic segmentation from arbitrary camera rigs." In Proceedings of the IEEE/CVF Winter Conference on Applications of Computer Vision, pp. 5935-5943. 2023.

[2] Liu, Yingfei, Junjie Yan, Fan Jia, Shuailin Li, Aqi Gao, Tiancai Wang, and Xiangyu Zhang. "Petrv2: A unified framework for 3d perception from multi-camera images." In Proceedings of the IEEE/CVF International Conference on Computer Vision, pp. 3262-3272. 2023.

---

> ### Comment · Reviewer_DVo9 · 2024-11-27
> **Waiting for author's rebuttal**
>
> Since the authors still didn't provide a rebuttal, I have to keep the current rating for this submission.

---

> > ### Author Response · Authors · 2024-11-27
> >
> > Dear reviewer DVo9,
> >
> > Thank you for your valuable feedback and insightful suggestions. We are grateful for the six-day extension of the discussion period. We are actively conducting additional experiments to demonstrate the effectiveness and robustness of our work across more benchmarks and backbones. As some of these experiments are time-intensive, we completed the collection of all empirical results yesterday. We plan to submit the revised PDF tomorrow, including the requested empirical results, necessary clarifications, and a detailed rebuttal addressing all reviewers' concerns. We look forward to further discussions to ensure we have thoroughly addressed your feedback.

---

> ### Author Response · Authors · 2024-11-29
> **Response to Question 1**
>
> **Q1: Justifying the Choice of Pseudo-OOD Classes.**
>
> We appreciate the reviewer’s feedback and would like to clarify the following related issues: (a) the selection criteria used to select the pseudo-OOD data, (b) the robustness of the model performance to different choices of pseudo-OOD data, and (c) the robustness of the model performance on the similarity between the chosen pseudo-OOD classes and true OOD examples.
>
> **(a) Criteria used to select the pseudo-OOD data**
>
> We initially adhered to the criteria outlined in the benchmark study by Franchi et al. [1], a seminal work in the field of autonomous driving, to identify candidate OOD classes. In this context, OOD data typically encompasses less frequently encountered dynamic objects (e.g., motorcycles, bicycles, bears, horses, cows, elephants) and static objects (e.g., food stands, barriers) that are distinct from primary segmentation categories like vehicles, road regions, and pedestrians. These objects were designated as candidate OOD classes. Subsequently, in our experiments, we randomly partitioned these candidate classes into pseudo-OOD and true OOD categories.
>
> It is worth noting that the OOD benchmark dataset MUAD (https://muad-dataset.github.io/) provided by Franchi et al. [1] was collected using a simulator based on front-camera imagery for image segmentation. However, it does not include a collection of images from multiple cameras or labels for BEV segmentation. To address this, we adopted a similar procedure and generated a BEV segmentation dataset with OOD objects using the well-established CARLA simulator.
>
> Reference:
> [1]: Franchi, Gianni, et al. "MuAD: Multiple uncertainties for autonomous driving, a benchmark for multiple uncertainty types and tasks." arXiv preprint arXiv:2203.01437 (2022).
>
> **(b) Robust of the model performance to different choices of pseudo-OOD data.**
>
> For the nuScenes and Lyft datasets, we adopted the following configuration for both datasets: “motorcycle” was designated as the true OOD class, while “bicycle” served as the pseudo-OOD class.
>
> To evaluate the sensitivity of model performance to different choices of pseudo-OOD and true OOD classes, we introduced an additional configuration for the nuScenes dataset: “traffic cones, pushable/pullable objects, and motorcycles” were selected as pseudo-OOD classes, and “barriers” were designated as the true OOD class.
>
> For the CARLA dataset, we utilized the simulator to randomly position various animal objects (e.g., deer, bears, horses, cows, elephants) on road regions across different towns. We tested two distinct configurations: (1) “deer” as the true OOD class and “bears, horses, cows, elephants” as pseudo-OOD classes; and (2) “kangaroo” as the true OOD class and “bears, horses, cows, elephants, deer” as pseudo-OOD classes.
>
> Across all configurations, our empirical findings remained consistent, indicating robustness in model performance irrespective of the specific choices of pseudo-OOD data.
>
> **(c) Robustness of the model performance on the similarity between the chosen pseudo-OOD classes and true OOD examples.**
>
> We discussed this issue in the last paragraph of Section 4 in the revised PDF (as well Appendix A.4). Intuitively, greater similarity between true and pseudo-OOD pairs enhances OOD detection performance while overfitting to pseudo-OODs raises concerns about the model’s generalization ability. To further investigate, we conducted experiments using dissimilar pseudo-OOD and true OOD pairs compared to the default setting (Table 2). The results for nuScenes are shown in Table 4, while the results for CARLA are provided in Table 17 in Appendix A.5.5. The findings confirm our intuition: higher similarity between true and pseudo-OOD pairs leads to better OOD detection performance. Notably, our proposed model consistently outperformed the best baseline methods across all eight scenarios, achieving improvements of up to 12.6% in AUPR. These results underscore the robustness of our approach, even in settings with less similar OOD pairs.

---

> ### Author Response · Authors · 2024-11-29
> **Response to Question 2**
>
> **Q2: Why LSS and CVT as backbones?**
>
> BEV-view semantic segmentation tasks in the literature are significantly different from image semantic segmentation tasks, where traditional (e.g, drop-out, deep ensembles) and recent (e.g., energy-based, evidential deep learning) uncertainty quantification methods been explored. BEV view is a virtual view around the ego vehicle generated based on the fusion of multiple camera images. One of the unique challenges in BEV-view segmentation is the need to learn mapping from pixels in different camera reviews to pixels in the BEV view.
>
> The **LSS** and **CVT** are two representative BEVSS (BEV semantic segmentation) backbones that rely on two representative ways to to learn the mapping. In particular,  **LSS** converts raw camera inputs into BEV representations by predicting depth distributions, constructing feature frustums, and rasterizing them onto a BEV grid, while **CVT** employs a transformer-based approach with cross-attention and camera-aware positional embeddings to align features into the BEV space. Due to the different ways in learning the mapping, their neural network architectures are significantly different, accordingly.
>
> We appreciate the reviewer’s suggestion on other backbone architectures: BEVSegFormer [1], PETR-V2[2]. BEVSegFormer is indeed a related backbone designed for the BEVSS task, but the authors did not publish their code. PETR-V2 is a framework designed to detect 3D objects based on the fusion of multiple cameras, but for the task of BEV semantic segmentation. We studied the architecture of  PETR-V2, but found that the adaption of PETR-V2 to conduct the task of BEV semantic segmentation is non-trivial.
>
> In our updated PDF, we have incorporated new experiments using a more recent BEVSS backbone, **Simple-BEV** [1] (ICRA 2023). In contrast to LSS and CVT, **Simple-BEV** bypasses depth estimation entirely, projecting 3D coordinate volumes onto camera images to sample features, emphasizing efficiency and robustness to projection errors. Detailed descriptions can be found in Appendix A.2.1. We selected Simple-BEV over the state-of-the-art model PointBEV [2] (CVPR 2024\) with public code due to its higher adoption and reliability: Simple-BEV has 90 citations and 496 GitHub stars, compared to PointBEV’s 6 citations and 94 stars. This made Simple-BEV a safer and more reliable choice for obtaining robust results within our limited time frame. Furthermore, as shown in Table 1 of the PointBEV paper, Simple-BEV achieves runner-up performance, outperforming other recent backbones, such as BEVFormer [3] (ECCV 2022\) and BAEFormer [4] (CVPR 2023), further supporting its inclusion in our study.
>
> The results based on Simple-BEV are presented in Table 16 and demonstrate consistent performance when compared to LSS and CVT as backbones. While our model with Simple-BEV as backbone delivers better segmentation performance compared to LSS and CVT as backbones, no significant improvement in OOD detection performance was observed. For our proposed model, we observed consistent results regardless of whether LSS, CVT, or Simple-BEV were used as the backbone. Our model consistently achieves the best performance in calibration and OOD detection while maintaining comparable performance in misclassification detection.
>
> References:
>
> [1]: Harley, Adam W., et al. "Simple-bev: What really matters for multi-sensor bev perception?." *2023 IEEE International Conference on Robotics and Automation (ICRA)*. IEEE, 2023\.
>
> [2]: Chambon, Loick, et al. "PointBeV: A Sparse Approach for BeV Predictions." *Proceedings of the IEEE/CVF Conference on Computer Vision and Pattern Recognition*. 2024\.
>
> [3]：Zhiqi Li, Wenhai Wang, Hongyang Li, Enze Xie, Chonghao Sima, Tong Lu, Yu Qiao, and Jifeng Dai. BEVFormer: Learning bird’s-eye-view representation from multi-camera images via spatiotemporal transformers. In ECCV, 2022\.
>
> [4]: Cong Pan, Yonghao He, Junran Peng, Qian Zhang, Wei Sui, and Zhaoxiang Zhang. BAEFormer: Bi-directional and early interaction transformers for bird’s eye view semantic segmentation. In CVPR, 2023

---

> ### Author Response · Authors · 2024-11-29
> **Response to Weakness**
>
> **Weakness 1: About the motivation of this paper.**
>
> Thank you for the opportunity to elaborate on the logical motivation behind the proposed new loss function.
>
> In our study, we introduced the first benchmark for evaluating uncertainty quantification methods in BEVSS, analyzing five representative approaches: softmax entropy, energy, deep ensemble, dropout, and evidential neural networks (ENN), across three popular datasets. Among these, we found ENNs to be the most promising in terms of both effectiveness and efficiency for BEVSS tasks. However, traditional ENNs are typically trained using the Uncertainty-Cross-Entropy (UCE) loss, which is defined as the expectation of the standard cross-entropy loss. Similar to cross-entropy loss, UCE loss is not well-suited for handling highly imbalanced datasets, such as BEVSS data, where pixels belonging to rare classes (e.g., vehicles) are vastly outnumbered by those of dominant classes (e.g., the background).
>
> Motivated by this observation and the focal loss for addressing class imbalance [1] , we proposed a novel loss function, the **Uncertainty-Focal-Cross-Entropy (UFCE)** loss, defined as the expected focal loss (Equation 8, Page 5). In **Section 3.2**, we provide a theoretical analysis demonstrating that UFCE loss is more effective than UCE loss in addressing aleatoric uncertainty and improving calibration.
>
> Additionally, to further enhance epistemic uncertainty quantification, we introduced two regularization terms in **Section 3.3** for training the ENN model:
>
> 1. **Evidence Regularization (ER)**:
>    This term incorporates pseudo-OOD data during training, improving the quality of epistemic uncertainty predictions by guiding the model to better distinguish between in-distribution (ID) and out-of-distribution (OOD) samples.
> 2. **Epistemic Uncertainty Scaling (EUS)**:
>    This term directs the model’s focus toward samples with high epistemic uncertainty, helping to reduce false positives in OOD detection by emphasizing challenging samples.
>
> Empirical results in Section 4 demonstrate that the ENN model, trained with the UFCE loss and the two regularization terms (ER and EUS), consistently achieves superior epistemic uncertainty quantification compared to baseline methods, as evaluated through OOD detection. Furthermore, the model delivers top-tier aleatoric uncertainty performance, as measured by calibration and misclassification detection, while maintaining high segmentation accuracy.
>
> For additional insights related to the UFCE loss function and the regularization terms (ER and EUS), please refer to our response to Reviewer CuPe's related question (Q1).
>
> Reference:
>
> [1] Lin T. Focal Loss for Dense Object Detection. arXiv preprint arXiv:1708.02002. 2017. (over 30k citations)
>
> **Weakness 2: Experimental Setup Clarifications: Camera, LiDAR, and BEV Grid Scale**
>
> We added Table 10 in Appendix to show these details. Note that, in this paper, we consider BEV semantic segmentation tasks based on the fusion of camera RGB images as input. No LiDAR sensor data is considered in our experiments.
>
> **Weakness 3: Some typos.**
>
> Thanks for pointing out the typos. “Aleatoric uncertainty, which arises from inherent randomnesses, such as noisy data and labels” should be corrected as: “Aleatoric uncertainty arises from inherent randomnesses, such as noisy data and labels”. The second typo is corrected as: “We propose the UFCE loss and theoretically demonstrate that this novel loss function can implicitly regularize sample weights, effectively mitigating both underfitting and overfitting.” We apologize we forgot to correct the typos in our revised PDF. We will correct them in our future revised version.

---

> > ### Comment · Reviewer_DVo9 · 2024-12-02
> >
> > Thanks for the provided experiment and clarification on the motivation of the OOD benchmarking regarding my question 1 and 2. While for the weakness:
> >
> > 1. The focal loss is a well-known technique for addressing data-imbalance in computer vision, which I assume as a common knowledge. Applying this to BEVSS and your benchmark seems to be trivial contribution as stated in your logic. Though I admit the combination of that loss into specific task setting (BEVSS) requires effort, that didn't provide new knowledge to the community, so I assume that is not a core contribution in this paper.
> >
> > 2. Thanks for the table to clarify the setting, and this is of great importance to the performance of different models, whichi should be highlighted to avoid any confusion.
> >
> > Overall my previous idea of this paper haven't changed much, that it is more of a study paper. But with those clarification I think it is above the acceptance threshold and hope it could contribute to the community to encourage further insight into this problem. I will change my rating to 6.

---

> > > ### Author Response · Authors · 2024-12-02
> > > **Comparing proposed UFCE and Focal Loss**
> > >
> > > Thank you so much for your encouraging feedback. We will clarify the related points in our future revised version. We appreciate your acknowledgment of focal loss as a widely utilized technique for addressing data imbalance in computer vision tasks. Traditionally, focal loss is designed to operate with deterministic class probability vectors produced by softmax neural networks, where these vectors are directly compared with true class labels. However, adapting focal loss for evidential neural networks (ENNs) introduces a unique challenge: ENNs output a Dirichlet distribution over class probability vectors. This stochastic nature renders the original focal loss formulation unsuitable for direct optimization as a loss function.
> > >
> > > To overcome this limitation, we propose the **Uncertainty Focal Cross-Entropy (UFCE)** loss, which is defined as the expectation of focal loss under the Dirichlet distribution predicted by ENNs. Our work is the first to introduce this novel adaptation of focal loss specifically for ENNs, derive its analytical form, and demonstrate its theoretical and empirical advantages over the standard Uncertainty Cross-Entropy (UCE) loss. We believe the UFCE loss, with its theoretical foundations and empirical validation, offers significant insights and value to the research community. We apologize if these points were not sufficiently emphasized in the original submission and will ensure greater clarity in the revised manuscript.
> > >
> > > To illustrate the loss functions, consider the $i$-th pixel in the bird’s-eye view (BEV). Let $\mathbf{X}$ represent the input camera images, $\mathbf{p}$ denote the class probability vector, and $\mathbf{y}$ be the one-hot encoded true class label for the pixel. The ENN predicts a Dirichlet distribution $\text{Dir}(\boldsymbol{\alpha})$ over $\mathbf{p}$, such that $\mathbf{p} \sim \text{Dir}(\boldsymbol{\alpha})$. Below, we define and discuss the relationships between four loss functions: cross-entropy (CE), focal loss, UCE, and our proposed UFCE loss.
> > >
> > > 1. **Cross-Entropy Loss (CE):**
> > >    $$
> > >    \mathcal{L}^{\text{CE}}(\mathbf{p}, \mathbf{y})
> > >    = -\sum_{c=1}^C y_c \log p_c.
> > >    $$
> > >
> > > 2. **Focal Loss:**
> > >    $$
> > >    \mathcal{L}^{\text{Focal}}(\mathbf{p}, \mathbf{y} )
> > >    = -\sum_{c=1}^C (1 - p_c)^{\gamma} y_c \log p_c,
> > >    $$
> > >    where $\gamma$ is a hyperparameter. The focal loss emphasizes harder-to-classify examples by modulating the loss contribution of well-classified samples.
> > >
> > > Both CE and focal loss are designed for deterministic class probability vectors $\mathbf{p}$, typically produced by softmax neural networks.
> > >
> > > 3. **Uncertainty Cross-Entropy Loss (UCE):**
> > >    In ENNs, the output is a Dirichlet distribution $\text{Dir}(\boldsymbol{\alpha})$ over $\mathbf{p}$. The UCE loss generalizes CE by taking its expectation under this distribution:
> > >    $$
> > >    \mathcal{L}^{\text{UCE}}(\boldsymbol{\alpha}, \mathbf{y})
> > >    = E_{\mathbf{p} \sim \text{Dir}(\boldsymbol{\alpha})} \left[ -\sum_{c=1}^C y_c \log p_c \right]
> > >    = \sum_{c=1}^C y_c \left( \psi(\alpha_0) - \psi(\alpha_c) \right),
> > >    $$
> > >    where $\psi(\cdot)$ is the digamma function, $\alpha_0 = \sum_{c=1}^C \alpha_c$, and $\alpha_c$ represents the Dirichlet parameter for class $c$. This loss encourages high evidential support for the true class $\alpha_{c^*}$ while reducing it for others.
> > >
> > > However, UCE does not account for data imbalance, limiting its effectiveness in such scenarios.
> > >
> > > 4. **Uncertainty Focal Cross-Entropy Loss (UFCE):**
> > >    To address data imbalance in ENNs, we introduce the UFCE loss, defined as the expectation of focal loss under the Dirichlet distribution:
> > >    $$
> > >    \mathcal{L}^{\text{UFCE}}(\boldsymbol{\alpha}, \mathbf{y})
> > >    = E_{\mathbf{p} \sim \text{Dir}(\boldsymbol{\alpha})} \left[ -\sum_{c=1}^C (1 - p_c)^{\gamma} y_c \log p_c \right]
> > >    = \frac{B(\alpha_{0}, \gamma)}{B(\alpha_0 - \alpha_{c^*}, \gamma)}
> > >       \left[ \psi(\alpha_0 + \gamma) - \psi(\alpha_{c^*}) \right],
> > >    $$
> > >    where $B(\cdot)$ is the Beta function and $\gamma$ is a hyperparameter. Notably, UFCE reduces to UCE when $\gamma = 0$.
> > >
> > > In summary, the UFCE loss integrates the benefits of UCE with the ability to handle data imbalance, marking a significant advancement in training ENNs effectively. We will incorporate these clarifications and detailed insights into the revised version for enhanced understanding.

---

### Official Review · Reviewer_QCfp · 2024-11-03

**Soundness:** 3
**Presentation:** 2
**Contribution:** 3
**Rating:** 6
**Confidence:** 2

**Summary:**

This paper presents a thorough study on predictive uncertainty quantification of BEVSS and introduces the Uncertainty-Focal-Cross-Entropy (UFCE) loss function to enhance model calibration.  Experiments demonstrate that the uncertainty quantification framework can outperforms baseline methods  in aleatoric uncertainty and calibration in various situations.

**Strengths:**

+ The paper proposes a novel loss function, Uncertainty-Focal-Cross-Entropy (UFCE), specifically designed for highly imbalanced data, which significantly enhances model calibration and segmentation accuracy.
+ The introduction of an uncertainty-scaling regularization term improves both uncertainty quantification and model calibration for BEV segmentation.
+ The comprehensive  experiments are conducted across three popular datasets, two representative backbones and five uncertainty quantification methods.
+ The theoretical proof is solid.

**Weaknesses:**

- Without pseudo OOD, the model performs not well. It is crucial to elucidate the criteria for selecting pseudo OOD data and to discuss its impact on the model's generalization capabilities. Furthermore, an explanation of how varying choices of pseudo OOD data might influence model training would be beneficial. What criteria were used to select the pseudo-OOD data? How sensitive is the model performance to different choices of pseudo-OOD data? Did the authors conduct any ablation studies with different types of pseudo-OOD data?
- As indicated in A.3.1, the selection of parameters like λ and γ varies across different datasets and backbone networks. The experiments should include a comprehensive hyperparameter analysis to illustrate the optimal setting of these parameters for diverse datasets and models. Please discuss any patterns observed in optimal hyperparameter settings across datasets and backbones. If possible, propose guidelines for selecting these hyperparameters on new datasets or architectures.
- While the proposed framework demonstrates notable enhancements over the baseline models, it would be strengthened by a comparison with the latest methods in uncertainty quantification. Is there any specific reason of not choosing more recent baselines instead of LSS and CVT?

**Questions:**

- It would be better to clarify the meaning and difference between OOD and pseudo-OOD.
- It is also better to provide the clear definition on the evaluation metrics.
- Is there anything missing in Appendix A.5? There are two empty subsections.

---

> ### Author Response · Authors · 2024-11-29
> **Response to Weakness 1&2**
>
> **W1-Q1: Criteria for Selecting Pseudo-OOD Data.**
>
> We initially adhered to the criteria outlined in the benchmark study by Franchi et al. (2022), an innovative work in the field of autonomous driving, to identify candidate OOD classes. In this context, OOD data typically encompasses less frequently encountered dynamic objects (e.g., motorcycles, bicycles, bears, horses, cows, elephants) and static objects (e.g., food stands, barriers) that are distinct from primary segmentation categories like vehicles, road regions, and pedestrians. These objects were designated as candidate OOD classes. Subsequently, in our experiments, we randomly partitioned these candidate classes into pseudo-OOD and true OOD categories.
>
> It is worth noting that the OOD benchmark dataset MUAD (https://muad-dataset.github.io/) provided by Franchi et al. (2022) was collected using a simulator based on front-camera imagery for image segmentation. However, it does not include a collection of images from multiple cameras or labels for BEV segmentation. To address this, we adopted a similar procedure and generated a BEV segmentation dataset with OOD objects using the well-established CARLA simulator.
>
> Reference:
> Franchi, Gianni, et al. "MuAD: Multiple uncertainties for autonomous driving, a benchmark for multiple uncertainty types and tasks." arXiv preprint arXiv:2203.01437 (2022).
>
> **W1-Q2: Sensitivity of Model Performance to Pseudo-OOD Data Selection.**
>
> For the nuScenes and Lyft datasets, we adopted the following configuration for both datasets: “motorcycle” was designated as the true OOD class, while “bicycle” served as the pseudo-OOD class.
>
> To evaluate the sensitivity of model performance to different choices of pseudo-OOD and true OOD classes, we introduced an additional configuration for the nuScenes dataset: “traffic cones, pushable/pullable objects, and motorcycles” were selected as pseudo-OOD classes, and “barriers” were designated as the true OOD class.
>
> For the CARLA dataset, we utilized the simulator to randomly position various animal objects (e.g., deer, bears, horses, cows, elephants) on road regions across different towns. We tested two distinct configurations: (1) “deer” as the true OOD class and “bears, horses, cows, elephants” as pseudo-OOD classes; and (2) “kangaroo” as the true OOD class and “bears, horses, cows, elephants, deer” as pseudo-OOD classes.
>
> Across all configurations, our empirical findings remained consistent, indicating robustness in model performance irrespective of the specific choices of pseudo-OOD data.
>
> **W1-Q3: Ablation Studies on Different Types of Pseudo-OOD Data.**
>
> In the updated PDF, we have included a new section, A.5.5, titled **“Robustness on Selection of Pseudo-OOD,”** on Pages 31 and 32\. This section discusses ablation studies conducted with different types of pseudo-OOD data. The empirical results from these studies are summarized in Table 4 (Page 9\) and Tables 17–18 (Pages 32). These results demonstrate the robustness of the model's performance across various types of pseudo-OOD data, highlighting its adaptability and consistency.
>
> **W2: About hyperparameter analysis.**
>
> Thank you for pointing this out. We have expanded the content in Appendix A.3.1 to address the following:
>
> 1. The optimal hyperparameters used for replicating our reported results across different models and datasets (Table 6).
> 2. The hyperparameter tuning strategy we employed which may serve as a guideline for selecting hyperparameters for new models or datasets.
> 3. A hyperparameter sensitivity analysis for the three parameters ($\lambda$, $\gamma$, and $\xi$) we tuned for the backbone LSS on the nuScenes dataset (Tables 7, 8, and 9).
>
> We summarize the hyperparameter analysis as follows: (details are in Appendix A3.1)
>
> * **$\lambda$ (Pseudo-OOD Regularization):** Larger values of $\lambda$ lead to a decline in segmentation and calibration performance. Initially, increasing $\lambda$ improves AUPR, AUROC, and FPR95 for OOD detection, but performance declines when $\lambda$ becomes too large. This suggests that moderate pseudo-OOD regularization benefits OOD detection, while excessive regularization negatively impacts overall performance.
> * **$\gamma$ (Focal Loss Weighting):** Variation in $\gamma$ does not significantly affect segmentation performance. However, selecting an appropriate value for $\gamma$ leads to better calibration, indicating its importance for ensuring reasonable confidence for predictions.
> * **$\xi$ (Epistemic Uncertainty Scaling):** Higher $\xi$ values generally improve AUROC for OOD detection, demonstrating that appropriately weighting epistemic uncertainty can enhance the model's ability to distinguish between ID and OOD pixels.

---

> ### Author Response · Authors · 2024-11-29
> **Response to Weakness 3 and Questions**
>
> **W3: About more recent baselines instead of LSS and CVT.**
>
> LSS and CVT are two representative BEVSS backbones for BEV segmentation tasks. BEVSS backbones define the model architecture, and in this study, we evaluated the performance of different uncertainty quantification methods based on these two backbones. In the updated PDF, we have clarified the BEVSS backbones and uncertainty quantification baselines used in our benchmark in Section 4.1 and Appendices A.2.1 and A.2.2.
>
> In addition, we incorporated new experiments using a more recent BEVSS backbone, Simple-BEV [1]. The results, presented in Table 16, demonstrate consistent performance when compared to LSS and CVT as backbones. We selected Simple-BEV over the state-of-the-art model PointBEV [2] due to its higher adoption and reliability: Simple-BEV has 90 citations and 496 GitHub stars, compared to PointBEV’s 6 citations and 94 stars. This made Simple-BEV a safer and more reliable choice for obtaining robust results within our limited time frame. Furthermore, as shown in Table 1 of the PointBEV paper, Simple-BEV achieves runner-up performance, further supporting its inclusion in our study.
>
> While our model with Simple-BEV as backbone delivers better segmentation performance compared to LSS and CVT as backbones, no significant improvement in OOD detection performance was observed. For our proposed model, we observed consistent results regardless of whether LSS, CVT, or Simple-BEV were used as the backbone. Our model consistently achieves the best performance in calibration and OOD detection while maintaining comparable performance in misclassification detection.
>
> To our knowledge, no prior work addresses uncertainty quantification specifically for the BEVSS task. We extend the literature in this area in Appendix A.2.2. Our benchmark includes widely used uncertainty quantification methods from traditional deep learning:
>
> * The “Entropy” and “Energy” models, which perform post-hoc processing on predicted logits. The Energy model is especially popular for its adaptability and strong OOD detection performance.
> * The “Dropout” and “Ensemble” models, commonly used as Bayesian frameworks across various domains.
> * The Evidential Neural Network (ENN), known for its interpretability and low computational cost.
>
> References:
>
> [1]: Harley, Adam W., et al. "Simple-bev: What really matters for multi-sensor bev perception?." *2023 IEEE International Conference on Robotics and Automation (ICRA)*. IEEE, 2023\.
> [2]: Chambon, Loick, et al. "PointBeV: A Sparse Approach for BeV Predictions." *Proceedings of the IEEE/CVF Conference on Computer Vision and Pattern Recognition*. 2024\.
>
> **Question 1: Clarifying the Meaning and Difference Between OOD and Pseudo-OOD**
>
> A combination of our answers to the questions (W1-Q1, W1-Q2, W1-Q3) in the first weakness in your review could answer this question. In our updated PDF, we provide the details in Appendix A.4.
>
> **Question 2: Evaluation Metrics.**
>
> We evaluate performance using four evaluation metrics: (1) Pure segmentation via Intersection-over-Union (IoU). (2) Calibration via Expected Calibration Error (ECE). (3) Aleatoric Uncertainty via the misclassification detection to identify the misclassified pixels, measured with Area Under ROC Curve (AUROC) and Area Under PR Curve (AUPR). (4) Epistemic Uncertainty via the OOD detection to identify the OOD pixels measured with AUROC and AUPR.
>
> Thanks for your suggestion, we have added the definitions of these evaluation metrics in the beginning of Appendix A.5.
>
> **Question 3: Missing Content in Appendix A.5.**
>
> Thanks for pointing out the empty subsections in Appendix A.5. We have updated Appendix A.5 in our revised PDF. This section has the following subsections: Full evaluation on Lyft (A.5.1), Quantitative evaluation on CARLA (A.5.2), Quantitative evaluation on nuScenes (Road Segmentation) (A.5.3), Quantitative evaluation on Simpble-BEV on nuScenes (A.5.4), Robustness on the selection of pseudo-OOD (A.5.5), Robustness on corrupted dataset (A.5.6), Robustness on diverse conditions (CARLA) (A.5.7), and Robustness on model initializations (A.5.8), and Qualitative evaluations for model comparison (A.5.9).

---

### Author Response · Authors · 2024-11-29
**General Response**

We would like to thank all the reviewers for their insightful comments and valuable suggestions. In response, we have conducted additional experiments and revised the manuscript to address the feedback which are summarized as follows.

Additional Experiments:

* Evaluation on Uncertainty Quantification Models with a more recent Simple-BEV Backbone on nuScenes (Table 16):
  We evaluated our proposed model and all uncertainty quantification baselines using the more recent bird’s-eye view semantic segmentation backbone, Simple-BEV, on the nuScenes dataset.
  Compared to the Evidential Neural Network (ENN) baseline with UCE loss, our model optimized with the proposed loss function consistently achieves better segmentation, calibration, and misclassification detection performance. It also delivers the best out-of-distribution (OOD) detection AUPR, with a 10% improvement over the runner-up model ENN-UCE, along with the second-best AUROC and fourth-best FPR95.
* Robustness Analysis:
  We expanded the robustness evaluation of our model from several perspectives:
* Hyperparameter Sensitivity (Tables 7-9 in Appendix): We performed a detailed sensitivity analysis to understand the impact of three hyperparameters on model performance.

  Moderate pseudo-OOD regularization (\\lambda) benefits OOD detection but excessive values harm performance, appropriate focal loss weighting (\\gamma) improves calibration without affecting segmentation, and higher epistemic uncertainty scaling (\\xi) enhances AUROC for OOD detection.

* Diverse Urban Layouts and Weather Conditions (Table 20-23 in Appendix): We evaluated our model on CARLA across four towns with diverse urban layouts and 10 weather conditions, including variations in brightness and precipitation. The analysis shows that our model consistently outperforms baselines in OOD detection, calibration, and misclassification detection, regardless of layout or weather variations, with detailed results provided in Appendices A.4 and A.5.7.
* Corrupted Datasets (nuScenes-C) (Table 19 in Appendix)**:** We evaluated our model on the nuScenes-C dataset, which includes seven types of corruption at three severity levels, to assess the robustness to data corruption. Our model outperforms the ENN-UCE baseline in most scenarios, achieving state-of-the-art performance in misclassification and OOD detection tasks. While these results demonstrate the strengths of our approach, they also highlight opportunities for improvement, particularly in managing lower-ranked predictions under severe domain shifts.

To improve clarity and comprehensiveness, we revised the manuscript with the following updates:

1. We clarified the distinction between the terms “backbone” and “uncertainty quantification models”. The "backbone" refers to the network architecture used for bird’s-eye view semantic segmentation (e.g., LSS, CVT, Simple-BEV), while "uncertainty quantification models" (e.g., entropy, energy, Bayesian-based methods, ENN) pertain to the network optimization strategies and uncertainty computation methods. (Section 4\)
2. We expanded the related work section to include studies on uncertainty quantification for camera-view semantic segmentation, calibration, robustness in BEVSS, and downstream applications.(Section A.2.2)
3. We provided additional details about the experimental settings, including dataset statistics, hyperparameters, and OOD configurations, to enhance reproducibility and understanding. (Section A.3,  A.4)
4. We qualitatively compared model performance in segmentation and uncertainty predictions, finding that our proposed optimization loss effectively reduces false high epistemic uncertainties compared to the UCE loss. (Section A.5.9 Figure 4-6)

If you have any questions or concerns, please let us know and we will be happy to address them.

---

### Meta-Review · Area_Chair_9zvT · 2024-12-22

**Metareview:**

The paper provides a detailed and comprehensive investigation on how the uncertainty quantification plays a significant role for BEV segmentation. It provides a benchmark and introduces a "novel" loss. Although some reviewers do not acknowledge the technical contribution of the propose loss (UFCE), the overall good merits of the paper (good motivation, comprehensive experiments) over-weigh the negative concerns (limitted novelty). In AC's perspective, this work is slightly above the acceptance bar. It provides a comprehensive benchmark for BEV segmentation. For further request, please incorporate all the necessary revisions during rebuttal and **open-source / maintain an active benchmark** for the sake of a healthy community.

**Additional Comments On Reviewer Discussion:**

During the rebuttal, authors have been actively engaged in the discussion (although lengthy). There are 5 reviews with 4 reviewer's feedback after author rebuttal. The negative rating does not respond to authors further (second-round) feedback. AC has read this negative feedback carefully, this reviewer is concerned on (a) lack of the latest SOTAs, (b) comparision and discussion with other loss, (c) other detailed implementation. Authors have provided further explanation to address reviewer concerns. Overall the paper is slightly above the acceptance at ICLR.

---

### Decision · Program_Chairs · 2025-01-22

Accept (Poster)